# EINSTEIN FIELDS: A NEURAL PERSPECTIVE TO COMPUTATIONAL GENERAL RELATIVITY

**Sandeep S. Cranganore**[*,1]     **Andrei Bodnar**[*,2]     **Arturs Berzins**[*,1]
**Johannes Brandstetter** [1,3]

[*]Equal contribution
[1]LIT AI Lab, Institute for Machine Learning, JKU Linz, Austria
[2]University of Manchester, United Kingdom
[3]Emmi AI GmbH, Linz, Austria
`{cranganore, berzins, brandstetter}@ml.jku.at`
`andrei.bodnar@student.manchester.ac.uk`

## ABSTRACT

We introduce *Einstein Fields*, a neural representation designed to compress computationally intensive *four-dimensional* numerical relativity simulations into compact implicit neural network weights. By modeling the *metric*, the core tensor field of general relativity, Einstein Fields enable the derivation of physical quantities via automatic differentiation. Unlike conventional neural fields (e.g., signed distance, occupancy, or radiance fields), Einstein Fields fall into the class of *Neural Tensor Fields* with the key difference that, when encoding the spacetime geometry into neural field representations, dynamics emerge naturally as a byproduct. Our novel implicit approach demonstrates remarkable potential, including continuum modeling of four-dimensional spacetime, mesh-agnosticity, storage efficiency, derivative accuracy, and ease of use. It achieves up to a $4,000$-fold reduction in storage memory compared to discrete representations while retaining a numerical accuracy of five to seven decimal places. Moreover, in single precision, differentiation of the Einstein Fields-parameterized metric tensor is up to five orders of magnitude more accurate compared to naive finite differencing methods. We demonstrate these properties on several canonical test beds of general relativity and numerical relativity simulation data, while also releasing an open-source `JAX`-based library: https://github.com/AndreiB137/EinFields, taking the first steps to studying the potential of machine learning in numerical relativity.

## 1 INTRODUCTION

General relativity (GR) describes gravity as the curvature of four-dimensional spacetime, encoded in the metric tensor and governed by the Einstein field equations (EFEs), a system of coupled, nonlinear hyperbolic-elliptic PDEs. Exact solutions are available only for idealized cases, so numerical relativity (NR) has become essential for accurate modeling of astrophysical events. Notable successes of NR include the high-precision modeling of black hole mergers (Abbott et al., 2016a;b;c), high-precision binary neutron star merger simulations (Hayashi et al., 2025), and neutron star–black hole systems (Abbott et al., 2017). NR has also been central to the confirmation of gravitational waves (GWs) detected by LIGO and Virgo interferometers, leading to Nobel-prize–winning discoveries.

Nonetheless, state-of-the-art NR is one of the most computationally intensive domains of scientific computing, requiring massive parallelization on petascale computing infrastructures (Lovelace, 2021; Huerta et al., 2019). This is due to several computational challenges in NR, including adaptive high-resolution spatial discretization and finite-difference (FD) methods on these, which, in addition, are vulnerable to numerical errors in sensitive regions. Moreover, NR simulations are equally storage-intensive, producing up to petabytes of data per run, prohibiting the storage and distribution of simulations on HPC systems (Reed & Dongarra, 2015).

Recent progress in machine learning for scientific computing has shown the potential of hybrid neural-classic workflows (Thuerey et al., 2021; Zhang et al., 2023; Brunton et al., 2020; Li et al.,

2021; Brandstetter et al., 2022; Bodnar et al., 2025; Brandstetter, 2025). In addition, neural fields (NeFs) (Park et al., 2019; Müller et al., 2022; Chen & Zhang, 2019) have emerged as a powerful tool in visual computing for compact and continuous representations of traditionally discrete data, such as images, shapes, and physical fields, with ease of querying and differentiating. This raises the question of whether such hybrid approaches can advance next-generation computational GR workflows, particularly in handling and compressing tensorial quantities and their derivatives defined on adaptive high-fidelity discretizations.

To this end, we propose and investigate `EinFields`, which provide the following contributions:

- **Neural compression.** `EinFields` encode geometric information in compact neural representations with typically fewer than two million parameters. They reproduce metric tensor components with relative precision up to seven decimal digits (and up to nine in favourable coordinate charts). This yields memory-efficient approximations of complex spacetime geometries, with compression factors up to $4000\times$ across analytical and numerical solutions.

- **Discretization-free representation.** `EinFields` are trained on arbitrary point samples, including regular, irregular, and unstructured sets. They provide continuous query access to tensor fields at any resolution by learning these as continuous functions from discrete samples, which removes discretization artefacts.

- **Enhanced tensor differentiation.** As smooth neural functions, `EinFields` support continuous evaluation of higher-order geometric quantities such as Christoffel symbols, Riemann tensors, and curvature invariants via point-wise automatic differentiation (Griewank & Walther, 2008). Initial results suggest that this approach can outperform high-order finite-difference methods on uniform grids, with accuracy gains up to $10^5$ in `FLOAT32`.

- **High-fidelity reconstruction of tests of General Relativity.** We evaluate the physical fidelity of `EinFields` on analytical GR solutions and assess derived observables in addition to standard ML metrics. The models faithfully reproduce key relativistic phenomena, including orbital precession in Schwarzschild and Kerr spacetimes, and allow accurate extraction of gravitational-wave distortions and strain. We further test a BSSN (Baumgarte & Shapiro, 2021) evolution of an oscillating neutron star. Early results indicate that `EinFields` scale to realistic numerical relativity workflows despite the complexity of the solution.

## 2 BACKGROUND

Our work lies at the intersection of two domains: GR, along with its computational framework of NR, and NeFs, a ubiquitous tool from computer vision. While a complete introduction to GR and its mathematical backbone, *differential geometry*, is beyond the scope of this Section (see detailed exposition in Appendix A or succinct version in Appendix D), we stress three key properties that pertain to our work: (i) GR is a field theory of *tensor-valued* quantities, (ii) GR is intrinsically coordinate-independent, and (iii) gravitational physics is entirely encapsulated in the metric and its first two derivatives.

**Tensors.** A rank $(r, s)$ tensor $T$ at a point $x \in \mathcal{M}$ is the multilinear map from $r$ covectors and $s$ vectors to a real number:

$$T : \underbrace{\mathcal{V}^* \times ... \times \mathcal{V}^*}_{r-\text{copies}} \times \underbrace{\mathcal{V} \times ... \times \mathcal{V}}_{s-\text{copies}} \to \mathbb{R} \, . \tag{1}$$

The $r$ vectors and $s$ covectors pair with the respective $r$ covariant and $s$ contravariant components of the tensor. As such, a tensor is an element that lives in a tensor product of *vector* and *dual spaces*, i.e., $T \in (\mathcal{V})^{\otimes r} \otimes (\mathcal{V}^*)^{\otimes s}$. A tensor in a particular choice of *basis* $\{e_{\alpha_n}\}_{1 \leq n \leq r} \in \mathcal{V}$ and $\{\vartheta^{\beta_n}\}_{1 \leq n \leq s} \in \mathcal{V}^*$ is given by $T = T^{\alpha_1 \alpha_2 ... \alpha_r}_{\beta_1 \beta_2 ... \beta_s} e_{\alpha_1} \otimes \cdots \otimes e_{\alpha_r} \otimes \vartheta^{\beta_1} \otimes \cdots \otimes \vartheta^{\beta_s}$ , where $T^{\alpha_1 \alpha_2 ... \alpha_r}_{\beta_1 \beta_2 ... \beta_s} \equiv T(\vartheta^{\alpha_1}, \cdots, \vartheta^{\alpha_r}, e_{\beta_s}, \cdots, e_{\beta_s})$ are the *components* of the tensor in this particular basis and transforms as shown in Eq. (20). A *tensor field* assigns to each point $x \in \mathcal{M}$ a multilinear map, i.e. a tensor, $T_x \in \mathcal{V}_x^{\otimes p} \otimes (\mathcal{V}^*)_x^{\otimes q}$. In appropriate coordinates, its components $T^{\alpha_1 ... \alpha_r}_{\beta_1 ... \beta_s}(x)$ vary smoothly across the manifold.

**General relativity** extends Newtonian gravity with a geometric interpretation of gravity, where *mass and energy tell spacetime how to curve, and curved spacetime tells objects how to move* (Misner et al., 2017). This is formalized by the Einstein's field equations (EFEs)

$$G_{\alpha\beta} + \Lambda g_{\alpha\beta} = 8\pi G \, T_{\alpha\beta} \, . \tag{2}$$

EFEs are a set of 10 coupled non-linear, tensor-valued, second-order partial PDEs and can be viewed as a tensorial generalization of the Newton-Poisson equation for gravity $\nabla^2 \Phi(r) = -4\pi G \rho(r)$ (Misner et al., 2017; Poisson, 2004). In EFEs, $G_{\alpha\beta} = R_{\alpha\beta} - \frac{1}{2} R g_{\alpha\beta}$ is the *Einstein tensor*, formed from the metric tensor field $g_{\alpha\beta}(x^\mu)$, which are solutions of the EFEs and tensorial generalization of the gravitational potential $\Phi(\mathbf{x})$. The *Ricci curvature tensor* $R_{\alpha\beta}$, and the *Ricci curvature scalar* $R$ are related by second derivatives $g_{\alpha\beta}$. Thus, the left-hand side of EFEs is entirely described by the metric and its derivatives, with $\Lambda$ being the cosmological constant. The right-hand side depends on the stress-energy tensor $T_{\alpha\beta}$ describing the matter distribution, with $G$ being Newton's constant.

**Metric tensor and its derivatives.** The metric tensor is a rank $(0, 2)$ symmetric bilinear form $g : T_x \mathcal{M} \times T_x \mathcal{M} \to \mathbb{R}$ that generalizes the notion of an inner product on the tangent space $T_x \mathcal{M}$ of a differentiable manifold $\mathcal{M}$ (Jost, 2008). It enables the computation of angles between vector fields and a means to compute distances via the line element: $ds^2 = g_{\alpha\beta}(x^\mu) \, dx^\alpha dx^\beta$. In GR, the components $g_{\alpha\beta}$ in a particular coordinate system can be seen as a $4 \times 4$ symmetric matrix with *ten* independent components. The metric defines the causal structure and contains all geometric information of spacetime. Importantly, its partial derivatives $\partial$ yield the *Christoffel symbols* $\Gamma_{\alpha\beta\gamma}(x^\mu)$, which describe the notion of parallel transport and defines a *covariant derivative* operation $\nabla_\alpha = \partial_\alpha + \Gamma_\alpha$ (all detailed in Appendix A.3.3). In turn, the connection's derivatives (i.e., metric second-derivatives) yield the *Riemann curvature tensor* $R^\delta_{\alpha\beta\gamma}(x^\mu)$, which encodes tidal forces of gravity. The trace part of $R^\delta_{\alpha\beta\gamma}$ (index contraction w.r.t. metric: $\mathrm{Tr}_g$ – see Eq. (44)) is the *Ricci tensor* $R_{\alpha\beta}$, also a symmetric rank $(0, 2)$ tensor. Its subsequent contraction yields the *Ricci scalar* $R$ (all detailed in Appendix A.3.5). This can be summarized schematically as follows (a more detailed pictorial version is available in Figure 10:

$$g_{\alpha\beta} \xrightarrow{\partial} \Gamma^\gamma_{\alpha\beta} \xrightarrow{\nabla} R^\delta_{\alpha\beta\gamma} \xrightarrow{\mathrm{Tr}_g} R_{\alpha\beta} \xrightarrow{\mathrm{Tr}_g} R. \tag{3}$$

**Higer-order methods.** FD methods with adaptive mesh refinement (AMR) (Berger & Oliger, 1984) have long underpinned tensor calculus in NR, discretizing space and time with high-order stencils (Appendix C). An $n$-th order stencil yields truncation errors of $\mathcal{O}(h^n)$, where $h$ is the grid spacing. Widely used fourth- or sixth-order schemes improve accuracy but incur larger communication costs due to broader stencil footprints in parallelized settings. In contrast, modern NR increasingly opts for *(pseudo-)spectral methods* (Scheel et al., 2025), which represent fields globally through polynomial bases (Fornberg, 1996), yielding an efficiency of up to $1000-5000\times$ faster on CPUs than FD approaches on GPUs at comparable accuracy (Rashti et al., 2025).

**Neural fields** (NeFs), also known as implicit neural representations (INR) or coordinate-based neural networks, are multi-layer perceptrons (MLP) using very specific activation functions that are memory-efficient, implicit, continuous, infinitely differentiable maps, capable of capturing high-fidelity detail across complex domains (Xie et al., 2021; Essakine et al., 2024). Some well known NeFs are SIREN with sinusoidal activations (Sitzmann et al., 2020), WIRE with Gabor wavelet activations (Saragadam et al., 2023) for example. These properties have motivated their primary development and adoption in computer vision domains for representation, generation, and inversion tasks. Considering scientific computing domains, NeFs, when integrated with physics-informed losses (e.g., constraints and conservation laws), can be used for solving forward and inverse problems, including spatiotemporal dynamics governed by PDEs. In these settings, NeFs effectively act as PDE solvers, more commonly referred to as physics-informed neural networks (PINNs) (Raissi et al., 2019; Karniadakis et al., 2021; Wang et al., 2025b).

## 3 METHOD – PARAMETRIZING TENSOR FIELDS VIA EINFIELDS

Consider the four-dimensional spacetime $(\mathcal{M}, g)$ (a manifold equipped with a metric) corresponding to an exact or numerical solution to the EFEs[1]: $G_{\alpha\beta} = 8\pi G \, T_{\alpha\beta}$. An `EinField` models the 10 independent components of the metric tensor field as a compact NeF, ultimately mapping the spacetime coordinates $x \equiv (x^0, x^1, x^2, x^3)$ to the symmetric rank $(0, 2)$ metric tensor field:

$$\hat{g} : x \in \mathcal{M} \to g_{\alpha\beta}(x) \in \mathrm{Sym}^2(T_x^* \mathcal{M}). \tag{4}$$

---

[1]From now on, we omit the cosmological constant term $\Lambda g_{\alpha\beta}$.

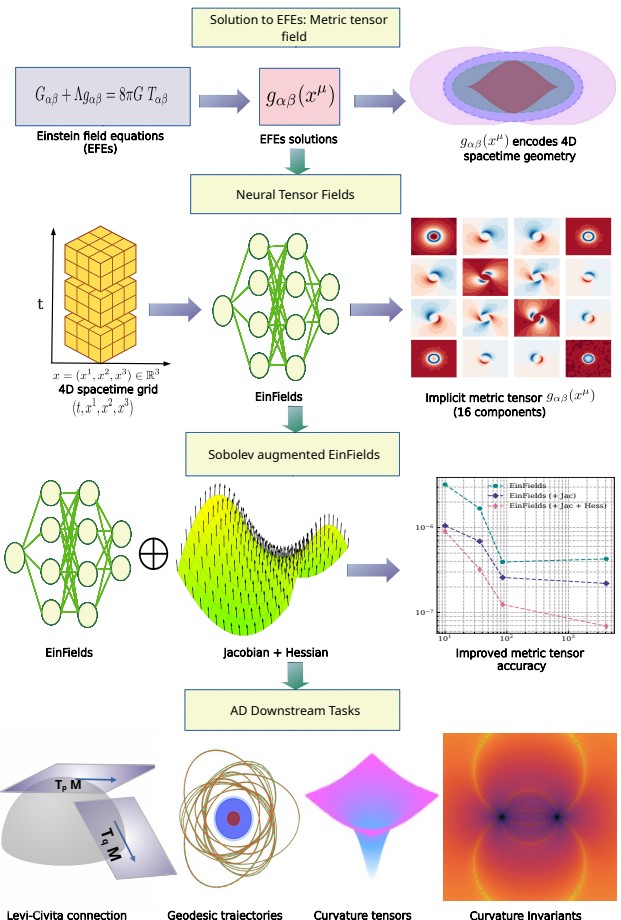

Figure 1: A conceptual overview of `EinFields` training and downstream pipeline. (i) **Premise:** The Einstein field equations (EFEs) in Eq. (2) are highly non-linear PDEs defined on a 4D spacetime manifold, describing the geometric nature of gravitation. Their solutions define the metric tensor field $g_{\alpha\beta}(x^\mu)$, which encodes the full spacetime geometry and serves as a tensorial generalization of the gravitational potential. In this work, we parametrize $g_{\alpha\beta}(x^\mu)$ using a neural network. (ii) **Training:** The training is conducted on the metric tensor fields defined on 4D spacetime points, such as uniform or hierarchical grids. `EinFields` instead fit a continuous signal on these discrete representations, thus modeling 4D spacetime as a continuum, and returning the metric tensor field for a 4D spacetime query coordinate $p \equiv (t, x) \in \mathcal{M}$ at arbitrary resolution. (iii) **Sobolev supervision:** The reconstruction quality of the metric and its derivatives is improved by augmenting Sobolev losses, i.e., metric Jacobian (neighborhood structure) and Hessian (curvatures). (iv) **Validation and downstream tasks:** Sobolev improved `EinFields`' AD-based derivatives enable accurate point-wise retrieval of differential geometric quantities, such as the Levi-Civita connection (covariant derivative), geodesics, curvature tensors, and their invariants.

We deploy an MLP $\hat{g}_\theta$ with parameters $\theta$, denoted $\hat{g}$ for simplicity, to over-fit on the ground truth tensor field. Methodologically, this enables directly compressing the entire geometric information into storage-cheap NN weights, yielding continuous access (different from the training points) at arbitrary resolution of the metric and its non-trivial tensor differentiation (e.g., for Lie derivative or covariant derivatives) information devoid of mesh (re)construction on curvilinear manifolds. This generalizes to an arbitrary rank $(r, s)$ tensor field $T^{\alpha_1\cdots\alpha_r}_{\beta_1\cdots\beta_s}(x^\mu)$. Thus, `EinFields` posit a neural alternative to address one or more of the challenges associated with traditional methods (typically utilizing higher-order finite differencing schemes) in NR by not relying on costly spatiotemporal discretizations. Our framework should be considered as a special case of *Neural Tensor Fields*.

## 3.1 DISTORTION

We define the *distortion* as the algebraic deviation of the spacetime metric from flat space,

$$\Delta_{\alpha\beta} = g_{\alpha\beta} - \eta_{\alpha\beta} , \tag{5}$$

with the Minkowski background $\eta_{\alpha\beta}$ in that particular coordinate system. From a learning perspective, this decomposition acts as a preprocessing step that removes the offset (fixed background metric). Flat contributions, that may even dominate numerically (e.g., $g_{tt} \sim 1/r$, $g_{\theta\theta} \sim r^2$), are removed, leaving only the non-trivial geometric content, such as the curvature. Thus, the network focuses its representational capacity on meaningful deviations rather than redundantly relearning flat-space structure. As we show in 4.2, this improves scaling, accelerates convergence, and emphasizes the dynamic, physically relevant features of the metric during training.

## 3.2 RETRIEVING PHYSICS VIA NEURAL TENSOR FIELD DERIVATIVES

**Higher derivative losses.** Beyond the metric tensor itself, its first- and second-order derivatives are critical to GR, as they govern geodesic motion, tidal forces, and curvature. Accurate trajectory reconstruction requires point-wise precise evaluation of Christoffel symbols and Riemann tensors. Such a high-fidelity extraction from `EinFields` is facilitated by *Sobolev training* (Czarnecki et al., 2017; Son et al., 2021), a formulation that explicitly incorporates higher derivative losses (Chetan et al., 2024; Wang et al., 2025b) – see Section F.4. The supervision on the metric Jacobian $\partial_\mu g_{\alpha\beta}$ (40 independent components) and Hessian $\partial_\mu \partial_\nu g_{\alpha\beta}$ (100 independent components) rectifies irregularities in the metric field and its derivatives, yielding substantial accuracy gains and consequentially improving the precision of point-wise Christoffel symbols $\Gamma^\gamma_{\alpha\beta}(x^\mu)$ and Riemann tensors $R^\sigma_{\alpha\beta\gamma}(x^\mu)$ queries by up to *two orders* of magnitude. This is given by the modified loss function:

$$\mathcal{L}^g_{\text{Sob}}(\theta) = \mathbb{E}_x \left[ \lambda_0 \|g_{\alpha\beta}(x) - \hat{g}_{\alpha\beta}(x)\|^2 + \sum_{j=1}^2 \lambda_j \left\| \partial_x^{(j)} g_{\alpha\beta}(x) - \partial_x^{(j)} \hat{g}_{\alpha\beta}(x) \right\|^2 \right] , \tag{6}$$

with $\lambda_j$ being some coefficients and $\partial_x^{(1)} \equiv \partial_\mu$ and $\partial_x^{(2)} \equiv \partial_\nu \partial_\mu$ written in a succinct notation. Instead of implementing higher-order FD stencils, our framework enables access to exact higher-order tensor derivatives via AD. This is illustrated in the AD workflow for differential geometry in Figure 2.

**Reconstructing dynamics.** Free-fall trajectories around massive objects follows a geodesic motion Eq. (56), which depends on Christoffel symbols $\Gamma(g, \partial g)$. In our workflow (Figure 2), `EinFields` reconstruct $\hat{\Gamma}(\hat{g}, \partial\hat{g})$ with Jacobian supervision, enabling $\nabla_\alpha = \partial_\alpha + \hat{\Gamma}_\alpha$ and, thus, accurate modeling of geodesic path and direct measurement of curvature via curvature tensors become possible.

**Characterizing intrinsic geometry.** Beyond dynamics, `EinFields` must reproduce the intrinsic geometry encoded in curvature tensors and invariants. This constitutes Riemann $R_{\alpha\beta\gamma\delta}$ tensor and the associated geodesic deviation Eq. (60), Weyl $C_{\alpha\beta\gamma\delta}$, Ricci $R_{\alpha\beta}$ tensors, scalar curvature $R$, and invariants such as the Kretschmann scalar $\mathscr{K}$ (detailed in A.3.5.2). With Jacobian and Hessian-level supervision (see tomography plots in Appendix F.8 showcasing improvements due to higher-derivative loss inclusion), the learned fields achieve strong agreement with analytic solutions across the domain, except near singularities $\lim_{r\to 0} \left(\frac{1}{r^n}\right) \forall n \in \mathbb{N}$, where curvature becomes infinite.

## 4 EXPERIMENTAL VERIFICATIONS

The performance of `EinFields` is assessed along two axes: (i) *compression*, i.e., meaningful reduction in permanent storage requirements as compared to high-resolution spatiotemporal meshes utilized in NR simulations which includes the metric (10 independent components) and higher-order derivatives (20–100 components) across millions of collocation points, and (ii) *reconstruction fidelity*, evaluated through key GR benchmarks: geodesic dynamics around compact objects (Appendix D) and curvature diagnostics such as the curvature scalars. The evaluation criteria used are either mean absolute error (MAE) or relative $\ell_2$ (Rel. $\ell_2$) between the ground truth and NeF parametrized tensors, which is detailed in Appendix F.1.

**4D training and validation data.** We overfit the NeFs over synthetic data generated from exact 4D analytic solutions (see Appendix B explaining each of these use-cases) of the EFEs: (i) Schwarzschild

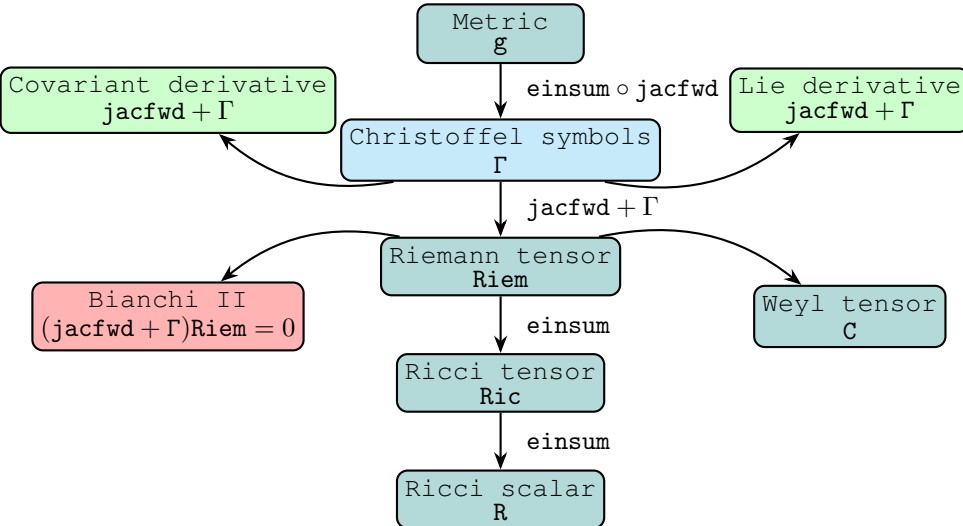

Figure 2: The directed-acyclic graph (DAG) for computing the differential geometric quantities from the metric tensor g in analogy to Figure 10 and Eq. (3) . The transformations include repeated differentiation implemented via *forward-mode Jacobian* jacfwd operations and tensor index manipulation using einsum. Tensors are in depicted in teal blue, connection in light-blue, tensor derivatives in green and conservation laws (Bianchi identities $(\texttt{jacfwd} + \Gamma)\texttt{Riem} = 0$) in red.

(static, spherically symmetric solution), (ii) Kerr (rotating, with spin parameter $a > 0$, oblate spheroidal), and (iii) propagating gravitational waves (GW) (time-varying, linearized gravity metric). Details regarding training and validation grid resolutions and parameter ranges (e.g., mass $M$, spin parameter $a$, etc) used to generate the distortion part of the metric, as shown in Eq. 5, are described in Appendix F.2.

**Training specifics.** For our tasks, the most effective architectures are MLPs with SiLU activations (Elfwing et al., 2018), which excel under derivative-based supervision. Given the sensitivity of training dynamics to the choice of optimizer (Wang et al., 2025a), we employ SOAP (Vyas et al., 2025), a scalable quasi-Newton method shown to enhance gradient alignment in PINNs (Raissi et al., 2019). We adopt a GradNorm-based scheme (Chen et al., 2018), enforcing unit-norm gradients mitigating gradient imbalances induced by Sobolev supervision. The explored models span widths and depths from $64 \times 3$ to $512 \times 8$, totaling less than $1.9 \times 10^6$ parameters ($\sim 7\,\texttt{MiB}$). The NeF training ranges between 100s (w/o Sobolev training) to 2000s (Sobolev training including metric Hessian) on a NVIDIA H200 SXM GPU.

### 4.1 ACCURACY AND STORAGE EFFICIENCY OF EINFIELDS: METRIC AND ITS DERIVED QUANTITIES.

Table 1: Performance evaluation (measured in Rel. $\ell_2$ and MAE metrics) and storage efficiency of EinFields parametrized metric tensor fields under different representations (i.e., with and w/o Sobolev trainings). The model with the lowest MAE is selected in each row.

| Representation | Rel. $\ell_2$ | MAE | Storage | Compression |
|---|---|---|---|---|
| EinFields | $(1.08 \pm 0.06)$e-6 | 2.11e-6 $\pm$**0.07**e-6 | **85** KiB | **4035** |
| EinFields (+ Jac) | $(3.37 \pm 0.84)$e-7 | $(9.49 \pm 1.51)$e-7 | 1.1 MiB | 311 |
| EinFields (+ Jac + Hess) | $(1.88 \pm 0.16)$e-7 | $(9.07 \pm 1.71)$e-7 | 202 KiB | 1698 |
| Explicit grid | — | — | 343 MiB | — |

Accurate reconstruction of higher-rank tensors from neural tensor fields is critical for recovering geodesics, tidal forces, and related physical quantities. We evaluate EinFields by comparing accuracy–memory tradeoffs for the metric (Figure 3a) and Christoffel symbols (Figure 3b), using the evaluation numbers reported in Table 1. Against higher-order FD baselines in FLOAT32, EinFields achieves systematically lower MAE, avoiding truncation and instability issues that limit FD stencils at small $h$. Through AD, we compute different derived quantities listed in Table 2 showing that EinFields outperforms FD stencils by $10 - 10^5$ in accuracy across these quantities.

Table 2: Performance evaluation of `EinFields` reconstructed differential geometric quantities for the Schwarzschild geometry in spherical coordinates. Columns 2–3 report memory usage for full and symmetry-reduced components. Columns 4–6 report MAE relative to analytical solutions: FD stencils for `h = 0.01` on the ground-truth (GT (FD)), `EinFields` via AD, and AD applied directly to the analytic solution (GT (AD)).

| Geometric quantity | Storage [GiB] | | MAE | | |
| --- | --- | --- | --- | --- | --- |
| | Full | Sym. | GT (FD) | EinFields (AD) | GT (AD) |
| Christoffel symbol | 2.6 | 1.6 | 5.37e-6 | **(9.98 ± 2.12)e-7** | 5.83e-9 |
| Riemann tensor | 10.4 | 0.8 | 1.78e-2 | **(1.25 ± 0.30)e-6** | 2.86e-8 |
| Weyl tensor | 10.4 | 4.0 | 1.72e-2 | **(1.67 ± 1.11)e-5** | 5.89e-8 |
| Ricci tensor | 0.6 | 0.4 | 4.81e-2 | **(9.66 ± 2.86)e-6** | 9.02e-8 |
| Ricci scalar | 0.04 | 0.04 | 5.35e-2 | **(3.80 ± 1.72)e-5** | 1.31e-8 |
| Kretschmann invariant | 0.04 | 0.04 | 1.33e-2 | **(1.07 ± 0.46)e-5** | 3.32e-8 |
| Bianchi identity (II) | — — | — — | 1.68e-2 | **5.00e-8** | 4.81e-8 |

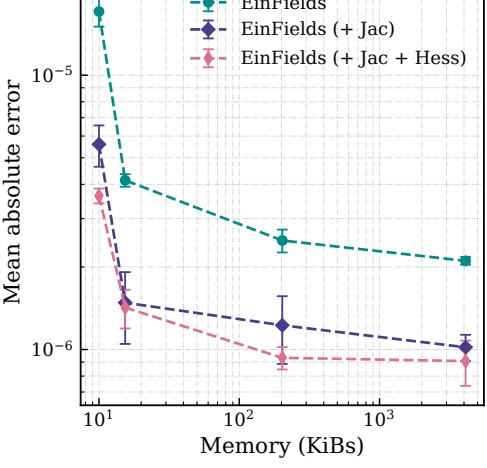

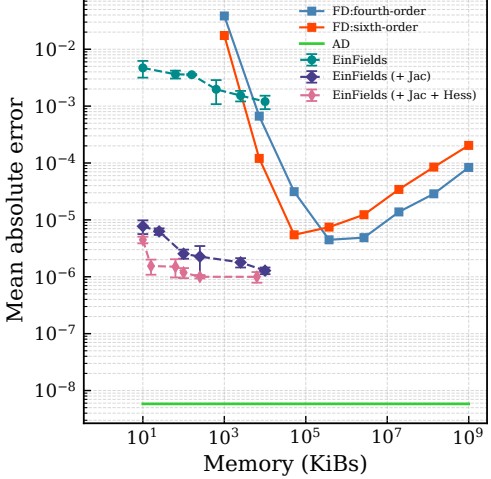

(a) Accuracy of `EinFields` for different NN sizes (measured in KiBs required to store all `FLOAT32` parameters) as trendlines for different training schemes. Each metric tensor component has MAE values ranging from 1e-5 to 1e-6. Apart from high accuracy, one additionally acquires $\sim 1000 - 4000$ times compression factors in storage memory gain, as detailed in Table 1.

(b) Accuracy of `EinFields`' Christoffel symbols derived from the metric tensor, shown as trendlines for different training schemes in `FLOAT32` (KiBs). The trendlines show different training schemes (with and without Sobolev training) with MAE values ranging from 1e-2 to 1e-6. We benchmark against fourth-order and sixth-order stencils with truncation errors $\mathcal{O}(h^5)$ and $\mathcal{O}(h^7)$, respectively. Our framework outperforms FD stencils by more than an order of magnitude in accuracy.

Figure 3: Trendlines of accuracy versus storage memory (KiB) requirement for the metric tensor and Christoffel symbols. For the explicit grid storage this is computed as `num of grid collocation points × 4`, with 4 bytes for single precision (`FLOAT32`). For the NeFs, this corresponds to the storage memory of the compact implicit NN weights.

**Reconstructing seminal tests of GR.** As a part of validation, we demonstrate high-fidelity reconstruction of seminal tests associated with general relativistic dynamics on curved manifolds: (i) geodesics curves around Schwarzschild black hole – Figs.(4a, 4b, 4c) and its ray-traced rendering – Fig. 6; (ii) Kerr solutions – Figs.(4d, 4e, 4f); (iii) geodesic deviation describing oscillating ring of test particles due to GW distortions – Fig. 5 (all detailed within Appendix F.6). Each of these use-cases shows excellent agreement with the analytic results, although they are subject to accumulated temporal rollout errors (see Appendix F.6 for specifics) and are heavily affected by floating-point errors requiring `FLOAT64` precision.

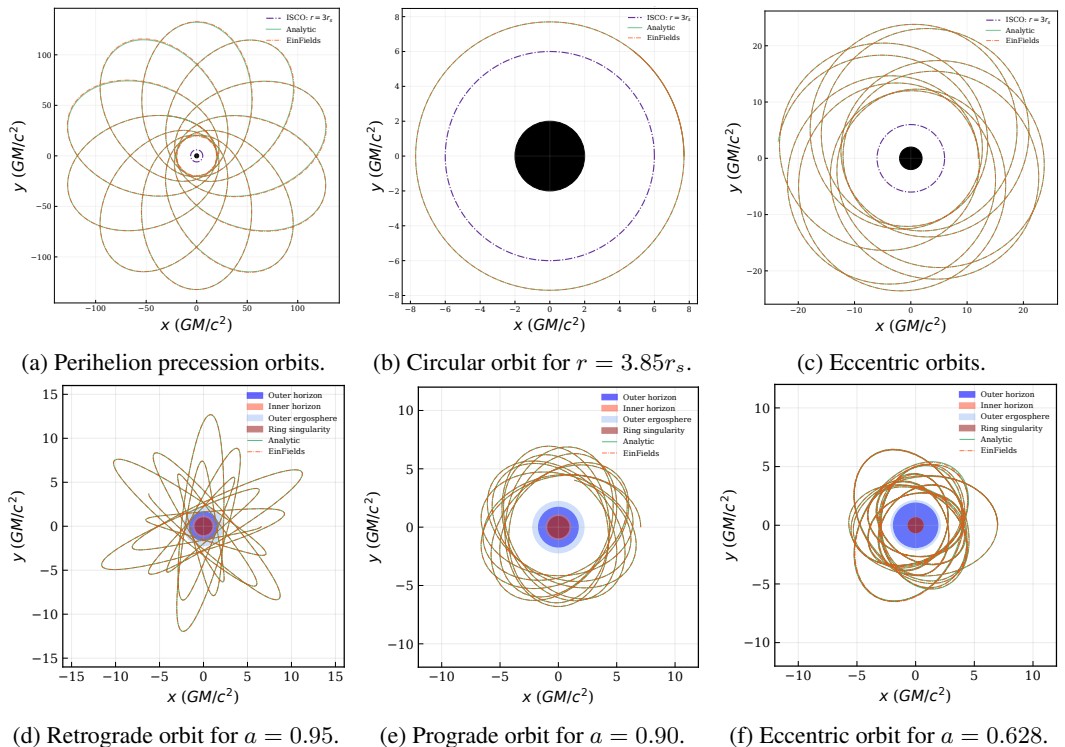

(a) Perihelion precession orbits.  (b) Circular orbit for $r = 3.85r_s$.  (c) Eccentric orbits.

(d) Retrograde orbit for $a = 0.95$.  (e) Prograde orbit for $a = 0.90$.  (f) Eccentric orbit for $a = 0.628$.

Figure 4: **Row 1:** Geodesics in Schwarzschild spacetime simulated in spherical coordinates – Eq. (70). **Row 2:** Geodesics in Kerr spacetime simulated in Boyer–Lindquist coordinates – Eq. (80). Distinct regions of the geometry are indicated in solid colors. Green solid lines represent ground-truth geodesics, while the red dotted lines represent our NeFs reconstructed orbits.

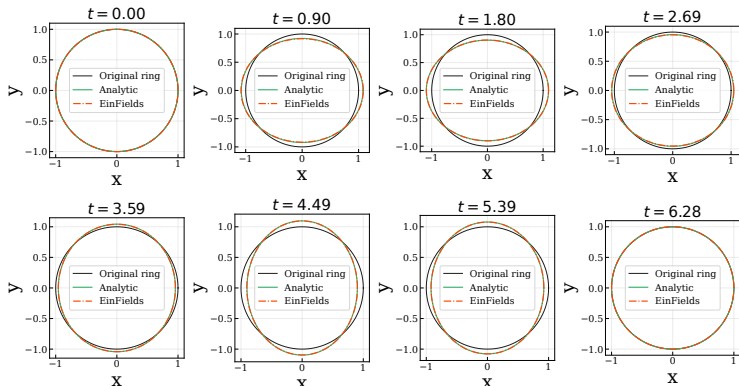

Figure 5: Spatial deformations (stretching and squeezing) of a circular ring of test particles due to "+" polarized gravitational wave – Eq. (111). The NeF-reconstructed $h_+ \cos\left(\omega(t - z)\right)$ and $h_\times \cos\left(\omega(t - z)\right)$ show excellent agreement with the analytic geodesic deviation for the linearized gravity use case. See Table 10 for a quantitative evaluation.

**Curvature associated reconstruction.** Additionally, we demonstrate high-precision reconstruction results for other seminal GR phenomena such as *gravitational waves extraction* via Weyl scalar $\Psi_4(r, t)$, Kerr metric *ring-singularity* structure (Kretschmann scalar) captured by EinFields. These are discussed in detail within Appendix F.6.

**Oscillating neutron star NR simulation.** Beyond analytical solutions, we evaluate EinFields on a widely used, dynamical test in numerical relativity: the oscillatory evolution of a perturbed neutron star. Unlike the previous cases, this problem is time-dependent, involves matter-spacetime coupling, has no analytical solution, and is computed using *fixed mesh refinement* (FMR), thus providing a more realistic NR setting for assessing model performance. The details of this setup are

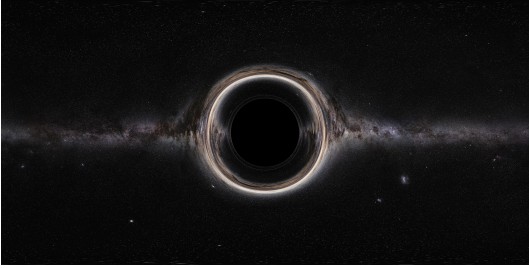

Figure 6: Neural rendering of a Schwarzschild black-hole in front of a celestial background. The render is constructed by tracing the geodesics of a `EinFields` represented metric showcasing its compatibility with complex downstream tasks.

provided in Appendix G and summarized here. We perform a simulation of a non-rotating neutron star of gravitational mass $M = 1.4\,M_\odot$, described by the Tolman–Oppenheimer–Volkoff (TOV) equations, which is evolved under a small initial perturbation. The coupled evolution of relativistic hydrodynamics and spacetime produces the characteristic oscillation spectrum of the star. This test serves as a standard benchmark for general-relativistic simulations in the *Einstein Toolkit* (Löffler et al., 2012).

**Fixed mesh refined training data.** The Einstein Toolkit software performs time-evolution using the BSSN formulation of Einstein's equations (Shibata & Nakamura, 1995; Baumgarte & Shapiro, 1998). The spacetime domain is discretized using *fixed mesh refinement* (FMR) (Schnetter et al., 2004; Hayashi et al., 2025), in which a hierarchy of nested grids provides higher resolution only where needed. The simulation employs five refinement levels, $\{\text{rl0}, \dots, \text{rl4}\}$ (cf. Fig 7b), with `rl0` covering the full domain and `rl4` resolving the stellar interior. The evolution proceeds to a final time of $T = 1000M$, with data written every $\Delta t = 1.0$; finer levels use proportionally smaller timesteps to satisfy the CFL condition (Courant et al., 1967). Training data (detailed in Table 12) at each time slice are obtained by collecting all spatial points $(x_i, y_i, z_i)$ from the refinement levels while discarding any duplicate points lying inside finer patches. This produces a single non-uniform multi-resolution grid.

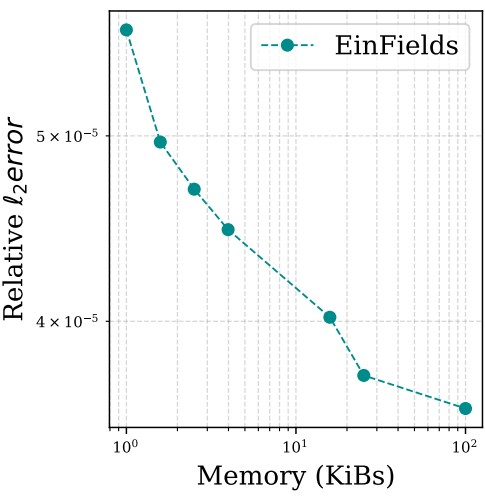

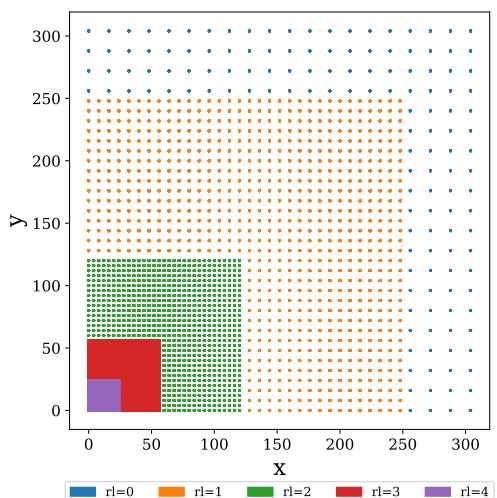

(a) Accuracy vs. compression of `EinFields` trained on the NR simulation data (w/o Sobolev training). The trendline indicates a maximum neural compression of $\sim 2000\times$ for Rel.$\ell_2$ of 3.60e-5, as detailed in Table 3.

(b) FMR (excluding ghost zones) increases resolution near the center of the neutron star. The refinement levels (rl) are evolved independently with progressively finer $\Delta \mathbf{x}$ and $\Delta t$.

Figure 7: `EinFields` applied to the NR simulation of an oscillating neutron star evolved numerically using a BSSN solver on a fixed-mesh refinement (FMR) grid.

**Training results.** Table 3 and Figure 7a summarize the evaluation for the metric, comprising of compression and accuracy for the best-performing 6x256 model. The evaluation procedure is

explained in detail in Appendix G. For completeness, the corresponding Christoffel symbol results are also reported in Table 13.

Table 3: Performance of EinFields tested on a single stable neutron star numerical relativity simulation. The table reports the relative $\ell_2$ error, MAE, storage footprint, and resulting compression ratio obtained when training EinFields on the coalesced fixed–mesh–refined (FMR) grid constructed from the simulation. EinFields achieves a compression ratio of approximately $2000\times$ while maintaining low reconstruction error, as indicated by the reported relative $\ell_2$ and MAE values.

| Representation | Rel. $\ell_2$ | MAE | Storage | Compression |
|---|---|---|---|---|
| EinFields | 3.60e-5 | 5.98e-5 | 1.4 MiB | 2121 |
| EinFields (+ Jacobian) | **6.95e-6** | **9.88e-6** | 1.4 MiB | **2121** |
| FMR coalesced grid | — | — | 2.9 GiB | — |

## 4.2 ABLATION STUDIES

We report ablation results relative to our best-performing baseline configuration, systematically examining the effects of matrix representations (full metric instead of distortions), activation functions, optimizers, learning rate schedulers, and Sobolev regularization. All evaluations are conducted in spherical coordinates.

Table 4: Ablation results for the Schwarzschild metric. Row 2 trains on the full metric (Eq. (70)) instead of its distortion (Eq. (98)). Row 3 and 4 ablate the learning rate schedule and the optimizer, respectively. Row 5 replaces SiLU with WIRE (Saragadam et al., 2023), a well-performing activation function for NeFs. Rows 6–7 ablate the derivative supervision, i.e., Sobolev training.

| Ablation | | Rel $\ell_2$ | Wallclock time [s] |
|---|---|---|---|
| Baseline: | Metric distortion, Jac + Hes, SiLU, SOAP, Cosine LR | 1.40e-7 | 1400 |
| Metric distortion $\Delta_{\alpha\beta} \rightarrow$ Metric $g_{\alpha\beta}$ | | 2.13e-6 | 1407 |
| Cosine $\rightarrow$ Const. LR | | 2.37e-5 | 1397 |
| SOAP $\rightarrow$ ADAM | | 4.16e-6 | 1150 |
| SiLU $\rightarrow$ WIRE | | 4.12e-6 | 3045 |
| Jac + Hes $\rightarrow$ Jac | | 1.51e-7 | 509 |
| Jac + Hes $\rightarrow$ - | | 2.37e-7 | 364 |

## 5 CONLUSION

EinFields introduces the first implicit neural framework for compressing four-dimensional relativity simulations with differentiable modeling. By combining neural tensor fields with automatic differentiation, it offers a scalable, discretization-free, resolution-invariant alternative to grid-based methods that preserves physics and is suitable for downstream tasks. Our framework achieves accuracies of 1e−7 − 1e−9 (see Table 8) with compression factors of 1000 − 4000 relative to uniform and FMR multi-resolution/heterogeneous grids. Derived quantities show up to five orders of magnitude improvement in tensor derivatives (in FLOAT32) over higher-order FD schemes.

**Limitations.** Compression with NeFs remains lossy: even with FLOAT64 training, Rel. $\ell_2$ errors below 1e−9 are currently unattainable. At present, the framework surpasses FD methods only in single precision. These errors propagate into Christoffel symbols, causing long-time divergence in geodesic solvers (see Figures (16a, 16b)) and curvature tensors. Moreover, the query time of these compressed NeF representations is non-trivial (a few ms for a batch of $10^5$ queries, see Figure 23), which can be prohibitive in downstream tasks that require repeated sequential evaluation. While taking the first step toward the integration of ML and NR techniques, our work does not evaluate advanced NR methods, such as adaptive mesh refinement and (pseudo-)spectral solvers.

**Future work.** We plan to extend the application of EinFields to other complex large-scale time-evolving NR simulations (e.g., binary black hole and binary neutron star mergers) and benchmark against advanced classical techniques such as (pseudo)spectral methods.

## ACKNOWLEDGMENTS

We sincerely thank Nils Deppe for valuable feedback on several numerical relativity-related aspects of the paper.

The ELLIS Unit Linz, the LIT AI Lab, the Institute for Machine Learning, are supported by the Federal State Upper Austria. We thank the projects FWF AIRI FG 9-N (10.55776/FG9), AI4GreenHeatingGrids (FFG- 899943), Stars4Waters (HORIZON-CL6-2021-CLIMATE-01-01). We thank NXAI GmbH, Audi AG, Silicon Austria Labs (SAL), Merck Healthcare KGaA, GLS (Univ. Waterloo), TÜV Holding GmbH, Software Competence Center Hagenberg GmbH, dSPACE GmbH, TRUMPF SE + Co. KG.

Sandeep S. Cranganore was supported by the FWF Bilateral Artificial Intelligence initiative under Grant Agreement number 10.55776/COE12.

## REPRODUCIBILITY STATEMENT

We have made every effort to ensure the reproducibility of our results. The codebase, including training scripts, neural field models used, and data generation and preprocessing pipelines, will be made accessible as a zip file in the supplementary material. All experiments can be reproduced using the instructions provided in the repository's `README.md` and `How_to_train_EinFields.md`, with detailed specifications of hyperparameters, optimizer settings etc.

For our synthetically generated analytic solutions data, we provide the essential configuration yaml files with appropriate parameter values in `data_generation/configs` and the data generation scripts are within `data_generation`. Copious example notebooks containing all the validation problems are contained with the folder `example_notebooks`. The blackhole render scripts and visualization can be found within the `bh_render` folder.

Software dependencies are specified in a `requirements.txt` file, and we provide `Conda` virtual environments for ease of setup, especially with the appropriate `CUDA` version. All experiments were run on [specify hardware, e.g., 1× NVIDIA A100 GPUs for prototyping and 1 NVIDIA H200 GPUs for the main runs], with a training time ranging from 100 - 2200 seconds depending on Jacobian, Hessian-inclusion in losses and the specific hardware used.

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

# Appendices

## A  INTRODUCTION TO GENERAL RELATIVITY

This appendix provides the mathematical background and intuition on differential geometry (covering every aspect of the paper and the library), and general relativity. We remark that the reader may appreciate several related works, such as (Jost, 2008; Kobayashi & Nomizu, 1963; Isham, 1999; Lee, 2012) for mathematically rigorous coverage of differential geometry. For more physics-oriented readers, the following books extensively cover general relativity and numerical relativity (Misner et al., 2017; Carroll et al., 2004; Poisson, 2004) as alternative resources. Additionally, the Geometric Deep Learning (GDL) community can also find more ML-centric introduction to differential geometry in the following work (Bronstein et al., 2021; Weiler et al., 2023).

Table 5: Table of notations

| Symbol | Description |
|---|---|
| $\mathcal{M}$ | Arbitrary manifold |
| $\mathscr{M}$ | 4-dimensional spacetime manifold |
| $\eta_{\mu\nu}$ | Flat Lorentzian metric |
| $x^{\mu}$ | Original coordinates |
| $\bar{x}^{\mu}$ | Transformed coordinates |
| $e_{\mu}$ | Basis set |
| $\vartheta^{\mu}$ | Dual basis set |
| $\bar{e}_{\mu}$ | Transformed basis set |
| $\bar{\vartheta}^{\mu}$ | Transformed dual basis set |
| $\dfrac{\partial}{\partial x^{\mu}}$ | Coordinate basis (equivalent to partial derivative operator) |
| $T_{p}\mathcal{M}$ | Tangent space at point $p$ |
| $T_{p}^{*}\mathcal{M}$ | Cotangent space at point $p$ |
| $\Omega^{1}(\mathcal{M})$ | Space of one-forms |
| $\Gamma(T\mathcal{M})$ | Smooth sections of a tangent bundle (collection of vector fields) |
| $\Gamma(T^{*}\mathcal{M})$ | Smooth sections of a cotangent bundle (collection of one-forms) |
| $\Phi^{*}$ | Pullback operation |
| $\Phi_{*}$ | Pushforward operation |
| $\texttt{Riem}(\mathcal{M})$ | Set of (pseudo-)Riemannian metrics on $\mathcal{M}$ |
| $\texttt{Diff}(\mathcal{M})$ | Set of diffeomorphism maps on $\mathcal{M}$ |
| $\times$ | Cartesian (tensor) product |
| $\otimes$ | Kronecker (tensor) product |
| $\mathcal{D}_{v}$ | Directional derivative |
| $\mathcal{L}_{v}$ | Lie derivative with respect to vector field $v$ |
| $\nabla_{\mu}$ | Covariant derivative |
| $\delta_{\nu}^{\mu}$ | Kronecker delta (identity matrix) |
| $c$ | Speed of light |
| $G$ | Newton's constant |

### A.1  FUNDAMENTAL CONCEPTS OF DIFFERENTIAL GEOMETRY & AND TENSOR CALCULUS

The main concepts covered in this appendix are:

1. *Fundamental concepts of differential geometry and tensor calculus*: We introduce contravariance and covariance, and further vector and dual vector spaces. This allows us to define tangent and cotangent spaces.

2. *Tensors and tensor fields*: Next, we define tensors and tensor fields, operations on tensor fields, and the Lie derivative as a generalization of the directional derivative for tensor fields.

3. *Riemannian and Lorentzian geometry*: This is the meat of Appendix A. We introduce 4-dimensional spacetime as a continuous differentiable manifold. Via the metric, we can

define Riemannian manifolds, and finally Lorentzian manifolds as a pseudo-Riemannian manifold. Next, we discuss connections, covariant derivatives, and Christoffel symbols. This is all mathematical background that is required to introduce parallel transport, geodesics, the Riemann curvature tensor, the Ricci tensor, the Ricci scalar, the Weyl tensor, and finally curvature invariants and the stress-energy-momentum tensor. We end with Einstein field equations, reflecting back on the coordinate-independency of GR.

### A.1.1 CONTRAVARIANT AND COVARIANT COMPONENTS

Loosely speaking, an $n-$dimensional vector $v \in \mathbb{R}^n$ can be expanded in its basis as $v = v^1 e_1 + v^2 e_2 + \ldots + v^n e_n$, or $v = v^i e_i$ if we use Einstein sum convention. In general relativity, we write $v = v^\mu e_\mu$, where Greek indices indicate $4-$ dimensional space-time. Thus, in this non-Euclidean setting, it is necessary to distinguish objects that carry an upper index (contravariant) versus objects that carry a lower index (covariant), since they satisfy different geometric properties and transformation laws.

**Definition 1** (**Contravariance of vector components**): Let $X \subset \mathbb{R}^n$ be a coordinate system (frame) that is spanned by a coordinate basis set $\{e_\mu\}_{1 \le \mu \le n}$, i.e., each basis vector can be expressed as $e_\mu = \frac{\partial}{\partial x^\mu}$. A vector $v \in X$ can be expanded in its coordinate basis as $v = v^\mu(x)e_\mu := v^\mu(x)\frac{\partial}{\partial x^\mu}$. When transforming the vector $v$ to a new coordinate system, spanned by another coordinate basis set $\{\bar{e}_\nu\}_{1 \le \nu \le n}$, i.e., $\bar{e}_\nu = \frac{\partial}{\partial \bar{x}^\nu}$, one can express the vector components in the new coordinate system $\bar{v}^\nu = \bar{v}^\nu(\bar{x})$ as

$$\bar{v}^\nu(\bar{x}) = \sum_\mu \frac{\partial \bar{x}^\nu}{\partial x^\mu} v^\mu(x) \, . \tag{7}$$

The ratio of change of the vector components is the inverse of the ratio of the base components. In other words, vector components transform inversely – or contravariantly – with respect to basis transformations, i.e., transform in the opposite way to the change in the coordinate system. Most contravariant objects represent physical quantities like displacement, velocity, and momentum, which must adjust when the coordinate basis changes.

**Definition 2** (**Covariance of basis set**): Let $X \subset \mathbb{R}^n$ be a coordinate system (frame) that is spanned by a coordinate basis set $\{e_\mu\}_{1 \le \mu \le n}$, i.e., each basis vector can be expressed as $e_\mu = \frac{\partial}{\partial x^\mu}$. A vector $v \in X$ can be expanded in its coordinate basis as $v = v^\mu(x)e_\mu := v^\mu(x)\frac{\partial}{\partial x^\mu}$. When transforming the vector $v$ to a new coordinate system, spanned by another coordinate basis set $\{\bar{e}_\nu\}_{1 \le \nu \le n}$, i.e., $\bar{e}_\nu = \frac{\partial}{\partial \bar{x}^\nu}$, then, the basis set itself transforms as,

$$\frac{\partial}{\partial \bar{x}^\nu} = \sum_\mu \frac{\partial x^\mu}{\partial \bar{x}^\nu} \frac{\partial}{\partial x^\mu} \, . \tag{8}$$

Note that we have introduced the concept of contravariant and covariant transformation by the example of vector components and the respective basis set. In general, we speak of contravariant w.r.t. their corresponding basis sets. I.e., contravariant components have covariant basis sets and covariant components have contravariant basis sets. As we introduce next, an object with covariant components is an object of the dual space. These covariant vectors, or covectors, typically represent gradients, such as the gradient of a function. A gradient represents the change w.r.t. an infinitesimal change in a direction. It is intuitive that if we make the direction larger, the change becomes larger as well. In other words, if we change the basis vectors in which we measure this change, the gradient transforms covariantly w.r.t. the basis vectors.

### A.1.2 DUAL SPACE

While the concept of a vector space is well known in the machine learning community, there is a closely associated concept of a *dual vector space* (succinctly called dual space), which is an algebraic dual to the vector space itself with the same dimensions.

**Definition 2** (**Dual vector space**): Let $(\mathcal{V}, +, \cdot)$ be a vector space over a field $F$ (e.g., $\mathbb{R}, \mathbb{C}$). The (algebraic) dual space $(\mathcal{V}^*, +, \cdot)$ is a vector space of linear functionals (maps) $\mathcal{V}^* := \{v^* | v^*(v) = $

$c \, \forall v \in \mathcal{V}, \forall c \in F\}$, satisfying:

$$(v^* + w^*)(v) = v^*(v) + w^*(v) \tag{9}$$

$$v^*(\alpha v + \beta w) = \alpha v^*(v) + \beta w^*(w) \tag{10}$$

$$(cv^*)(v) = c(v^*(v)) \tag{11}$$

for all $v^*, w^* \in \mathcal{V}^*$, $v, w \in \mathcal{V}$, $\alpha, \beta, c \in F$. Elements of the dual space $\mathcal{V}^*$ are sometimes referred to as *covectors* or *one-forms* Bott & Tu (1982).[2].

For example, suppose, we are given a basis $\{e_1, \ldots, e_n\}$ of a vector space $\mathcal{V}$. Then, one can introduce a dual basis set $\{\vartheta^1, \ldots, \vartheta^n\}$ of the dual space $\mathcal{V}^*$. Let $v = (\alpha_1 e_1 + \alpha_2 e_2 + \ldots + \alpha_n e_n) \in \mathcal{V}$, $\forall \{\alpha_i\}_{1 \le i \le n} \in F$. Thus, the action of the linear functionals $\vartheta^i$ on the vector reads: $\vartheta^i(v) = \vartheta^i(\alpha_1 e_1 + \alpha_2 e_2 + \ldots + \alpha_n e_n) = \alpha_i \vartheta^i(e_i) = \alpha^i$, $i = 1, \ldots, n$ and $\vartheta^i(e_j) = \delta^i_j$, where we have used the orthonormality condition, and $\delta^i_j$ is the Kronecker delta symbol. Conceptually, the covector $\phi$ is a (complex-conjugated if $F = \mathbb{C}$) row-vector, which acts on a column vector $\boldsymbol{v}$ to produce $\alpha \in F$. Colloquially speaking, an element of the dual space $\mathcal{V}^*$ "eats up" an element of the vector space $\mathcal{V}$ and returns a scalar (duality pairing).

### A.1.3 TANGENT AND COTANGENT SPACES

**Definition 1** *(Tangent space): Let $\mathcal{M}$ be a smooth ($C^\infty$) manifold of dimension $n$. The tangent space $T_p\mathcal{M}$ at point $p \in \mathcal{M}$ is a set of $d$-dimensional vectors (called tangent vectors) attached at point $p$, defined as $T_p\mathcal{M} := \{(p, v) : v \in \mathbb{R}^d\}$, and carries the structure of a real vector space. Every tangent space is spanned by an ordered basis $\{e_\mu|_p\}_{1 \le \mu \le n} = \left\{ \frac{\partial}{\partial x^1}\big|_p, \ldots, \frac{\partial}{\partial x^n}\big|_p \right\} \in T_p\mathcal{M}$, and vectors can be expanded in this basis as:*

$$v|_p = v^\mu(x) \frac{\partial}{\partial x^\mu}\bigg|_p \,, \tag{12}$$

*where $v^\mu(x)$ are the components of the vector in this basis $\{e_\mu|_p\}_{1 \le \mu \le n}$ of the tangent space $T_p\mathcal{M}$.*

It is worth noting that $\dim(T_p\mathcal{M}) = \dim(\mathcal{M})$. With this setup, we can now formally introduce the definition of tangent vectors.

**Definition 2** *(Tangent vector): A vector $v|_p \in T_p\mathcal{M}$ is called as a tangent vector if it acts as a derivation, i.e, a linear map acting on smooth functions $f \in C^\infty(\mathcal{M})$ at a point $p \in \mathcal{M}$. Specifically, the map $v|_p : C^\infty(\mathcal{M}, \mathbb{R}) \to \mathbb{R}$ satisfies [3]:*

*i) $v(f + g) = v(f) + v(g) \, \forall \, f, g \in C^\infty(\mathcal{M}, \mathbb{R})$ (linearity)*

*ii) $v(f) = 0$, when $f$ is a constant function, i.e., $v$ acts trivially on constants.*

*iii) $v(fg) = f(p)v(g) + g(p)v(f) \;\; \forall f, g \in C^\infty(\mathcal{M}, \mathbb{R})$ (Leibniz product rule)*

From the definition above, it follows that tangent vectors should be regarded as derivation maps. This is equivalent to the notion of a "directional derivative" (Isham, 1999)

$$(\mathcal{D}_v f)(p) := \frac{d}{dt} f(p + tv|_p)\bigg|_{t=0} = v^\mu(x)\frac{\partial f}{\partial x^\mu}\bigg|_p \,, \quad \forall v = v^\mu(x)\frac{\partial}{\partial x^\mu}\bigg|_p \in T_p\mathcal{M}, \, \forall f \in C^\infty(\mathcal{M}, \mathbb{R}) \,. \tag{13}$$

Directional derivatives are traditionally defined only for scalar-valued functions. This shall be revisited rigorously for a more generalized concept called the "Lie-derivatives", which operates on general tensors, c.f. Section A.2.3.

---

[2]A finite dimensional vector space is isomorphic to its double dual, i.e. $\mathcal{V} \cong (\mathcal{V}^*)^*$.

[3]For sake of ease, we will drop $|_p$ whenever possible.

Thus, $T_p\mathcal{M}$ is the space of directional derivatives. The disjoint union of all the tangent spaces at every point $p \in \mathcal{M}$ forms a structure called *tangent bundles* Isham (1999):

$$T\mathcal{M} = \bigsqcup_{p\in\mathcal{M}} T_p\mathcal{M} = \bigcup_{p\in\mathcal{M}} \left\{ (p, v|_p) : v|_p \in T_p\mathcal{M} \right\} . \tag{14}$$

**Tangent vectors as vector fields.**   In physics, quantities that vary spatiotemporally as a continuum representation are defined as *fields*, featuring in domains such as electrodynamics, gravity, fluid dynamics, or continuum mechanics.

A *vector field* $V$ is a smooth assignment of a tangent vector $v|_p$ to each point $p \in \mathcal{M}$. Thus, a vector field is a map $V : C^\infty(\mathcal{M}) \to C^\infty(\mathcal{M})$, and is defined as:

$$\big(V(f)\big)(p) = v|_p(f) . \tag{15}$$

**Theorem 1** *(Cotangent space): Let $\mathcal{M}$ be a smooth ($C^\infty$)-manifold (differentiable). The cotangent space $T_p^*\mathcal{M} := \{(p, v^*|_p)|\langle v^*|_p, v|_p\rangle = \kappa, \ \forall \, p \in \mathcal{M}, v|_p \in T_p\mathcal{M}, \kappa \in \mathbb{R}\}$ at point $p \in \mathcal{M}$ is the set of all linear maps $v^*|_p : T_p\mathcal{M} \to \mathbb{R}$, i.e., dual to the tangent space. The cotangent space $T_p^*\mathcal{M}$ is spanned by an ordered basis set $\big\{ dx_{|p}^1, dx_{|p}^2, \dots, dx_{|p}^d \big\}$. Thus, any $v^*|_p \in T_p^*\mathcal{M}$ can be expanded as:*

$$v^*|_p = v_\mu^*(x)dx^\mu = v_\mu^*(x)dx^\mu\big|_p . \tag{16}$$

It follows that $dx^\mu\big|_p \left( \dfrac{\partial}{\partial x^\nu}\Big|_p \right) := \left( \dfrac{\partial x^\mu}{\partial x^\nu}\Big|_p \right) = \delta_\nu^\mu$, and $\dim T_p^*\mathcal{M} = \dim T_p\mathcal{M} = \dim \mathcal{M}$. The disjoint union of all the cotangent spaces at every point $p \in \mathcal{M}$ are known as *cotangent bundles* Isham (1999):

$$T^*\mathcal{M} = \bigsqcup_{p\in\mathcal{M}} T_p\mathcal{M} = \bigcup_{p\in\mathcal{M}} \left\{ (p, v^*|_p) : v^*|_p \in T_p^*\mathcal{M} \right\} . \tag{17}$$

One can also construct fields of cotangent vectors (*cotangent fields*) by picking up an element of $T_p^*(\mathcal{M}) \ \forall \, p \in \mathcal{M}$ in a smooth manner. I.e., by assigning one cotangent vector smoothly at each point of the manifold, one obtains a cotangent field (i.e., a smooth section of the cotangent bundle). These cotangent fields are known in mathematical literature as *one-forms*. The set of all smooth one-forms on $\mathcal{M}$ is commonly denoted as $\Omega^1(\mathcal{M})$.

## A.2   TENSORS AND TENSOR FIELDS

**Definition 3** *(Tensors): A rank $(r, s)$ tensor $T$ at a point $p \in \mathcal{M}$ is described as a multilinear map:*

$$T : \underbrace{\mathcal{V}^* \times \dots \times \mathcal{V}^*}_{r-copies} \times \underbrace{\mathcal{V} \times \dots \times \mathcal{V}}_{s-copies} \to \mathbb{R} , \tag{18}$$

where $\times$ denotes the *Cartesian* product and the resultant tensor has a total rank of $r + s$. A tensor takes in $r$ covectors and $s$ vectors, returning a real number, in a multilinear way (linear in each argument separately). The $r$ and $s$ input vectors and covectors pair with the $r$ and $s$ being the convariant and contravariant components, respectively. Equivalently, a tensor is an element that lives in a tensor product of vector and dual spaces, i.e., $T \in (\mathcal{V})^{\otimes r} \otimes (\mathcal{V}^*)^{\otimes s}$. A tensor in a particular basis choice $\{e_{\alpha_n}\}_{1 \leq n \leq r} \in \mathcal{V}$ and $\{\vartheta^{\beta_n}\}_{1 \leq n \leq s} \in \mathcal{V}^*$ is given by

$$T = T^{\alpha_1\alpha_2\dots\alpha_r}{}_{\beta_1\beta_2\dots\beta_s} e_{\alpha_1} \otimes \dots \otimes e_{\alpha_r} \otimes \vartheta^{\beta_1} \otimes \dots \otimes \vartheta^{\beta_s} , \tag{19}$$

where $T^{\alpha_1\alpha_2\dots\alpha_r}{}_{\beta_1\beta_2\dots\beta_s} := T(\vartheta^{\alpha_1}, \dots, \vartheta^{\alpha_r}, e_{\beta_s}, \dots, e_{\beta_s})$ are the coefficients of the tensor w.r.t. the basis set.

A.2.1 TENSOR TRANSFORMATION PROPERTIES

A pivotal criterion for an object to be classified as a tensor(field) is that it transforms according to a well-defined rule under changes of coordinates. Let $\{e_{\alpha_n}\}_{1\leq n\leq r} \in \mathcal{V}$, $\{\vartheta^{\beta_n}\}_{1\leq n\leq s} \in \mathcal{V}^*$, and $\{\bar{e}_{\alpha_n}\}_{1\leq n\leq r} \in \mathcal{V}$, $\{\bar{\vartheta}^{\beta_n}\}_{1\leq n\leq s} \in \mathcal{V}^*$ be two coordinate systems on a smooth manifold $\mathcal{M}$, related by a smooth invertible map. Consider a tensor field of type $(r,s)$ with components $T^{\alpha_1\ldots\alpha_r}{}_{\beta_1\ldots\beta_s}$ in the original coordinate system $\{e_{\alpha_n}\}_{1\leq n\leq r} \in \mathcal{V}$, $\{\vartheta^{\beta_n}\}_{1\leq n\leq s} \in \mathcal{V}^*$. Under a change of coordinate systems, the components in the new coordinate system $\{\bar{e}_{\alpha_n}\}_{1\leq n\leq r} \in \mathcal{V}$, $\{\bar{\vartheta}^{\beta_n}\}_{1\leq n\leq s} \in \mathcal{V}^*$ transform according to the following tensor transformation law:

$$\bar{T}^{\mu_1\ldots\mu_r}{}_{\nu_1\ldots\nu_s}(\bar{x}) = \mathcal{J}^{\mu_1}_{\alpha_1}\cdots\mathcal{J}^{\mu_r}_{\alpha_r}\ T^{\alpha_1\ldots\alpha_r}{}_{\beta_1\ldots\beta_s}(x)\ \left(\mathcal{J}^{-1}\right)^{\beta_1}_{\nu_1}\cdots\left(\mathcal{J}^{-1}\right)^{\beta_s}_{\nu_s}, \tag{20}$$

where $\mathcal{J}^{\mu_k}_{\alpha_k} \equiv \frac{\partial \bar{x}^{\mu_k}}{\partial x^{\alpha_k}}$ and $\left(\mathcal{J}^{-1}\right)^{\beta_l}_{\nu_l} \equiv \frac{\partial x^{\beta_l}}{\partial \bar{x}^{\nu_l}}$ are the Jacobian and Jacobian inverse matrices in the coordinate basis, respectively. $\mathcal{J}^{\mu_k}_{\alpha_k}$ is the contravariant transformation of the contravariant components of $T^{\alpha_1\ldots\alpha_r}{}_{\beta_1\ldots\beta_s}$, whereas $\left(\mathcal{J}^{-1}\right)^{\beta_l}_{\nu_l}$ is the covariant transformation of the covariant components of $T^{\alpha_1\ldots\alpha_r}{}_{\beta_1\ldots\beta_s}$. The indices $\mu_k, \nu_l$ label components in the new coordinates and $\alpha_k, \beta_l$ are dummy indices summed over the old coordinates. A key feature of a tensor is that, if it is zero in one coordinate system, it is zero in every other coordinate system. This transformation law ensures that the tensorial nature of the object is preserved independent of the coordinate chart chosen.

**Tensor fields.** A tensor field is a collection of tensor-valued rank quantities $(r,s)$ such that at each point $p \in \mathcal{M}$, the multilinear function associates a value $T_p \in \mathcal{V}_p^{\otimes r} \otimes \left(\mathcal{V}_p^*\right)^{\otimes s}$. Thus, the components $T^{\alpha_1\ldots\alpha_r}{}_{\beta_1\ldots\beta_s}(p)$ are functions of the points of the manifold.

By definition, some known examples of tensor fields in physics and machine learning are:

- Rank 0 tensor, e.g., temperature field $\varphi : \mathbb{R}^m \to \mathbb{R}$ (scalar field)
- Rank $(1,0)$ tensor, e.g., (velocity, momentum, displacement) vector fields $v: \mathbb{R}^m \to \mathbb{R}^n$ (contravariant vector field). These rank $(1,0)$ tensors have one component that transforms contravariantly, and "eats up" a covariant component, e.g., $v^T$ to produce a scalar.
- Rank $(0,1)$ tensor, e.g., gradient vector fields $\nabla : \mathbb{R} \to \mathbb{R}^m$ (covariant vector field). These rank $(0,1)$ tensors have one component that transforms covariantly, and "eat up" a contravariant component to produce a scalar.
- Rank $(0,2)$ tensor, e.g., a matrix representing a bilinear form that takes in two vectors and outputs a scalar. We will see the metric tensor $g_{\mu\nu}$ as an example. In continuum and structural mechanics, a known example is the *strain* tensor $\epsilon_{ij}$ representing the deformation of a crystal (body) caused by external forces such as stress.
- Rank $(2,0)$ tensor, e.g., a matrix as a multilinear map that takes in two covectors and outputs a scalar. An example for a rank $(2,0)$ tensor is the outer product of two vectors. An example is the *Cauchy stress tensor* $\boldsymbol{\sigma}^{ij}$ from structural mechanics, which represents the internal forces per unit area acting inside a material body. The stress tensor takes in two vectors, i.e., the normal vector to the surface (describing orientation), and the direction vector along which the force acts (projection), and returns a scalar (force per unit area in that direction).

A.2.2 OPERATIONS ON TENSOR FIELDS

For multiple tensors of the same type $(r,s)$, the algebraic operations such as addition, subtraction or multiplication by functions are straightforward. Here, we address multiplication of tensors of different ranks.

Let $T$ be a rank $(r,s)$ tensor and $S$ a rank $(p,q)$ tensor. One can construct a *tensor product* $T \otimes S$ resulting in a new tensor of rank $(r+p, s+q)$, defined by

$$T \otimes S(e_{\alpha_1},\ldots,e_{\alpha_r}, e_{\eta_1},\ldots,e_{\eta_p}, \varphi^{\beta_1},\ldots,\varphi^{\beta_s}, \varphi^{\delta_1},\ldots,\varphi^{\delta_q}) \tag{21}$$

$$= T(e_{\alpha_1},\ldots,e_{\alpha_r}, e_{\eta_1},\ldots,e_{\eta_p})S(\varphi^{\beta_1},\ldots,\varphi^{\beta_s}, \varphi^{\delta_1},\ldots,\varphi^{\delta_q}), \tag{22}$$

and the components of this composite tensor read,

$$(T \otimes S)^{\alpha_1 \ldots \alpha_r \eta_1 \ldots \eta_p}_{\qquad \beta_1 \ldots \beta_s \delta_1 \ldots \delta_q} := T^{\alpha_1 \ldots \alpha_r}_{\qquad \beta_1 \ldots \beta_s} S^{\eta_1 \ldots \eta_p}_{\qquad \delta_1 \ldots \delta_q} . \tag{23}$$

Another useful rule is that of contracting over repeated index/indices each from the vector and dual space respectively. Consider a rank $(r, s)$ tensor

$$T^{\alpha_1 \ldots \alpha_p \ldots \alpha_r}_{\quad \beta_1 \ldots \alpha_p \ldots \beta_s} = T^{\alpha_1 \ldots \alpha_r}_{\quad \beta_1 \ldots \beta_s} . \tag{24}$$

I.e., $\alpha_p$ is summed-over in the contravariant and covariant indices, and, thus it gets contracted. The resulting tensor is of rank $(r - 1, s - 1)$ .

### A.2.3 LIE DERIVATIVE: GENERALIZING THE NOTION OF DIRECTIONAL DERIVATIVES FOR TENSOR FIELDS

Directional derivatives are of great importance and often appear in domains such as fluid dynamics, where a scalar field is differentiated with respect to a vector flow field, capturing infinitesimal dragging of scalar fields along flows generated by a vector field. Flows can be viewed as "diffeomorphisms" Poisson (2004) induced by these vector fields.

However, generalizing the notion of directional derivatives require defining derivatives of a set of tensor fields of arbitrary rank $(r, s)$ w.r.t. a set of vector fields. This is often not possible on arbitrary manifolds, and requires a concept of differentiating in a tensorial setting. Geometrically, to compare tensors at infinitesimally separated points on a manifold $\mathcal{V}$, say at points $p, q \in \mathcal{M}$ requires to "drag" the tensor from $p$ to $q$ (also called parallel transporting, c.f. Section A.3.3.1).

Alternatively, a simpler approach to describe the dragging is via coordinate transformation from $p$ to $q$. This is the idea behind the *Lie derivative*. The Lie derivative along a vector field $v|_p \in T_p\mathcal{M}$ measures by how much the changes in a tensor along $v$ differ from a mere infinitesimal passive coordinate transformation of the tensor generated by $v$. In other words, the Lie derivative compares the actual rate of change of the tensor as you move along $v$ against the change you'd get if everything were just shifted passively via a coordinate transformation. We provide a rough sketch of the derivation, but detailed explanations can be found here (Lee, 2012; Poisson, 2004).

Consider an infinitesimal coordinate transformation which maps the vector with coordinates $x^\mu|_p$ at point $p$ to $\bar{x}^\mu|_q$ at point $q$:

$$\bar{x}^\mu|_q = x^\mu|_p + \delta\xi \, v^\mu(x)|_p . \tag{25}$$

It is to explicitly note that the original coordinates $x^\mu|_p$ and the transformed coordinates $\bar{x}^\mu|_q$ are components of the same set of basis vectors. Such transformations fall under the category of *active coordinate transformations* that map points (or tensors at those points) at old locations to new locations in the old coordinate system – in this case by "moving" a small amount $\delta\xi$ along the vector field $v|_p \in T_p\mathcal{M}$. In other words, an active coordinate transformation maps points (and tensors) to new locations in the old coordinate system keeping the basis set intact. Whereas, passive transformations assign new coordinates to the old points (and tensors) by transforming the basis set itself.

Assuming a coordinate basis, one can differentiate the transformation w.r.t. the original coordinates, which yields

$$\frac{\partial \bar{x}^\mu}{\partial x^\nu} = \delta^\mu_\nu + \delta\xi \frac{\partial v^\mu(x)}{\partial x^\nu} . \tag{26}$$

The result contains the identity matrix $\delta^\mu_\nu$ and a small correction due to the flow field $v^\mu(x)$. To the first order, the inverse of the above Jacobian is $\dfrac{\partial x^\nu}{\partial \bar{x}^\mu} = \delta^\mu_\nu - \delta\xi \dfrac{\partial v^\mu(x)}{\partial x^\nu}$.

The Lie-derivative of a tensor field $T^\mu_\nu$ with respect to $v^\mu$ follows a similar pattern and is defined via the limes:

$$\mathcal{L}_v T^\mu_\nu = \lim_{\delta\xi \to 0} \frac{T^\mu_\nu(\bar{x}) - \bar{T}^\mu_\nu(\bar{x})}{\delta\xi} . \tag{27}$$

In this scheme, it is important to distinguish three distinct tensor field evaluations: a) $T_\nu^\mu(x)$ (original tensor in untransformed coordinates), b) $\bar{T}_\nu^\mu(\bar{x})$ (transformed tensor in the transformed coordinates) and c) $T_\nu^\mu(\bar{x})$ (original tensor in transformed coordinates).

In order to compute Eq. (27), we need two important concepts from differential geometry called *pushforward* and *pull-back* operations. We direct the interested readers to more advanced literature Isham (1999); Lee (2012); Kobayashi & Nomizu (1963)

These three separate tensors fields can be related in the following manner: Firstly, the tensor mapped to the new set of coordinates $\bar{T}_\nu^\mu(\bar{x})$ can be obtained via Eq. (20),

$$\bar{T}_\nu^\mu(\bar{x}) = (\mathcal{J}^{-1})_\rho^\mu \, \mathcal{J}_\nu^\sigma \, T_\sigma^\rho(x) \equiv T_\nu^\mu(x) + \delta\xi \left( \frac{\partial v^\mu}{\partial x^\sigma} T_\nu^\sigma(x) - \frac{\partial v^\sigma}{\partial x^\nu} T_\sigma^\mu(x) \right) + \mathcal{O}(\delta\xi^2) \,. \quad (28)$$

Secondly, the original tensor in transformed coordinates $T_\nu^\mu(\bar{x})$ can be evaluated at $q$, by a Taylor expansion:

$$T_\nu^\mu(\bar{x}) = T_\nu^\mu(\bar{x}^\sigma) = T_\nu^\mu(x^\sigma + \delta\xi \, v^\sigma) = T_\nu^\mu(x) + \delta\xi \, v^\sigma \frac{\partial T_\nu^\mu}{\partial x^\sigma} + \mathcal{O}(\delta\xi^2) \,. \quad (29)$$

Substituting Eqs. (28, 29) into the Lie-derivative definition of Eq. (27), and $\delta\xi \to 0$ one finds the following final expression:

$$\mathcal{L}_v T_\nu^\mu = v^\sigma \frac{\partial T_\nu^\mu}{\partial x^\sigma} - \underbrace{\frac{\partial v^\mu}{\partial x^\sigma} T_\nu^\sigma(x)}_{\text{pullback}} + \underbrace{\frac{\partial v^\sigma}{\partial x^\nu} T_\sigma^\mu(x)}_{\text{pushforward}} \,. \quad (30)$$

The pushforward and pullback operations drag the transformed tensor field onto the original point, where differences can be computed.

Thus, tensors are being compared in the same tangent/cotangent space. Mathematically, for smooth maps[4] (diffeomorphisms) $\Phi : \mathcal{M} \to \mathcal{N}$ the pushforward $\Phi_* : T_p\mathcal{M} \to T_{\Phi(p)}\mathcal{N}$ pushes vector fields forward from one tangent space of a domain $T_p\mathcal{M}$ to the tangent space of another tangent space $T_{\Phi(p)}\mathcal{N}$. The pullback, a dual linear map to pushforward, drags covectors (one-forms) living in cotangent spaces $(\Phi_*)^* \equiv \Phi^* : T_{\Phi(p)}^*\mathcal{N} \to T_p^*\mathcal{M}$ in the reverse direction to the domain. Hence, the contributions from the pushforward on the vector field components and pullback on the covector field components jointly determine the structure of the Lie derivative of a mixed tensor field, as expressed in Eq. (30). These operations offer a coherent mathematical framework for transitioning between tangent and cotangent bundles mapped onto other tangent and cotangent bundles via smooth maps, acting appropriately on vector fields and one-forms, respectively.

For any arbitrary rank $(r, s)$ tensor, Eq. (30) can be generalized to:

$$(\mathcal{L}_v T)_{\nu_1 \dots \nu_s}^{\mu_1 \dots \mu_r} = v^\sigma \frac{\partial}{\partial x^\sigma} T_{\nu_1 \dots \nu_s}^{\mu_1 \dots \mu_r} - \sum_{i=1}^r T_{\nu_1 \dots \nu_s}^{\mu_1 \dots \sigma \dots \mu_r} \frac{\partial v^{\mu_i}}{\partial x^\sigma} + \sum_{j=1}^s T_{\nu_1 \dots \sigma \dots \nu_s}^{\mu_1 \dots \mu_r} \frac{\partial v^\sigma}{\partial x^{\nu_j}} \,. \quad (31)$$

Lie-derivatives do not require the notion of a connection. Connections will be introduced in detail in Section A.3.3 and intuitively stating, connects two distinct Tangent spaces at different points, which is not to be confused with a pullback operation. Here, is an instructive comparison table for that compares different differentiation schemes:

Table 6: Comparison between actions of directional, covariant, and Lie derivatives.

| Feature | Directional derivative | Covariant derivative | Lie derivative |
|---|---|---|---|
| Input function | Scalar fields | Tensor fields | Tensor fields |
| Connection dependence | ✗ | ✓(Explicit) | ✗ |
| Captures curvature | ✗ | ✓ | ✗ |
| Measures | Scalar changes | Intrinsic curvature | Diffeomorphisms (flows) |

[4]for e.g., dragging of coordinates as in Eq. (25) due to flows induced by vector fields.

## A.3 RIEMANNIAN AND LORENTZIAN GEOMETRY

### A.3.1 FOUR DIMENSIONAL SPACETIME AS A CONTINUOUS DIFFERENTIABLE MANIFOLD

The fabric of spacetime according to general relativity is a combination of three-dimensional space and a strictly positively progressing time direction into a single four-dimensional continuum. Thus, space and time mix between each other through special orthogonal transformations $SO(1,3)$ called the *Lorentz transformations*. In order to rigorously define the four-dimensional spacetime, it is necessary to define the following:

**Definition 4** (**Manifold**): *A $n$-dimensional manifold $\mathcal{M}$ is a space, that, locally resembles the n-dimensional Euclidean space $\mathbb{R}^n$. However, combining these local patches together, globally, the space deviates from $\mathbb{R}^n$.*

**Definition 5** (**Hausdorff space**): *Let $\mathcal{K}$ be a topological space. Then $\mathcal{K}$ is said to be a Hausdorff space if: For every pair of distinct points $x, y \in \mathcal{K}$ with $x \neq y$, there exist open sets $U, V \subset \mathcal{K}$ such that:*

$$x \in U, \quad y \in V, \quad and \quad U \cap V = \emptyset \,. \tag{32}$$

**Definition 6** (**Differentiable manifold**): *An $n$-dimensional differentiable manifold is a Hausdorff topological space $\mathcal{K}$ such that:*

*i) Locally $\mathcal{K}$ is homeomorphic to $\mathbb{R}^n$. Thus, $\forall \, p \in \mathcal{K}$ there is an open set $\mathcal{U}$ such that $p \in \mathcal{U}$ and a homeomorphism $\phi : \mathcal{U} \to \mathcal{Z}$ with $\mathcal{Z}$ an open subset of $\mathbb{R}^n$.*

*ii) For two subsets $\mathcal{U}_\alpha$ and $\mathcal{U}_\beta$ with $\mathcal{U}_\alpha \bigcap \mathcal{U}_\beta \neq \emptyset$, the homeomorphisms (topologically isomorphic) $\phi_\alpha : \mathcal{U}_\alpha \to \mathcal{Z}_\alpha$ and $\phi_\beta : \mathcal{U}_\beta \to \mathcal{Z}_\beta$ are compatible, i.e., the map $\phi_\beta \circ \phi_\alpha^{-1} : \phi_\alpha\big(\mathcal{U}_\alpha \bigcap \mathcal{U}_\beta\big) \to \phi_\beta\big(\mathcal{U}_\alpha \bigcap \mathcal{U}_\beta\big)$ is smooth (infinitely differentiable $\mathcal{C}^\infty$), and so is its inverse map.*

The $\phi_\alpha$ are often called charts and a collection (union) of them $\bigcup_\alpha \phi_\alpha$ is called an *atlas*. These charts provides a coordinate system, labeling $\mathcal{U}_\alpha \subset \mathcal{K}$. The coordinate associated to $p \in \mathcal{U}_\alpha$ is:

$$\phi_\alpha(p) = \big(x^1(p), x^2(p), ...., x^n(p)\big)$$

Mathematically, the spacetime continuum denoted as $\mathcal{M}$, is a *differentiable manifold* with the structure of an *Hausdorff* topological space.

To summarize, a differentiable manifold is a space that may be curved or complicated globally, but looks like Euclidean space up close, and allows for smooth calculus to be done on it. The Hausdorff space ensures than one can separate points nicely with open sets. This avoids weird pathological cases and makes limits and continuity well-behaved. Locally Euclidean means that one can do calculus as if we were on flat space – even if the whole space is curved. And finally, the compatibility between overlapping charts ensures that one can do calculus consistently across different charts.

### A.3.2 METRIC TENSOR

**Definition 7** (**Metric**): *A metric $g$ is a rank $(0,2)$ tensor field that is defined as a symmetric bilinear map that assigns to each $p \in \mathcal{M}$ a positive-definite inner product $g : T_p\mathcal{M} \times T_p\mathcal{M} \to \mathbb{R}$ such that*

*i) $g(v|_p, w|_p) = g(v, w) = g(w, v) \,\forall v, w \in T_p\mathcal{M}$ (symmetric)*

*ii) For any $p \in \mathcal{M}$, $g(v, w) = 0 \,\forall w|_p \in T_p\mathcal{M}$ implying $v|_p = 0$ (non-degenerate).*

Represented in the basis set of the tangent space, the metric components at each point $p$ is given by

$$g_{\mu\nu} = g_{\nu\mu} := g_p\left(\left.\frac{\partial}{\partial x^\mu}\right|_p, \left.\frac{\partial}{\partial x^\nu}\right|_p\right) \tag{33}$$

and the metric can be expanded as,

$$g = g_{\mu\nu}(x) \, dx^\mu \otimes dx^\nu \,. \tag{34}$$

Geometrically, the metric defined in Eq. (34) generalizes the notion of distances and induces a norm $||.||_p : T_p\mathcal{M} \to \mathbb{R}$ for generic coordinates such as curvilinear and/or manifolds possessing geometries that are intrinsically non-Euclidean in nature, for e.g., spaces of constant positive sectional curvature $K = 1$ (e.g., a 2-sphere $\mathbb{S}^2$ embedded in $\mathbb{R}^3$), spaces of constant sectional curvature $K = -1$ such as hyperbolic geometry (Bolyai-Lobachevsky spaces $\mathbb{H}^2$).

The distance between two points in such cases is called the *line element*, which is defined as,

$$ds^2 = g_{\mu\nu}(x)dx^\mu dx^\nu. \tag{35}$$

For an $n$-dimensional manifold $\mathcal{M}$, the metric tensor $g_{\mu\nu}$ is a $n \times n$ symmetric matrix, $g(\phi_\mu, \phi_\nu) := \langle \phi_\mu, \phi_\nu \rangle$, with $\frac{n(n+1)}{2}$ independent components (not necessarily expanded in the coordinate basis):

$$g_{\mu\nu} = \begin{pmatrix} \langle \phi_0, \phi_0 \rangle & \langle \phi_0, \phi_1 \rangle & \cdots & \langle \phi_0, \phi_{n-1} \rangle \\ \langle \phi_1, \phi_0 \rangle & \langle \phi_1, \phi_1 \rangle & \cdots & \langle \phi_1, \phi_{n-1} \rangle \\ \vdots & \vdots & \ddots & \vdots \\ \langle \phi_{n-1}, \phi_0 \rangle & \langle \phi_{n-1}, \phi_1 \rangle & \cdots & \langle \phi_{n-1}, \phi_{n-1} \rangle \end{pmatrix}. \tag{36}$$

**Definition 8** (*Metric bundle*): *Let $\mathcal{M}$ be a smooth manifold and $(x_0, \cdots, x_n)$ be local coordinates on $\mathcal{U} \subset \mathcal{M}$. The bundle of symmetric $(0,2)$-tensors on $\mathcal{M}$ is the subbundle $\mathrm{Sym}^2(T^*\mathcal{M}) \subset T^{0,2}\mathcal{M} = T^*\mathcal{M} \times T^*\mathcal{M}$.*

In fact, sections of $\mathrm{Sym}^2(T^*\mathcal{M})$ contains all the symmetric bilinear forms, i.,e. symmetric $(0,2)$-tensor fields, and includes the pseudo-Riemannian metrics on $\mathcal{M}$.

**Riemannian manifolds.** A metric $g$ where all diagonal entries of the metric are positive, i.e., $g_{\mu\mu} > 0, \mu = 0, \ldots, \dim(\mathcal{M}) - 1$ is called a *Riemannian* metric. Thus, a manifold $\mathcal{M}$ endowed with a Riemannian metric $g$ is known as a *Riemannian manifold* denoted as a tuple $(\mathcal{M}, g)$ (Jost, 2008).

For the the Euclidean space $\mathbb{R}^n$ with Cartesian coordinates representation

$$g = dx^1 \otimes dx^1 + \ldots + dx^n \otimes dx^n \tag{37}$$

the metric tensor amounts to $g_{ij} = \delta_{ij}$. and boils down to Pythagoras' theorem. A general Riemannian metric prescribes a method to measure the norm of a vector $v$ as $\sqrt{g(v,v)} = ||v||$ and also allows for measuring angles between any two vectors $v, w$ at each point $\cos\vartheta = \frac{g(v,w)}{\sqrt{g(v,v)g(w,w)}}$. Like any other tensor, the components of the metric tensor transform under a coordinate change according to Eq. (20):

$$\bar{g}_{\alpha\beta}(\bar{x}) = \left[ (\mathcal{J}^{-1})^\mu_\alpha \right]^{\mathrm{T}} g_{\mu\nu}(x) \, (\mathcal{J}^{-1})^\nu_\beta . \tag{38}$$

**Definition 9** (*Arc length*): *Let $\gamma : [0,1] \to \mathcal{M}$ be a piecewise smooth curve on a differentiable manifold $\mathcal{M}$, with $\gamma(0) = p$ and $\gamma(1) = q$. The velocity vector along the curve is denoted by $\dot{\gamma}(t)$, which lives in the tangent space $T_{\gamma(t)}\mathcal{M}$. If the curve is expressed in local coordinates $x^\mu(t)$, then the components of the tangent vector $\dot{\gamma}(t)$ are given by $\frac{dx^\mu(t)}{dt}$. The arc length $\mathcal{L}(\gamma)$ (distance) of the curve is then defined by*

$$\mathcal{L}(\gamma) = \int_0^1 \|\dot{\gamma}(t)\|_{\gamma(t)} \, dt = \int_0^1 \sqrt{g_{\mu\nu}(x(t)) \frac{dx^\mu(t)}{dt} \frac{dx^\nu(t)}{dt}} \, dt . \tag{39}$$

This arc length[5] is reparameterization invariant, i.e., it does not depend on the choice of parameterization of the curve $\gamma(t)$. It is a very important result that every smooth manifold admits a Riemannian metric.

---

[5]It is also called an action in physics.

**Lorentzian manifolds.**  Unlike Riemannian manifolds, spacetime is actually a *pseudo-Riemannian* manifold[6], that is, the metric is not positive definite. Thus, the underlying metric carries a signature $(-, +, +, +)$, meaning, $g_{tt} < 0$. Consequently, spacetime is a Lorentzian manifold $\mathscr{M}$, and, forms the basis for electromagnetism and special relativity. The simplest example of a Lorentzian manifold of arbitrary dimension is the *Minkowksi* metric, which is flat (meaning no curvature):

$$\eta = -dx^0 \otimes dx^0 + dx^1 \otimes dx^1 + .... + dx^{n-1} \otimes dx^{n-1} \ , \tag{40}$$

where, the components of the Minkowksi metric are $\eta_{\mu\nu} = \text{diag}(-1, +1, ..., +1)$. It is possible to find an orthonormal basis $\{e_\mu\}$ of $T_p\mathcal{M}$ around a small neighborhood of point $p$ of a Lorentzian manifold such that, "locally", the metric resembles the Minkowski metric

$$g_{\mu\nu}|_p = \eta_{\mu\nu} \ . \tag{41}$$

In the case of Lorentzian manifolds $\mathscr{M}$, the arc length $\mathcal{L}(\gamma)$ in Eq. (39) is modified due to non positive-definiteness

$$\tau(\sigma) = \int_0^\sigma \sqrt{-g_{\mu\nu}(x) \frac{dx^\mu(\sigma')}{d\sigma'} \frac{dx^\nu(\sigma')}{d\sigma'}} \, d\sigma', \tag{42}$$

and is sometimes $\tau$ is referred to as *proper-time*. The minus sign under the square root ensures the integrand is positive for timelike paths, since for timelike intervals, the inner product of the velocity vector with itself (under the Lorentzian metric) is negative, i.e.,

$$ds^2 = g_{\mu\nu}dx^\mu dx^\nu. \tag{43}$$

**Natural isomorphism between vector spaces and dual spaces.**  The metric provides a natural *isomorphism* between vector spaces and dual spaces and allows the switch between contravariant and covariant components[7]. This is done via the following mapping $g : T_p\mathcal{M} \to T_p^*\mathcal{M}$, where at each point $p$, a one-form (covectors) is obtained via contraction operation of a vector field $v|_p$ with the metric $g$.

For a vector $v|_p = v^\mu \frac{\partial}{\partial x^\mu}$ and the covector $v^*|_p = v_\mu dx^\mu$, the components are related by

$$v_\mu = g_{\mu\nu}v^\nu \ .$$

Since $g$ is non-degenerate, it is invertible. We denote the inverse metric as $g^{\mu\nu}$, such that $g^{\mu\sigma}g_{\sigma\nu} = \delta_\nu^\mu$. This is a rank $(2, 0)$ tensor of the form $\hat{g} = g^{\mu\nu} \frac{\partial}{\partial x^\mu} \otimes \frac{\partial}{\partial x^\nu}$. Through the inverse metric indices can be raised, e.g. $x^\mu = g^{\mu\nu}x_\nu$.

Such index contraction rules with the metric apply to tensors of rank $(r, s)$ or even quantities that are not tensors:

$$S^{\beta_1....\beta_s}_{\phantom{\beta_1....\beta_s}\alpha_1....\alpha_r} = \left( \prod_{i=1}^s g^{\beta_i\gamma_i} \right) \left( \prod_{i=1}^r g_{\alpha_i\delta_i} \right) S^{\delta_1....\delta_r}_{\phantom{\delta_1....\delta_r}\gamma_1....\gamma_s} \ . \tag{44}$$

### A.3.3  CONNECTIONS & COVARIANT DERIVATIVE

Transporting vector and tensor fields systematically on manifolds requires mapping vector spaces at one point to vector spaces at another. While this can be done trivially in the Euclidean setting, for Riemannian and Lorentzian manifolds this is a non-trivial since these vector fields and tensor fields live in different vector spaces. This necessitates a geometric object that behaves as a "connector" between vector spaces. This is achieved via a geometric entity called the *affine-connection*, which is a vector-valued one-form.

**Definition 10** *(Affine connection): Let $\mathcal{M}$ be a smooth manifold and $\Gamma(T\mathcal{M})$ be the space of vector fields on $\mathcal{M}$, that is the space of smooth sections of the tangent bundle (i.e., the collection of all tangent spaces). An* affine connection *is a bilinear map*

$$\nabla : \Gamma(T\mathcal{M}) \times \Gamma(T\mathcal{M}) \to \Gamma(T\mathcal{M})$$
$$(v, w) \mapsto \nabla_v w \ .$$

---

[6]In general, a pseudo-Riemannian manifold has a signature $(\underbrace{-, ..., -}_{m}, \underbrace{+, ..., +}_{n})$.

[7]In the context of numerical relativity the switch between contravariant and covariant components is called "raising and lowering indices".

*The differential operator $\nabla_v$ is the* covariant derivative *satisfying the following for tangent vectors $v, w$ (short for $v|_p, w|_p$):*

*i)* $\nabla_v(w + z) = \nabla_v w + \nabla_v z$

*ii)* $\nabla_{fv} w = f \nabla_v w \;\; \forall f \in C^\infty(\mathcal{M}, \mathbb{R})$

*iii)* $\nabla_v(fw) = (\mathcal{D}_v f)w + f \nabla_v w \;\; \forall f \in C^\infty(\mathcal{M}, \mathbb{R}), \;\; \mathcal{D}_v f = v(f)$ *is the directional derivative.*

The affine connection is completely independent of the metric. However, if a manifold is endowed with a metric, this enables expressing the connection in terms of the metric. In GR, one looks at a special subclass of affine connections called *Levi-Civita connection*, due to the symmetry property of the metric tensor.

**Definition 11** *(**Levi-Civita connection**): An affine connection is an* Levi-Civita *connection for tangent vectors $v, w$ (short for $v|_p, w|_p$) if:*

$$\nabla_v g = 0 \;\; \forall v \in \Gamma(T\mathcal{M}) \;\; \text{(metricity condition)} \tag{45a}$$

$$\nabla_v w - \nabla_w v = [v, w] \;\; \forall v, w \in \Gamma(T\mathcal{M}) \;\; \text{(torsion-free condition)}, \tag{45b}$$

*where,* $[v, w] = \left(v^\mu \partial_\mu w^\nu - w^\mu \partial_\mu v^\nu\right)\partial_\nu$ *is the* Lie-bracket *of vector fields* Kobayashi & Nomizu *(1963).*

**Intuition behind the torsion-free condition.** Imagine you are moving on a smooth surface (like walking on a hill), and you have two "directions" $v|_p$ and $w|_p$ at a point p. Now: First move along $v$ a tiny bit, then subsequently along $w$. Alternatively, move along $w$ first, then along $v$, akin to constructing a parallelogram. In flat, Euclidean space, doing these two moves would land you at the same final point, because partial derivatives commute. On a curved surface (a manifold $\mathcal{M}$), they don't generally commute – you end up slightly shifted. The Lie bracket $[v, w]$ measures how far off (deficit) you are after moving in $v$ and then $w$, compared to $w$ and then $v$. It captures the "non-commutativity" of the vector transport along the two distinct directions, which leads to a non-closure of the parallelogram. Thus, the Lie bracket is intrinsic to the manifold, and shows how the transport of $v$ and $w$ interact. In torsion-free connections, the "commutation failure" is purely due to the manifold's structure – not any extra "twisting" introduced by the connection itself.

### A.3.3.1 Parallel transport

**Definition 12** *(**Parallel transport**): Let $\gamma : [0, 1] \to \mathcal{M}$ be a smooth curve on the manifold, and let $T$ be a smooth $(r, s)$-rank tensor field defined along the curve $\gamma$. The* parallel transport *of $T$ along the curve $\gamma(\tau)$ is defined by the condition that its directional covariant derivative along the curve's tangent vector vanishes:*

$$\nabla_{\dot{\gamma}(\tau)} T = 0, \;\; \forall \tau \in [0, 1] \,, \;\; \dot{\gamma}(t) \in T_p\mathcal{M} \,. \tag{46}$$

In local coordinates $\{x^\mu(\tau)\}$, the parallel transport condition for the components of the tensor field $T^{\mu_1 \dots \mu_r}_{\nu_1 \dots \nu_s}(\tau)$ along the curve $\gamma(t)$ is:

$$\frac{d}{d\tau} T^{\mu_1 \dots \mu_r}_{\nu_1 \dots \nu_s}(\tau) + \sum_{i=1}^{r} \Gamma^{\mu_i}_{\lambda \rho} \dot{x}^\rho(\tau) T^{\mu_1 \dots \lambda \dots \mu_r}_{\nu_1 \dots \nu_s}(\tau) - \sum_{j=1}^{s} \Gamma^{\lambda}_{\nu_j \rho} \dot{x}^\rho(\tau) T^{\mu_1 \dots \mu_r}_{\nu_1 \dots \lambda \dots \nu_s}(\tau) = 0 \,. \tag{47}$$

These equations are a set of coupled ODEs, and can be solved uniquely for an initial condition to find a unique vector at each point along the curve $\gamma(\tau)$. This ensures that as the tensor is transported along the curve, its components change in such a way that their covariant rate of change along the curve vanishes.

### A.3.3.2 Christoffel symbols

The Levi-Civita covariant derivative contains, apart from the partial derivative term, a correction field that calibrates the deficit between vector (tensor) fields transported along a path on the manifold. For a basis $\{e_\mu\}$ that is transported the covariant derivative is given by,

$$\nabla_{e_\nu} e_\mu(x) = \Gamma^\sigma_{\mu\nu}(x) e_\sigma(x) \,. \tag{48}$$

The *covariant derivative*, denoted by $\nabla_{e_\nu} \equiv \nabla_\nu = \partial_\nu + \Gamma_\nu$, defines a "modified" differentiation operator that preserves tensorial character under general coordinate transformations. The quantities $\Gamma^\sigma_{\mu\nu}$, known as the *Christoffel symbols*, represent the components of the *Levi-Civita connection*, which is uniquely determined by the requirement that the connection is torsion-free and compatible with the metric. Notably, these symbols are symmetric in their lower two indices, i.e., $\Gamma^\sigma_{\mu\nu} = \Gamma^\sigma_{\nu\mu}$. The action of the covariant derivative on a general tensor field of type $(r, s)$ ensures that derivatives of tensors transform covariantly, thereby extending the notion of differentiation from vector calculus to curved manifolds.

$$\nabla_\mu T^{\alpha_1 \ldots \alpha_r}_{\phantom{\alpha_1 \ldots \alpha_r}\beta_1 \ldots \beta_s}(x) = \frac{\partial}{\partial x^\mu} T^{\alpha_1 \ldots \alpha_r}_{\phantom{\alpha_1 \ldots \alpha_r}\beta_1 \ldots \beta_s}(x) + \sum_{i=1}^{r} \Gamma^{\alpha_i}_{\mu\sigma}(x) T^{\alpha_1 \ldots \sigma \ldots \alpha_r}_{\phantom{\alpha_1 \ldots \sigma \ldots \alpha_r}\beta_1 \ldots \beta_s}(x) \tag{49}$$

$$- \sum_{j=1}^{s} \Gamma^\sigma_{\mu\beta_j}(x) T^{\alpha_1 \ldots \alpha_r}_{\phantom{\alpha_1 \ldots \alpha_r}\beta_1 \ldots \sigma \ldots \beta_s}(x) \, . \tag{50}$$

The action of the covariant derivative on a scalar field, simply reduces to a partial derivative

$$\nabla_\mu \phi(x) = \frac{\partial \phi(x)}{\partial x^\mu} \, .$$

Christoffel symbols can be solely expressed in terms of the metric and its partial derivatives:

$$\Gamma^\rho_{\mu\nu}(x) := \frac{1}{2} g^{\rho\sigma} \left( \partial_\mu g_{\sigma\nu}(x) + \partial_\nu g_{\sigma\mu}(x) - \partial_\sigma g_{\mu\nu}(x) \right) = \Gamma^\rho_{\nu\mu}(x) \, . \tag{51}$$

A crucial feature of any connection is that it is **not** a tensorial quantity. Connections don't obey the transformation law in Eq. (20) under coordinate changes. This can be easily seen through the components of the Christoffel symbols in the coordinate basis:

$$\bar{\Gamma}^\rho_{\mu\nu}(\bar{x}) = \underbrace{\frac{\partial \bar{x}^\rho}{\partial x^\gamma} \frac{\partial x^\alpha}{\partial \bar{x}^\mu} \frac{\partial x^\beta}{\partial \bar{x}^\nu} \Gamma^\gamma_{\alpha\beta}(x)}_{\text{tensorial contribution}} + \underbrace{\frac{\partial^2 x^\sigma}{\partial \bar{x}^\mu \partial \bar{x}^\nu} \frac{\partial \bar{x}^\rho}{\partial x^\sigma}}_{\text{non-tensorial contribution}} \, . \tag{52}$$

Christoffel symbols play a significant role in defining most stationary trajectories (shortest or longest) in the non-Euclidean setting.

**Lie derivatives revisited: Levi-Civita connection included.** In case of a nonzero Levi-civita connection, the partial derivatives of an ordinary Lie derivative in Eq. (31) is replaced by the covariant derivatives:

$$(\mathcal{L}_v T)^{\mu_1 \ldots \mu_r}_{\phantom{\mu_1 \ldots \mu_r}\nu_1 \ldots \nu_s} = v^\sigma \nabla_\sigma T^{\mu_1 \ldots \mu_r}_{\phantom{\mu_1 \ldots \mu_r}\nu_1 \ldots \nu_s} - \sum_{i=1}^{r} T^{\mu_1 \ldots \sigma \ldots \mu_r}_{\phantom{\mu_1 \ldots \sigma \ldots \mu_r}\nu_1 \ldots \nu_s} \nabla_\sigma v^{\mu_i} + \sum_{j=1}^{s} T^{\mu_1 \ldots \mu_r}_{\phantom{\mu_1 \ldots \mu_r}\nu_1 \ldots \sigma \ldots \nu_s} \nabla_{\nu_j} v^\sigma \, . \tag{53}$$

The first term advects (drags) the tensor along the flow of $v$, i.e., this is the "naive" directional derivative part. The second and third terms account for how the basis vectors themselves are changing, due to curvature and due to the vector field $v$, respectively. It is easy to show that the three terms lead to pair-wise cancellations between the Christoffel symbols present in the three different covariant derivative terms of Eq. (53). Due to this, the whole expression boils down to Eq. (31), thus corroborating the connection independence of this derivative operator. Differently put, in the covariant derivatives, the Christoffel symbols introduce extra terms. However, the extra Christoffel terms cancel out between the different contributions (first, second, and third terms). One ends up getting exactly the same final expression for the Lie derivative as if only partial derivatives had been used – this is what is meant by "connection independence".

### A.3.4 GEODESIC EQUATION

*Geodesics* are paths that correspond to the most stationary trajectories (shortest or longest distance) that connect two points $p$ and $q$ on a manifold. Often, we only consider locally distance minimizing curves, and refer to them as *geodesics*. Geodesics are obtained by solving a calculus of variations problem on the distance metric $\mathcal{L}(\gamma)$ in Eq. (39), i.e.,

$$\delta\mathcal{L}(\gamma) := 0 \, , \tag{54}$$

(or alternatively on $\tau(\sigma)$ in the Lorentzian setting of Eq. (42)). Solving the calculus of variations problem boils down to solving the Euler-Lagrange equations, which mathematically is equivalent to the condition

$$\nabla_{\dot{\gamma}(t)}\dot{\gamma}(t) = 0 \, , \tag{55}$$

where $\gamma(t)$ is the curve (path) on the manifold, $\dot{\gamma}(t)$ the tangent vector (velocity vector), and $\nabla$ the covariant derivative. Eq. (55) intuitively says that the tangent vector is parallel transported along itself – meaning, one is moving without "acceleration" relative to the curved space. Thus, parallel transporting the tangent vector along the curve preserves the tangent vector. In numerical relativity, these corresponds to the equations of motion, i.e. a generalization of the Newton's acceleration equation $\frac{d^2x^\mu}{d\tau^2} = F^\mu/m$. For full derivations, we direct the readers to refer to Carroll et al. (2004); Misner et al. (2017); Poisson (2004).

We shall present the final form of a very central second-order ODE describing motion (acceleration) of objects executing geodesic paths around heavy gravitating bodies, namely, the *geodesic equation*

$$\frac{d^2x^\mu}{d\tau^2} + \Gamma^\mu_{\rho\sigma}(x)\frac{dx^\rho}{d\tau}\frac{dx^\sigma}{d\tau} = 0 \, , \tag{56}$$

where, $\tau$ is some affine paramter (typically, chosen to be the proper-time in Eq. (42)), $d^2x^\mu/d\tau^2$ is the four-acceleration vector, $dx^\rho/d\tau$ is the four-velocity and $\Gamma^\mu_{\rho\sigma}$ is the Christoffel symbols as seen in Eq. (51)).

Importantly, Eq. (55) is the geometric statement of the geodesic equation. It's coordinate-free, i.e., it's expressed entirely in terms of geometric objects. Eq. (56) is the coordinate version of the same idea. Here, one chooses a coordinate system $x^\mu$ on the manifold, and the covariant derivative $\nabla$ acting on a vector becomes the partial derivative plus correction terms involving the Christoffel symbols[8].

### A.3.5  Curvature tensors and scalars

Curvature tensors arise naturally in differential geometry as tensorial objects that capture the intrinsic and, where appropriate, extrinsic geometric properties of a manifold. They provide a coordinate-independent way to quantify the curvature of space or spacetime by encoding how the geometry deviates from flatness through the second derivatives, constructed out of Hessians of the metric tensor. Unlike artifacts that may arise from curvilinear coordinate choices on flat manifolds, curvature tensors reflect the true geometric content of a space. These generalize classical notions such as *Gaussian curvature* to higher dimensions and arbitrary signature. Being multilinear objects containing several tensor components, curvature tensors systematically characterize the variation of the metric across different directions. We shall introduce the key curvature related quantities, which include the Riemann relevant ones used in our paper in the following section.

**A.3.5.1  Riemann curvature tensor**  The *Riemann curvature tensor* $R^\mu_{\gamma\alpha\beta}(x)\, e_\mu \otimes \vartheta^\gamma \otimes \vartheta^\alpha \otimes \vartheta^\beta$ is a rank $(1,3)$ tensor, which quantifies the measure to which a vector that is transported along a small loop (also called *holonomy*) fails to return to its original orientation – due to the effect of the intrinsic curvature that the vector field picks up during the transport. The Riemann curvature tensor is defined via the commutators of covariant derivatives acting on components of a vector field $v$:

$$[\nabla_\alpha, \nabla_\beta]v^\delta(x) = \left(\nabla_\alpha\nabla_\beta - \nabla_\beta\nabla_\alpha\right)v^\delta(x) = R^\delta_{\alpha\beta\gamma}(x)v^\gamma(x) \, . \tag{57}$$

The components of the Riemann curvature tensor are expressed in terms of the Christoffel symbols

$$R^\delta_{\alpha\beta\gamma}(x) = \frac{\partial\Gamma^\delta_{\alpha\gamma}(x)}{\partial x^\beta} - \frac{\partial\Gamma^\delta_{\beta\gamma}(x)}{\partial x^\alpha} + \Gamma^\sigma_{\gamma\alpha}(x)\Gamma^\delta_{\beta\sigma}(x) - \Gamma^\sigma_{\beta\gamma}(x)\Gamma^\delta_{\sigma\alpha}(x) \, . \tag{58}$$

---

[8]Ideally, concepts such as connections, parallel transport and covariant derivatives are metric-independent formulation

The Riemann tensor $R_{\alpha\beta\gamma\delta} = g_{\alpha\sigma}R^{\sigma}_{\beta\gamma\delta}$ obeys the following identities:

$$R_{\alpha\beta\gamma\delta} = -R_{\alpha\beta\delta\gamma} \, ,$$
$$R_{\alpha\beta\gamma\delta} = -R_{\beta\alpha\gamma\delta} \, ,$$
$$R_{\alpha\beta\gamma\delta} = -R_{\gamma\delta\alpha\beta} \, .$$

The Riemann tensor in a $n$-dimensional manifold has $n^2(n^2 - 1)/12$ independent components. Importantly, it satisfies two additional identities, called the *Bianchi identities*

$$R_{\alpha\beta\gamma\delta} + R_{\alpha\gamma\delta\beta} + R_{\alpha\delta\beta\gamma} = 0 \text{ (Bianchi Identity I) } , \tag{59a}$$

$$\nabla_{\alpha}R_{\beta\gamma\delta\sigma} + \nabla_{\beta}R_{\gamma\alpha\delta\sigma} + \nabla_{\gamma}R_{\alpha\beta\delta\sigma} = 0 \text{ (Bianchi Identity II) } . \tag{59b}$$

Unlike the Christoffel symbols, which may be non-zero purely due to the choice of coordinates – e.g., when imposing curvilinear coordinates such as polar coordinates $(r, \vartheta)$ on the flat Cartesian plane – the Riemann curvature tensor encapsulates the true geometric curvature of a manifold. Since Christoffel symbols represent connection coefficients rather than tensorial objects, their non-vanishing components can give the false impression of intrinsic curvature, even on a flat manifold. In contrast, the Riemann tensor is a bona fide tensor and its vanishing is a coordinate-invariant statement: if the Riemann tensor vanishes in one coordinate system, it vanishes in all coordinate systems. Thus, it provides a definitive criterion for distinguishing truly curved spaces from flat ones, independent of coordinate artifacts.

**Geodesic deviation.** An important consequence of the existence of a non-zero Riemann tensor is that it encapsulates directional information about how geodesics path converge or diverge. Intuitively, it implies that in Euclidean space $\mathbb{R}^d$, parallel lines always remain parallel, but in the case of spherical geometry, say $\mathbb{S}^{d-1}$ (constant positive curvature) the parallel lines converge at a point, while for hyperbolic spaces $\mathbb{H}^d$, the parallel lines continue diverging. This is captured by the *geodesic deviation equation* (sometimes referred to as *Jacobi equation*) Isham (1999); Jost (2008); Poisson (2004), which shows how an infinitesimal neighborhood of a given geodesics diverge or converge. Here, we shall give the equation with a brief sketch.

**Theorem 2** (*Jacobi equation*): *Let $\gamma_s(\tau)$ be a family of closely spaced geodesics indexed by a smooth one-paramter family $s$ and $\tau \in \mathbb{R}$ the affine parameter. Let $x^{\mu}(s, \tau)$ be the coordinates of the geodesics $\gamma_s(\tau)$, then the tangent vector field is a directional derivative expressed in these coordinates as $X^{\mu} = \frac{dx^{\mu}(s,\tau)}{d\tau}$. Let the set of deviation vector fields $S^{\mu} = \frac{\partial x^{\mu}(\tau,s)}{\partial s}|_{\tau}$. Then, the deviation vector fields that satisfy the acceleration equation are called (Jacobi fields) and read*

$$\frac{D^2 S^{\mu}}{D\tau^2} = R^{\mu}_{\alpha\beta\gamma}X^{\alpha}X^{\beta}S^{\gamma} \, , \tag{60}$$

*where, $\frac{D}{D\tau} = X^{\alpha}\nabla_{\alpha}$ is the directional covariant derivative, i.e., the derivative of a vector field along a given direction on a manifold, while accounting for the manifold's curvature.*

### A.3.5.2 Contracted curvature tensors, scalars and invariants

**Ricci tensor.** From the rank $(1, 3)$ Riemann tensor, one can construct a traced (contracted) symmetric curvature tensor of rank $(0, 2)$, called the *Ricci tensor* $R_{\alpha\beta} \vartheta^{\alpha} \otimes \vartheta^{\beta}$,

$$R^{\gamma}_{\alpha\gamma\beta} = \text{Tr}_g(R^{\gamma}_{\alpha\delta\beta}) := R_{\alpha\beta} \, . \tag{61}$$

Mathematically, the Ricci tensor aggregates directional curvature along orthogonal planes. Thus, it can be considered as a curvature average of the Riemann tensor. It is closely related to the concept of sectional curvature and reflects how volume deformations occurs as one evolve under geodesic flow.

**Ricci scalar.** The traced (contracted) Ricci tensor yields a scalar field called the scalar curvature, also called the *Ricci scalar*. It is defined as

$$R^{\alpha}_{\alpha} = \text{Tr}_g(R_{\alpha\beta}) := R \, . \tag{62}$$

Mathematically, the scalar curvature corresponds to the sum/average over all sectional curvatures, i.e., $R(p) = \sum_{\alpha \neq \beta} \text{Sec}(e_\alpha, e_\beta)|_p \; \forall p \in \mathcal{M}$. For a point $p$, in an $n$-dimensional Riemannian manifold $(\mathcal{M}, g)$, it characterizes the volume of an $\epsilon$-radius ball in the manifold to the corresponding ball in Euclidean space, given by,

$$\text{Vol}\big(B_\epsilon(p)) \subset \mathcal{M}\big) = \text{Vol}\big(B_\epsilon(0) \subset \mathbb{R}^n\big)\left(1 - \frac{R}{6(n+2)}\epsilon^2 + \mathcal{O}(\epsilon^3)\right).$$

**Weyl tensor.** Another important tensor field of rank$(0, 4)$ is the Weyl tensor, which is obtained as the "trace-free" part of the Riemann tensor. Physically, the Weyl tensor describes the tidal force experienced by a body when moving along geodesics, and quantifies the shape distortion a body experiences due to tidal forces (e.g., water tides caused by the gravitational pull of the moon). In an $n$-dimensional manifold it is defined as:

$$C_{\alpha\beta\gamma\delta} = R_{\alpha\beta\gamma\delta} - \frac{1}{(n-2)}\left(R_{\alpha\delta}g_{\beta\gamma} - R_{\alpha\gamma}g_{\beta\delta} + R_{\beta\gamma}g_{\alpha\delta} - R_{\beta\delta}g_{\alpha\gamma}\right) \tag{63}$$
$$+ \frac{1}{(n-1)(n-2)}R\big(g_{\alpha\gamma}g_{\beta\delta} - g_{\alpha\delta}g_{\beta\gamma}\big).$$

Mathematically, the Weyl tensor corresponds to the only non-zero components of the Riemann tensor when looking at *Ricci-flat* manifolds, i.e. $R_{\alpha\beta} = 0$. This become relevant for e.g., vaccum solutions, a class of exact solutions where $R_{\alpha\beta} = 0$ of the Einstein equations in the absence of matter distribution.

**Curvature invariants.** Curvature invariants play a central role in the analysis of spacetime geometries in general relativity. These scalar quantities are constructed from contractions of curvature tensors and are manifestly invariant under general coordinate transformations. As such, they serve as powerful diagnostic tools for characterizing the local and global geometric and physical properties of spacetime, which includes the identification of "true" (genuine spacetime singularities) and "false" singularities (artifact of choice of coordinate charts).

Among the most prominent quadratic curvature invariants that is relevant to our simulations and features in our paper is the *Kretschmann scalar*, defined as the full contraction of the Riemann curvature tensor with itself:

$$\mathscr{K}(x) := R^{\alpha\beta\gamma\delta}(x)R_{\alpha\beta\gamma\delta}(x) = g^{\alpha\rho}(x)g^{\beta\sigma}(x)g^{\gamma\zeta}(x)g^{\delta\eta}(x)R_{\rho\sigma\zeta\eta}(x)R_{\alpha\beta\gamma\delta}(x). \tag{64}$$

The Kretschmann scalar provides a coordinate-independent measure of the magnitude of the curvature of spacetime and the singularity becomes blows-up, due to infinite curvature. Examples are Kretschmann scalars $\mathscr{K}$ for blackholes, and Weyl scalars $\Psi_4$ for gravitational wave astrophysics. They capture the presence of intrinsic curvatures even when the Ricci tensor itself vanishes. Thus, the Kretschmann scalar encodes geometric information in a frame-independent manner.

### A.3.6 STRESS-ENERGY-MOMENTUM TENSOR

The stress-energy-momentum tensor (or simply called the energy-momentum tensor) is a symmetric rank $(2, 0)$ tensor

$$T^{\alpha\beta} = T^{\alpha\beta}e_\alpha \otimes e_\beta. \tag{65}$$

Physically, $T^{\alpha\beta}$ is a generalization of the stress tensor in continuum and fluid mechanics. It stores the information of distribution of matter fields, i.e., sources or sinks as a $4 \times 4$ tensor, such as energy-density, energy-flux, momentum density, and momentum flux.

These matter fields satisfy the conservation laws, i.e., conservation of mass and energy via the four-dimensional *continuity equation* and corresponds to the *divergence-free* condition of the energy-momentum tensor

$$\nabla_\alpha T^{\alpha\beta}(x) = 0. \tag{66}$$

The stress-energy-momentum tensor features on the right hand side of the Einstein field equations (EFEs), and influences spacetime geometry by causing distortions on it.

### A.3.7 EINSTEIN FIELD EQUATIONS

The EFEs are a set of second-order non-linear PDEs containing geometry on the left hand side and the source on the right hand side. EFEs are obtained by combining all the differential geometric quantities from Eqs. (36, 61, 62, 65), together with the conservation laws for matter distribution of Eq. (66), resulting in

$$G_{\alpha\beta} = 8\pi G\, T_{\alpha\beta} \; . \tag{67}$$

Here, $G_{\alpha\beta} := R_{\alpha\beta} - \frac{1}{2}g_{\alpha\beta}R + \Lambda g_{\alpha\beta}$ is called the *Einstein tensor* and also satisfies the divergence-free condition $\nabla_{\alpha}G^{\alpha\beta} = 0$, which, is a consequence of the IInd Bianchi identity of Eq. (59b).

### A.3.8 COORDINATE-INDEPENDENCE OF GR

Fundamentally, GR posits a deeper symmetry class: *diffeomorphism covariance* (Misner et al., 2017). It asserts that the laws of physics are independent of any particular choice of coordinates or parametrization of the underlying smooth manifold. For example, the metric around the star, say sun, can be expressed in terms of the Schwarzschild metric (introduced in Section B.1). Here, the diffeomorphism acts as a gauge transformation (Tao, 2008) between two sets of metrics defined on the Lorentzian manifold $\mathcal{M}$, in this case an equivalence class of Lorentzian metrics $\mathtt{Riem}(\mathcal{M})$ describing the same spacetime geometry. This makes sure equations of motion, conservation laws, physical fields, etc. remain intact, hence, the term "covariance". In mathematical terms, Let, $\Phi \in \mathtt{Diff}(\mathcal{M})$ be a smooth, invertible map between $\mathcal{M}$ with a smooth inverse, $\Phi : \mathcal{M} \xrightarrow{\cong} \mathcal{M}$ such that:

$$\big(\Phi^*g\big)(v, w) := g\big(\Phi_*(v), \Phi_*(w)\big) \; . \tag{68}$$

Here, $\Phi_* : T\mathcal{M} \to T\mathcal{M}$, is the pushforward map defined on the tangent bundles. This means under diffeomorphisms the metric transforms via a pullback operation $\Phi^*g = g'$. I.e., $\bar{g}_{\alpha\beta}(\bar{x}) = \frac{\partial x^{\mu}}{\partial \bar{x}^{\alpha}}\frac{\partial x^{\nu}}{\partial \bar{x}^{\beta}}g_{\mu\nu}(x)$ are gauge equivalent. Additionally, GR also admits changes of local frames or bases (external symmetries) via the *general linear group* $\mathrm{GL}(4, \mathbb{R})$, i.e., invertible linear transformations at each point $p \in \mathcal{M}$. Thus, GR enjoys coordinate independence from two symmetry transformations, i.e., (i) between any particular choice of coordinates or parameterization of the underlying smooth manifold $\mathcal{M}$, and (ii) general linear group transformations that locally change frames of reference.

## B EXACT SOLUTIONS OF THE EINSTEIN FIELD EQUATIONS

This Appendix contains a detailed description of the exact solutions of EFEs corresponding to a class of metrics $g_{\mu\nu}$ that are solutions of Eq. (67). While there exist several geometries that satisfy the EFEs, we shall consider three prominent geometries: the Schwarzschild metric, the Kerr metric, and gravitational waves. These solutions not only have a high theoretical and historical relevance, but are also of great interest in computational black hole astrophysics and gravitational wave and multi-messenger astronomy. From here on, we work in naturalized units by setting $c = G = 1$.

Our work predominantly uses the exact solutions for generating synthetic training data, which are analytic expressions for (i) Schwarzschild, (ii) Kerr, and (iii) linearized gravity metrics, on which we fit the NeFs.

### B.1 SCHWARZSCHILD METRIC

The Schwarzschild metric is the simplest non-trivial solution to the EFEs. It describes the geometry around a non-rotating spherical body, such as a star or a black hole, constituting spherically symmetric vacuum solutions, i.e., $T_{\mu\nu} = 0$. A famous result of GR called the Birkhoff's theorem (Birkhoff & Langer, 1923) proves that any spherically symmetric vacuum solution corresponds to a static (non-rotating), time-independent (stationary), and asymptotically flat metric (i.e., for $r \to \infty$ the metric converges to the flat Minkowski spacetime), and must essentially be equivalent to the Schwarzschild solution.

### B.1.1 COORDINATE SYSTEMS FOR SCHWARZSCHILD METRICS

**Spherical polar coordinates.** Schwarzschild solution is typically written in the convential spherical polar coordinates $(t, r, \theta, \phi)$ where $t \in \mathbb{R}$, $r \in \mathbb{R}^+$, $\theta \in (0, \pi)$, and $\phi \in [0, 2\pi)$. The metric can be written either using the quadratic line element

$$ds^2 = -\left(1 - \frac{2M}{r}\right)dt^2 + \left(1 - \frac{2M}{r}\right)^{-1}dr^2 + r^2\left(d\theta^2 + \sin^2\theta d\phi^2\right) . \tag{69}$$

or in the equivalent matrix notation

$$g_{\mu\nu}^{\text{Sph}} = \begin{pmatrix} -\left(1 - \frac{2M}{r}\right) & 0 & 0 & 0 \\ 0 & \left(1 - \frac{2M}{r}\right)^{-1} & 0 & 0 \\ 0 & 0 & r^2 & 0 \\ 0 & 0 & 0 & r^2\sin^2\theta \end{pmatrix} . \tag{70}$$

The true singularity of the Schwarzschild metric is at the origin and can be identified from the divergent Kretschmann scalar (Eq. (64)):

$$\mathscr{K}(r) = \frac{48M^2}{r^6} \xrightarrow{r \to 0} \infty . \tag{71}$$

Although a (fake) coordinate singularity exists at $r = r_s = 2M$, where the Kretschmann scalar is well defined. This special radius $r_s$ is called the Schwarzschild radius. It demarcates the location of the *event horizon* of a non-rotating black hole and delineates a region from which no causal signal can escape to asymptotic infinity, meaning, it is a point of no return for any body (including light) once it crosses this critical radius.

**Cartesian Kerr-Schild coordinates.** The Kerr-Schild (KS) form is a beneficial representation for finding exact solutions to the EFEs. These are perturbative corrections to a spacetime metric only upto the linear order (Kerr & Schild, 2009). The KS form enables the following simplifications to the nonlinear field equations : (i) It expresses the resultant metric as a linearized perturbation to the background metric, and (ii) gets rid of the coordinate singularities, which are mere artifacts of an unsuitable choice of coordinate systems. The corresponding line element expressed in the KS form reads

$$ds^2 = (\bar{g}_{\alpha\beta} + V(x)\ell_\alpha\ell_\beta)dx^\alpha dx^\beta , \tag{72}$$

where $\bar{g}_{\alpha\beta}$ is some background metric, $l_\alpha$ are the components of a null vector $\boldsymbol{\ell}$ with respect to the background metric and $V(x)$ is a scalar.

For a spherically symmetric non-rotating blackhole such as Schwarzschild, the Cartesian Kerr-Schild line element is obtained by setting $\bar{g}_{\alpha\beta} = \eta_{\alpha\beta}$, $\boldsymbol{\ell} = \left(1, \frac{x}{r}, \frac{y}{r}, \frac{z}{r}\right)$ and $V = \frac{2M}{r}$:

$$ds^2 = -dt^2 + dx^2 + dy^2 + dz^2 + \frac{2M}{r}\left[dt + \frac{x}{r}dx + \frac{y}{r}dy + \frac{z}{r}dz\right]^2 . \tag{73}$$

Unlike the spherical coordinate form in Eq. (69), $r = 2M$ is not singular, hence removing the coordinate singularities. The metric tensor components read:

$$g_{\mu\nu}^{\text{KS}} = \begin{pmatrix} -1 + 2M/r & \dfrac{2Mx}{r^2} & \dfrac{2My}{r^2} & \dfrac{2Mz}{r^2} \\ \dfrac{2Mx}{r^2} & 1 + \dfrac{2Mx^2}{r^3} & \dfrac{2Mxy}{r^3} & \dfrac{2Mxz}{r^3} \\ \dfrac{2My}{r^2} & \dfrac{2Mxy}{r^3} & 1 + \dfrac{2My^2}{r^3} & \dfrac{2Myz}{r^3} \\ \dfrac{2Mz}{r^2} & \dfrac{2Mxz}{r^3} & \dfrac{2Myz}{r^3} & 1 + \dfrac{2Mz^2}{r^3} \end{pmatrix} . \tag{74}$$

$$\tag{75}$$

**Ingoing (advanced) Eddington-Finkelstein coordinates** The ingoing version of the Eddington-Finkelstein (EF) coordinates is obtained by replacing time $t$ with an advanced time coordinate $v = t + r_\star(r)$, where $r_\star = r + M \log\left|\frac{r-2M}{2M}\right|$. Thus, $dt$ in these transformed coordinates amounts to:

$$dt = dv - dr_* = dv - \left(1 - \frac{2M}{r}\right)^{-1} dr$$

The ingoing EF version of the Schwarzschild metric reads:

$$ds^2 = -\left(1 - \frac{2M}{r}\right)dv^2 + 2dv\, dr + r^2\left(d\theta^2 + \sin^2\theta\, d\phi^2\right) , \tag{76}$$

With the metric tensor being:

$$g_{\mu\nu}^{\text{EF}} = \begin{pmatrix} -\left(1 - \dfrac{2M}{r}\right) & 1 & 0 & 0 \\ 1 & 0 & 0 & 0 \\ 0 & 0 & r^2 & 0 \\ 0 & 0 & 0 & r^2 \sin^2\vartheta \end{pmatrix}. \tag{77}$$

This metric is smooth (and non-degenerate), devoid of coordinate singularities at the event horizon $r = r_s = 2M$, and can be continued down to the curvature singularity at $r = 0$ (Carroll et al., 2004; Chandrasekhar, 1984; Frolov & Novikov, 1998).

## B.2 KERR METRIC

The Kerr solution describes a massive gravitating body rotating with an angular momentum $J$. From the physics perspective, it is not symmetric under time-reversal symmetry, i.e., $t \to -t$, hence corresponds to a stationary but a non-static solution (Teukolsky, 2015). Due to a finite angular momentum $J$, or equivalently, rotation parameter $a = \frac{J}{M} > 0$, the Kerr metric introduces an asymmetry, and is oblate. Thus, the Kerr metric corresponds to an oblate spheroid geometry.

### B.2.1 COORDINATE SYSTEMS FOR KERR METRIC

**Boyer-Lindquist coordinates.** The Boyer-Lindquist (BL) coordinates are a special and convenient representation for the Kerr metric (Boyer & Lindquist, 1967; Visser, 2008; Teukolsky, 2015). The BL form $(t, r, \vartheta, \phi)$ is described by oblate spheroidal coordinates (Krasiński, 1978):

$$x = \sqrt{r^2 + a^2}\, \sin\vartheta\, \cos\phi \tag{78a}$$

$$y = \sqrt{r^2 + a^2}\, \sin\vartheta\, \sin\phi \tag{78b}$$

$$z = r\, \cos\vartheta . \tag{78c}$$

Notice that the zenith angle $\vartheta \neq \theta$ differs from the Schwarzschild case, while the azimuthal angle $\phi$ is the same in both. As $a \to 0$, the Kerr metric boils down to the non-rotating spherical case of the Schwarzschild metric.

$$ds^2 = -\left(1 - \frac{2Mr}{\Sigma}\right)dt^2 - \frac{4Mar\sin^2\vartheta}{\Sigma}dtd\phi + \frac{\Sigma}{\Delta}dr^2 + \Sigma d\vartheta^2 + \left(r^2 + a^2 + \frac{2Mra^2\sin^2\vartheta}{\Sigma}\right)\sin^2\vartheta d\phi^2 ,$$
$$\tag{79}$$

where the length scales are $a = \frac{J}{M}$ (angular momentum per unit mass), $\Sigma \equiv r^2 + a^2\cos^2\vartheta$, and $\Delta \equiv r^2 - 2Mr + a^2$. The Kerr curvature singularity occurs at $\Sigma := r^2 + a^2\cos^2\vartheta = 0$, implying $r = 0$ and $\vartheta = \frac{\pi}{2}$. The metric tensor of Eq. (79) is:

$$
g_{\mu\nu}^{\text{BL}} = \begin{pmatrix} -\left(1 - \dfrac{2Mr}{\Sigma}\right) & 0 & 0 & -\dfrac{2Mar\sin^2\vartheta}{\Sigma} \\[2ex] 0 & \dfrac{\Sigma}{\Delta} & 0 & 0 \\[2ex] 0 & 0 & \Sigma & 0 \\[2ex] -\dfrac{2Mar\sin^2\vartheta}{\Sigma} & 0 & 0 & \left(r^2 + a^2 + \dfrac{2Ma^2r\sin^2\vartheta}{\Sigma}\right)\sin^2\vartheta \end{pmatrix} . \tag{80}
$$

In the Boyer-Lindquist form of the metric, there also exist coordinate singularities at $\Delta = r^2 - 2Mr + a^2 = 0$. Thus, the roots of $\Delta = 0$ are $r_{\pm} = M \pm \sqrt{M^2 - a^2}$, which demarcate the outer and inner horizons.

It is easy to see the existence of a curvature singularity at $r = 0$ on the equatorial plane corresponding to the zenith angle $\vartheta = \frac{\pi}{2}$. Thus, unlike Schwarzschild, the singularity in Kerr geometry takes the form of a ring, also known as a *ring singularity*.

**Cartesian Kerr-Schild coordinates.** The Cartesian KS form of the Kerr metric is obtained by setting in Eq. (72) $\ell = \left(1, \frac{rx+ay}{r^2+a^2}, \frac{ry-ax}{r^2+a^2}, \frac{z}{r}\right)$ and $V = \frac{mr^3}{r^4+a^2z^2}$ in Eq. (72). The Kerr metric in Kerr coordinates are often used to write initial data for hydro simulations. The line-element in the Cartesian Kerr-Schild form reads (Teukolsky, 2015):

$$
ds^2 = -dt^2 + dx^2 + dy^2 + dz^2 + \frac{2mr^3}{r^4 + a^2z^2}\left[dt + \frac{r(xdx + ydy)}{a^2 + r^2} + \frac{a(ydx - xdy)}{a^2 + r^2} + \frac{z}{r}dz\right]^2 . \tag{81}
$$

Here, $r \equiv r(x, y, z)$ is not a coordinate, and is given implicitly by solving the quadratic equation $\frac{x^2+y^2}{r^2+a^2} + \frac{z^2}{r^2} = 1$: The solution for the implicit function $r$ is given by the discriminant (Visser, 2008):

$$
r^2(x, y, z) = \frac{x^2 + y^2 + z^2 - a^2}{2} + \sqrt{\frac{(x^2 + y^2 + z^2 - a^2)^2 + 4a^2z^2}{4}} .
$$

The corresponding Cartesian coordinates are expressed as:

$$
x = (r\cos\varphi - a\sin\varphi)\sin\vartheta = \sqrt{r^2 + a^2}\,\sin\vartheta\,\cos\left(\varphi + \tan^{-1}(a/r)\right) ,
$$
$$
y = (r\sin\varphi + a\cos\varphi)\sin\vartheta = \sqrt{r^2 + a^2}\,\sin\vartheta\,\sin\left(\varphi + \tan^{-1}(a/r)\right) ,
$$
$$
z = r\cos\vartheta .
$$

In the BL coordinates the ring singularity for Kerr exists at $r = 0$ & $\vartheta = \frac{\pi}{2}$, translating to $z = 0$ (equatorial-plane), and the ring occurring at $x^2 + y^2 = a^2$. In contrast, the KS representation is devoid of *coordinate singularities*, making it suitable to work in numerics, especially around the event-horizons.

**Ingoing Eddington-Finkelstein coordinates.** In the original formulation, the Kerr metric is written in the advanced time coordinates/ingoing EF coordinates $v$. The line element in this representation reads (Teukolsky, 2015):

$$
ds^2 = -\left(1 - \frac{2Mr}{r^2 + a^2\cos^2\theta}\right)(dv + a\sin^2\theta\,d\tilde{\phi})^2 \tag{82}
$$
$$
+ 2(dv + a\sin^2\theta d\tilde{\phi})(dr + a\sin^2\theta d\tilde{\phi})
$$
$$
+ (r^2 + a^2\cos^2\theta)(d\theta^2 + \sin^2\theta d\tilde{\phi}^2) ,
$$

where the ingoing EF coordinates are related to the Boyer-Lindquist coordinates Eq. (78) by the following transformation:

$$
v = t + \int \frac{(r^2 + a^2)}{\Delta}dr ,
$$
$$
\tilde{\phi} = \phi + a\int \frac{dr}{\Delta} ,
$$

where, $\Delta \equiv r^2 - 2Mr + a^2$. and the metric tensor components corresponding to the line element is given by

$$g_{\mu\nu}^{\text{EF}} = \begin{pmatrix} -\left(1 - \dfrac{2Mr}{\Sigma}\right) & 1 & 0 & \dfrac{2Mar\sin^2\theta}{\Sigma} \\ 1 & 0 & 0 & a\sin^2\theta \\ 0 & 0 & \Sigma & 0 \\ \dfrac{2Mar\sin^2\theta}{\Sigma} & a\sin^2\theta & 0 & \left(r^2 + a^2 + \dfrac{2Ma^2r\sin^2\theta}{\Sigma}\right)\sin^2\theta \end{pmatrix}. \tag{83}$$

The coordinate (fake) singularities ($\mathscr{K} < \infty$) of the Kerr metric is given by the zeros of $\Delta \equiv r^2 - 2Mr + a^2 = 0$. Solving for the zeros, one finds

$$r_\pm = M \pm \sqrt{M^2 - a^2}\,,$$

where, $r_+$ is the outer event horizon, while $r_-$ demarcates the inner event horizon.

Apart from that, rotating metrics also possess a highly interesting region known as the *ergosphere*, which fundamentally captures the non-Euclidean and non-inertial nature of general relativistic effects induced by rotation. This domain is situated outside the outer event horizon $r_+$, and is created owing to the frame-dragging (Lense-Thirring) effect. Consequently, no physical observer (test body) can remain static within the ergosphere and is compelled to co-rotate with the black hole depending on the value of $a$. The location of the ergosphere is given by

$$r_\pm^{\text{ergo}}(\vartheta) = M \pm \sqrt{M^2 - a^2\cos^2\vartheta}\,,$$

where, $r_+^{\text{ergo}}$ is the outer ergosphere, while $r_-^{\text{ergo}}$ demarcates the inner ergosphere. In Figure 8, the following regions of the Kerr metric are demarcated:

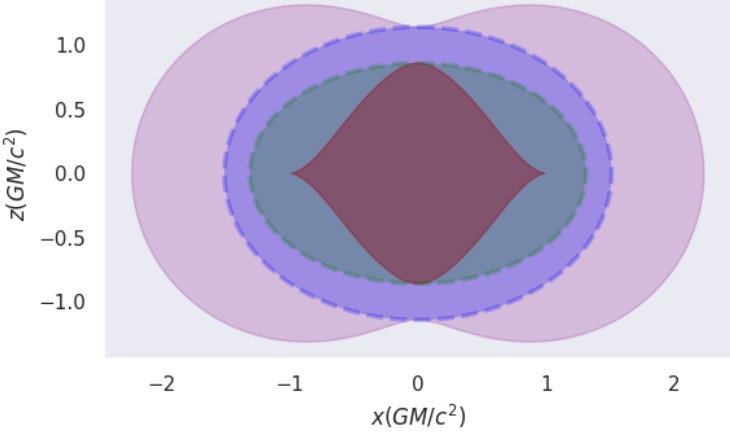

Figure 8: Kerr metric 2D slice in the x-z plane ($y = 0$) for a spin parameter $a = 0.99$. The following regions plotted are: i) inner ergosphere $r_-^{\text{ergo}}$: red region, ii) inner event-horizon $r_-$: green region, iii) outer event-horizon $r_+$: blue region and iv) outer Ergosphere $r_+^{\text{ergo}}$: purple region.

## B.3 GRAVITATIONAL WAVES

Linearized gravity models the metric as tiny fluctuations or *perturbations* $h_{\alpha\beta}$ of the flat background metric $\eta_{\alpha\beta}$:

$$g_{\alpha\beta} \approx \eta_{\alpha\beta} + h_{\alpha\beta} + \mathcal{O}(h_{\alpha\beta})^2\,,$$

where $|h_{\alpha\beta}| \ll 1$. To describe gravitational wave propagation, it is often convenient to reduce the linearized field equations into a simplified form via two *gauge fixing* conditions, namely, a) *harmonic*

*gauge*, and b) *transverse-traceless* (TT) gauge. Thus, the Einstein field equations for gravitational waves assume a succinct wave equation type form:

$$\Box \overline{h}^{(\epsilon)}_{\alpha\beta} = -16\pi T_{\alpha\beta} \ . \tag{84}$$

Here, $\overline{h}^{(\epsilon)}_{\alpha\beta} = h^{(\epsilon)}_{\alpha\beta} - \frac{1}{2}h^{(\epsilon)}\eta_{\alpha\beta}$ and $\Box \equiv \eta^{\alpha\beta}\partial_\alpha\partial_\beta$ is the d'Alembert operator (wave operator). It can be shown that the PDEs in Eq. (84) produce gravitational wave solutions.

The transverse-traceless (TT) perturbation (we drop the superscript $(\epsilon)$ for the sake of ease) satisfies the following conditions:

**Transverse:** $\partial_\beta h^{\mathrm{TT}\,\alpha\beta} = 0$, i.e., wave propagates perpendicular to perturbation direction,

**Traceless:** $h^{\mathrm{TT}\,\alpha}_{\ \ \alpha} = 0$,

**Purely spatial:** $h^{\mathrm{TT}}_{0\alpha} = 0$, i.e., no time components.

Thus, a gravitational wave propagating in the $z$-direction with frequency $\omega$ is given in the TT gauge as:

$$h^{\mathrm{TT}}_{\alpha\beta} = \begin{pmatrix} 0 & 0 & 0 & 0 \\ 0 & h_+ & h_\times & 0 \\ 0 & h_\times & h_+ & 0 \\ 0 & 0 & 0 & 0 \end{pmatrix} \cos\big(\omega(t-z)\big) \ . \tag{85}$$

$h_+$ and $h_\times$ are the amplitudes of the "+" (plus) polarization and "×" (cross) polarization.

The complete metric tensor in the linearized gravity setting is given by:

$$g_{\alpha\beta} = \eta_{\alpha\beta} + h^{\mathrm{TT}}_{\alpha\beta} = \begin{pmatrix} -1 & 0 & 0 & 0 \\ 0 & 1 + h_+\cos\big(\omega(t-z)\big) & h_\times\cos\big(\omega(t-z)\big) & 0 \\ 0 & h_\times\cos\big(\omega(t-z)\big) & 1 + h_+\cos\big(\omega(t-z)\big) & 0 \\ 0 & 0 & 0 & 1 \end{pmatrix} \ . \tag{86}$$

The corresponding line-element in the linearized gravity setting reads:

$$ds^2 = -dt^2 + \big[1 + h_+\cos(\omega(t-z))\big]\,dx^2 + \big[1 - h_+\cos(\omega(t-z))\big]\,dy^2$$
$$+ 2h_\times\cos(\omega(t-z))\,dx\,dy \tag{87}$$

**Spin-weighted spherical harmonics (SWSH) metric representation.** We start from the decomposition of the complex gravitational wave strain with the spherical harmonic basis-set expansion (Newman & Penrose, 1962). With the expansion in mode weights $h^{lm}(t, r)$, one can ignore (remove) the angular dependence:

$$h(t, r, \theta, \phi) = h_+(t, r, \theta, \phi) - ih_\times(t, r, \theta, \phi) = \frac{M}{r} \sum_{\ell=|s|}^{\infty} \sum_{|m|\leq\ell}^{\ell} h^{\ell m}(t)\,_{-2}Y_{\ell m}(\theta, \phi) \ , \tag{88}$$

where, $h(t, r, \theta, \phi) = h_+(t, r, \theta, \phi) - ih_\times(t, r, \theta, \phi)$ is the complex strain. Thus, for each orbital and azimuthal indices $(\ell, m)$, one can extract the mode $h^{\ell m}(t)$ at a fixed radius $r$, one uses the orthogonality of the spin-weighted spherical harmonics (SWSHs) elements:

$$\frac{r}{M}h^{\ell m}(t) = \int_0^{2\pi}\int_0^\pi h(t, r, \theta, \phi)\,_{-2}\bar{Y}_{\ell m}(\theta, \phi)\,d\Omega \tag{89}$$

where, $d\Omega = \sin\theta\,d\theta\,d\phi$ is the spherical volume element and $_{-2}\bar{Y}_{\ell m}$ denotes the complex conjugate of the $s = 2 - n = -2$ where ($n = 4$ for $\Psi_4$) spin-weighted spherical harmonics. One typically carries out the integral over the 2-sphere $\mathbb{S}^2$ numerically on a finite angular grid.

The general formula for SWSHs is:

$$_sY_{\ell m}(\theta,\phi) = (-1)^{l+m-s}\left[\frac{2\ell+1}{4\pi}\frac{(\ell+m)!(\ell-m)!}{(\ell+s)!(\ell-s)!}\right]^{1/2}\sin^{2\ell}\left(\frac{\theta}{2}\right)e^{im\phi}$$

$$\sum_{r=0}^{\ell-s}(-1)^r\binom{\ell-s}{r}\binom{\ell+s}{r+s-m}\cot^{2r+s-m}\left(\frac{\theta}{2}\right).$$

where the parameters $\ell, m$ are the familiar Laplace spherical harmonics (orbital-angular momentum and azimuthal indices), while $s$ is the additional spin-weight introduced by some underlying gauge group such as $U(1)$. We especially plot for the integers $s = -2, l = m = 2$, since they are relevant for GWs and are depicted in Figure 9.

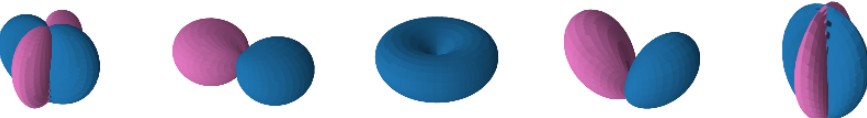

Figure 9: Spin weighted spherical harmonics for $s = -2$, and $l = 2$ for $|m| \leq l$. The dominant contributions for the Weyl scalar $\Psi_4$ and the associated metric coefficients in the spherical harmonic basis $h^{2,\pm2}(t)$ are shown.

### B.4    MINKOWKSI METRIC

#### B.4.1    COORDINATE SYSTEMS FOR MINKOWSKI METRIC

The flat Minkowski metric, which is a spacetime that has no curvature ($M \to 0$) can be expressed in other coordinate systems as well.

**Spherical polar coordinates.**   In spherical coordinates $(t, r, \vartheta, \varphi)$, the Minkowski metric is described by the quadratic line element,

$$ds^2 = -c^2dt^2 + dr^2 + r^2(d\theta^2 + \sin^2 d\phi^2) \tag{91}$$

here, $r \in \mathbb{R}^+$, $\theta \in (0, \pi)$, and, $\phi \in [0, 2\pi)$ are the usual spherical polar coordinates. Thus, the metric tensor describing the Schwarzschild solution reads:

$$\eta_{\mu\nu}^{\text{Sph}} = \begin{pmatrix} -1 & 0 & 0 & 0 \\ 0 & 1 & 0 & 0 \\ 0 & 0 & r^2 & 0 \\ 0 & 0 & 0 & r^2\sin^2\theta \end{pmatrix} \tag{92}$$

**Boyer-Lindquist coordinates.**   Setting $M \to 0$ in the Boyer-Lindquist form of the Kerr metric Eq. (79), the corresponding line element reduces to an unfamiliar "oblate-spheroidal" represtation:

$$ds^2 = -dt^2 + \frac{r^2 + a^2\cos^2\vartheta}{r^2 + a^2}\,dr^2 + (r^2 + a^2\cos^2\vartheta)\,d\vartheta^2 + (r^2 + a^2)\sin^2\vartheta\,d\phi^2\,, \tag{93}$$

and the components of the

$$\eta_{\mu\nu}^{\text{BL}} = \begin{pmatrix} -1 & 0 & 0 & 0 \\ 0 & \frac{r^2+a^2\cos^2\vartheta}{r^2+a^2} & 0 & 0 \\ 0 & 0 & r^2+a^2\cos^2\vartheta & 0 \\ 0 & 0 & 0 & (r^2+a^2)\sin^2\vartheta \end{pmatrix}. \tag{94}$$

The usual Cartesian coordinates can be related to the oblate spheroid ones via:

$$x = \sqrt{r^2 + a^2}\,\sin\vartheta\,\cos\phi$$

$$y = \sqrt{r^2 + a^2}\,\sin\vartheta\,\sin\phi$$

$$z = r\cos\vartheta\,.$$

**Eddington-Finkelstein coordinates.** The Minkowski metric can be written in the ingoing (advanced) Eddington-Finkelstein form in two different cases, namely for the non-rotating and the rotating case.

- i) non-rotating, $a \to 0$ (Schwarzschild): :
$$ds^2 = -dv^2 + 2\, dv\, dr + r^2(d\theta^2 + \sin^2\vartheta\, d\phi^2) \,,$$
and the metric tensor reads:
$$\eta_{\mu\nu}^{\text{EF}} = \begin{pmatrix} -1 & 1 & 0 & 0 \\ 1 & 0 & 0 & 0 \\ 0 & 0 & r^2 & 0 \\ 0 & 0 & 0 & r^2\sin^2\theta \end{pmatrix} \,. \tag{95}$$

- ii) rotating: $a > 0$ (Kerr):
$$ds^2 = -(dv + a\,\sin^2\vartheta d\phi)^2 + 2(dv + a\,\sin^2\vartheta d\phi)(dr + a\sin^2\vartheta d\phi)$$
$$+ (r^2 + a^2\cos^2\vartheta)(d\theta^2 + \sin^2 d\phi^2) \,, \tag{96}$$
and the metric tensor reads:
$$\eta_{\mu\nu}^{\text{EF}} = \begin{pmatrix} -1 & 1 & 0 & 0 \\ 1 & 0 & 0 & a\sin^2\vartheta \\ 0 & 0 & \Sigma & 0 \\ 0 & a\sin^2\vartheta & 0 & (r^2 + a^2)\sin^2\vartheta \end{pmatrix} \,. \tag{97}$$

### B.5 TRAINING ON NON-TRIVIAL METRIC FIELDS (DISTORTIONS)

*Distortion part of Schwarzschild geometry in spherical coordinates*:

- **Spherical coordinates**: obtained by subtracting Eq. (92) from Eq. (70):
$$g_{\mu\nu}^{\text{Sph}} - \eta_{\mu\nu}^{\text{Sph}} = \begin{pmatrix} \dfrac{r_s}{r} & 0 & 0 & 0 \\ 0 & \dfrac{r_s}{(r - r_s)} & 0 & 0 \\ 0 & 0 & 0 & 0 \\ 0 & 0 & 0 & 0 \end{pmatrix} \,. \tag{98}$$

- **Kerr-Schild coordinates** obtained by subtracting $\eta_{\mu\nu} = \text{diag}(-1, +1, +1, +1)$ from Eq. (74):
$$g_{\mu\nu}^{\text{KS}} - \eta_{\mu\nu} = \begin{pmatrix} \dfrac{2M}{r} & \dfrac{2Mx}{r^2} & \dfrac{2My}{r^2} & \dfrac{2Mz}{r^2} \\ \dfrac{2Mx}{r^2} & \dfrac{2Mx^2}{r^3} & \dfrac{2Mxy}{r^3} & \dfrac{2Mxz}{r^3} \\ \dfrac{2My}{r^2} & \dfrac{2Mxy}{r^3} & \dfrac{2My^2}{r^3} & \dfrac{2Myz}{r^3} \\ \dfrac{2Mz}{r^2} & \dfrac{2Mxz}{r^3} & \dfrac{2Myz}{r^3} & \dfrac{2Mz^2}{r^3} \end{pmatrix} \,. \tag{99}$$

- **Ingoing Eddington-Finkelstein coordinates** obtained by subtracting Eq. (95) from Eq. (77):
$$g_{\mu\nu}^{\text{EF}} - \eta_{\mu\nu}^{\text{EF}} = \begin{pmatrix} \dfrac{r_s}{r} & 0 & 0 & 0 \\ 0 & 0 & 0 & 0 \\ 0 & 0 & 0 & 0 \\ 0 & 0 & 0 & 0 \end{pmatrix} \,. \tag{100}$$

*Distortion part of Kerr geometry*:

- **Boyer-Lindquist coordinates**: obtained by subtracting Eq. (94) from Eq. (80):

$$g^{\text{BL}}_{\mu\nu} - \eta^{\text{BL}}_{\mu\nu} = \begin{pmatrix} \dfrac{2Mr}{\Sigma} & 0 & 0 & \dfrac{2Mar\sin^2\theta}{\Sigma} \\ 0 & \dfrac{2M\Sigma}{r\Delta} & 0 & 0 \\ 0 & 0 & 0 & 0 \\ \dfrac{2Mar\sin^2\theta}{\Sigma} & 0 & 0 & \dfrac{2Ma^2r\sin^4\vartheta}{\Sigma} \end{pmatrix}. \tag{101}$$

- **Kerr-Schild coordinates**: obtained by subtracting $\eta = \text{diag}(-1, +1, +1, +1)$ from Eq. (81):

- **Eddington-Finkelstein coordinates**: obtained by subtracting Eq. (97) from Eq. (83):

$$g^{\text{EF}}_{\mu\nu} - \eta^{\text{EF}}_{\mu\nu} = \begin{pmatrix} \dfrac{2Mr}{\Sigma} & 0 & 0 & \dfrac{2Mar\sin^2\theta}{\Sigma} \\ 0 & 0 & 0 & 0 \\ 0 & 0 & 0 & 0 \\ \dfrac{2Mar\sin^2\theta}{\Sigma} & 0 & 0 & \dfrac{2Ma^2r\sin^4\vartheta}{\Sigma} \end{pmatrix}. \tag{102}$$

## C  FINITE-DIFFERENCE METHOD (FDM) FOR TENSOR DIFFERENTIATION

The main concept in this appendix section details numerical differentiation methods for tensor-valued quantities, focusing on the practical use of higher-order finite-difference schemes (in particular, sixth-order stencils). We outline the treatment of discretization errors and the use of neighboring grid collocation points as part of a numerical tensor calculus toolbox.

To compare the performance against automatic differentiation on tensor fields defined on the four dimensional spacetime, throughout the paper we opt for the highly accurate sixth-order order *forward difference* ($n = 6$ accuracy). This scheme queries six neighboring points per evaluation and the general formula of the differential operators are given by:

$$\left[\partial_{x^i} f(\mathbf{x})\right] \approx -\frac{49}{20h}f(\mathbf{x}) + \frac{6}{h}f(\mathbf{x}+\mathbf{h}) - \frac{15}{2h}f(\mathbf{x}+2\mathbf{h}) + \frac{20}{3h}f(\mathbf{x}+3\mathbf{h})$$
$$- \frac{15}{4h}f(\mathbf{x}+4\mathbf{h}) + \frac{6}{5h}f(\mathbf{x}+5\mathbf{h}) - \frac{1}{6h}f(\mathbf{x}+6\mathbf{h}) + \mathcal{O}(\mathbf{h}^7) \tag{103}$$

Here, $\mathbf{x} = (x_1, ..., x_d) \in \mathbb{R}^d$ and the $\mathbf{h} = h\mathbf{e}_i$, s.t. $\mathbf{e}_i = (0, .., \overbrace{h}^{\text{i-th index}}, .., 0)$, depending with respect to the variable $x_i$ the partial derivative is performed over. Thus, this is accurate upto $\mathcal{O}(\boldsymbol{h}^6)$, and the truncation error occurs at seventh-order.

In general, for an $n$-th order finite-difference approximation, the stencil is constructed by querying $n$ neighboring collocation points on the voxel grid. The resulting truncation error on function evaluated on the grid (gridfunctions) scales as follows (Ruchlin et al., 2018):

$$\mathscr{E}^n_{\text{FD}}[f] = \mathcal{O}\left(h^n|\partial_{\mathbf{x}}^{n+1}f|\right).$$

Thus, higher order stencils enable larger step size $h$ choices since the error scales exponential to the stencil order, i.e, $\mathscr{E} \propto h^n$. This results in not only better accuracy, but also lesser memory consumption due to lower grid resolution.

**Finite-difference method bottlenecks in NR**

- Higher-order finite-difference stencils require a collection of padded grid points exterior to the cube as boundary handling. These are often called ghost cells (zones). For e.g., if $n_g$ ghost cells are required for the $n$-th order forward difference stencil, the grid size increases of a 3D spatial voxel grid (Ruchlin et al., 2018), $\mathbb{N}_x \times \mathbb{N}_y \times \mathbb{N}_z \to (\mathbb{N}_x + n_g) \times (\mathbb{N}_y + n_g) \times (\mathbb{N}_z + n_g)$.
- Sensitive to numerical noise especially for tensor-valued functions defined on multidimensional voxel grids.

# D SUCCINCT INTRODUCTION TO GENERAL RELATIVITY, EQUATIONS OF MOTION AND EXACT SOLUTIONS

**Derivative operators.** The metric and its partial derivatives can be used to construct the Christoffel symbols

$$\Gamma^\rho_{\mu\nu}(x) := \frac{1}{2} g^{\rho\sigma} \left( \partial_\mu g_{\sigma\nu}(x) + \partial_\nu g_{\sigma\mu}(x) - \partial_\sigma g_{\mu\nu}(x) \right).$$

The Christoffel symbols denote how the metric varies across spacetime and define a parallel transport machinery to translate tensor fields around the manifold. With these, it is possible to construct two pivotal modified tensor differentiation operators, namely: (i) The *covariant derivative* (also called the *Levi-Civita connection*), which can be seen as a "calibration" of the partial derivative operator for parallel transportation in the curvilinear setting:

$$\nabla_\mu T^{\alpha_1 \dots \alpha_r}{}_{\beta_1 \dots \beta_s} = \frac{\partial}{\partial x^\mu} T^{\alpha_1 \dots \alpha_r}{}_{\beta_1 \dots \beta_s} + \sum_{i=1}^r \Gamma^{\alpha_i}_{\mu\sigma} T^{\alpha_1 \dots \sigma \dots \alpha_r}{}_{\beta_1 \dots \beta_s} - \sum_{j=1}^s \Gamma^\sigma_{\mu\beta_j} T^{\alpha_1 \dots \alpha_r}{}_{\beta_1 \dots \sigma \dots \beta_s},$$

(ii) The *Lie derivative*, which generalizes the notion of a directional derivative that is connection independent (cf. Appendix A.3.3.2). The Lie derivative captures infinitesimal dragging of the tensor field along the flow generated by the vector field $\xi$:

$$(\mathcal{L}_\xi T)^{\alpha_1 \dots \alpha_r}{}_{\beta_1 \dots \beta_s} = \xi^\mu \partial_\mu T^{\alpha_1 \dots \alpha_r}{}_{\beta_1 \dots \beta_s} - \sum_{i=1}^r T^{\alpha_1 \dots \mu \dots \alpha_r}{}_{\beta_1 \dots \beta_s} \partial_\mu \xi^{a_i} + \sum_{j=1}^s T^{\alpha_1 \dots \alpha_r}{}_{\beta_1 \dots \mu \dots \beta_s} \partial_{\beta_j} \xi^\mu.$$

**Differential geometric objects.** Using the modified derivatives, we can construct a hierarchy of higher-rank differential geometric quantities, such as the Riemannian curvature tensor $R^\delta_{\alpha\beta\gamma}$ or the Ricci tensor $R_{\alpha\beta}$, via a series of derivatives $\partial$, covariant derivatives $\nabla := \partial + \Gamma$, and tensor index contractions $\mathscr{C} : \mathcal{V}^{r+1}_{s+1} \to \mathcal{V}^r_s$ (typically, $\mathrm{Tr}_g$).

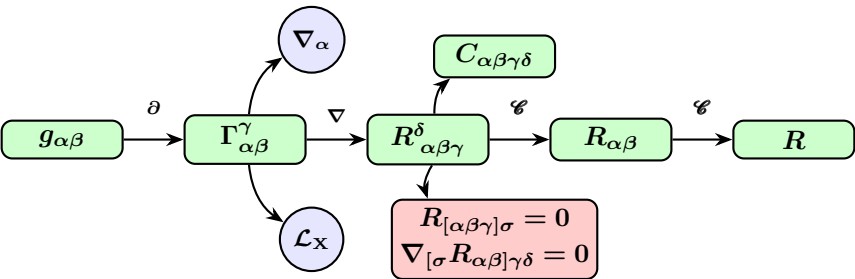

Figure 10: Differential geometry workflow in general relativity (only lhs of the EFEs – Eq. (2)): We describe each quantity starting left: The metric tensor $g_{\alpha\beta}$ defines the spacetime geometry. Its partial derivatives $\partial$ yield the Christoffel symbols $\Gamma^\gamma_{\alpha\beta}$, which describe the notion of parallel transport and defines a covariant derivative operation $\nabla_\alpha = \partial_\alpha + \Gamma_\alpha$. The connection also defines the Lie derivative $\mathcal{L}_v$ along vector fields $v$. The connection's derivatives $\nabla$ give the Riemann curvature tensor $R^\delta_{\alpha\beta\gamma}$, which encodes tidal forces. The contraction operator $\mathscr{C} = \mathrm{Tr}_g$ contracts with the metric, producing the trace part, i.e., the Ricci tensor $R_{\alpha\beta}$. Its subsequent contraction yields the Ricci scalar $R$. The Riemann tensor further splits into the Weyl tensor $C_{\alpha\beta\gamma\delta}$ (trace-free curvature part) and satisfies the algebraic and differential Bianchi identities $R_{[\alpha\beta\gamma]\sigma} = 0$ and $\nabla_{[\sigma} R_{\alpha\beta]\gamma\delta} = 0$. Together, these geometric objects form the backbone of general relativity, ultimately entering the Einstein field equations through the Einstein tensor $G_{\alpha\beta}$.

**Conservation laws.** It follows from the contracted *Bianchi identities*, i.e., cyclic sum of Riemann curvature tensor covariant derivatives (II Bianchi identity – see Eq. (59b)) vanishes identically:

$$\nabla_\alpha R_{\beta\gamma\delta\sigma} + \nabla_\beta R_{\gamma\alpha\delta\sigma} + \nabla_\gamma R_{\alpha\beta\delta\sigma} = 0 \ .$$

This is a geometric identity that holds for any (torsion-free) connection compatible with the metric. The identity above consequently leads to the covariant derivative of the stress-energy tensor vanishing, that is, $\nabla_\beta T^{\alpha\beta} := 0$ (see Eq. (66)), which corresponds to the energy-momentum conservation in general relativity. If required, conservation laws typically feature as soft constraints in PDEs, and are relevant especially when matter distribution/fields are considered.

**Equations of motion.** The *geodesic equation* is a central second-order ODE that describes the motion of objects following geodesic paths in the curved spacetime background

$$\frac{d^2 x^\mu}{d\tau^2} + \Gamma^\mu_{\rho\sigma} \frac{dx^\rho}{d\tau} \frac{dx^\sigma}{d\tau} = 0 \ .$$

Here, $\tau \in \mathbb{R}$ is some affine parameter (often the *proper time*; not to be confused with the coordinate time $t$). The geodesic equation is the general relativistic analogue of Newton's second law and generalizes the concept of a "straight line" to curved spacetime by describing the trajectories of bodies under the influence of a gravitational field. Related, the *geodesic deviation equation* describes how nearby geodesics diverge or converge due to the intrinsic curvature of the manifold, quantified by the *separation vector* $S^\mu(\tau)$:

$$\frac{D^2 S^\mu}{D\tau^2} = R^\mu_{\alpha\beta\gamma} X^\alpha X^\beta S^\gamma \ ,$$

where, $X^\alpha$ is a vector field and $\frac{D}{D\tau} = X^\alpha \nabla_\alpha$ denotes the directional covariant derivative (see Definition 2). Thus, it encodes information about the tidal forces of gravitation.

### D.0.1 ANALYTICAL (EXACT) SOLUTIONS

Exact solutions of the EFEs are metric tensor fields $g_{\alpha\beta}$ that satisfy Eq. (2). Many exact solutions are known, which can be classified into exterior (vacuum) solutions and interior solutions (Misner et al., 2017). While there exist several geometries that satisfy the EFEs, we shall focus on three prominent vacuum solutions: the Schwarzschild metric, the Kerr metric, and gravitational waves. These geometries not only have a high theoretical and historical relevance, but are also of great interest in computational black hole astrophysics and gravitational wave and multi-messenger astronomy. Appendix B discusses these solutions in more detail, including other prominent coordinates, as well as real and fake (coordinate) singularities. From here on, we work in naturalized units by setting $c = G = 1$.

**Schwarzschild metric** It is the simplest non-trivial solution to the EFEs and describes a static spherically symmetric geometry around spherical non-rotating gravitating bodies, such as stars or black-holes. Although simple, the Schwarzschild metric predicts many phenomena beyond Newtonian gravity, most notably the precession of elliptical orbits and the bending of light rays. Both of these predictions have been experimentally verified in the Solar system, using the motion of Mercury perihelion and in the Eddington experiment during the 1919 Solar eclipse, respectively. The metric is typically written in spherical polar coordinates $(t, r, \theta, \phi)$ where $t \in \mathbb{R}$, $r \in \mathbb{R}^+$, $\theta \in (0, \pi)$, and $\phi \in [0, 2\pi)$:

$$ds^2 = -\left(1 - \frac{2M}{r}\right)dt^2 + \left(1 - \frac{2M}{r}\right)^{-1} dr^2 + r^2\left(d\theta^2 + \sin^2\theta d\phi^2\right) \ .$$

**Kerr metric** generalizes the Schwarzschild solution to rotating bodies with the angular momentum $J$ or, equivalently, the rotation parameter $a = \frac{J}{M}$. The solution forms a rotating, stationary (but not static) geometry, which is oblate around the rotation axis that breaks spherical symmetry. This geometry again permits new phenomena, notably the geodetic effect and frame dragging, both of which have been experimentally verified in the Earth's orbit by the Gravity Probe B.

The metric can be described in the corresponding *oblate spheroidal coordinates* also known as the *Boyer-Lidquist* (BL) coordinates $(t, r, \vartheta, \phi)$ – see Eq. (78) (Boyer & Lindquist, 1967):

$$ds^2 = -\left(1 - \frac{2Mr}{\Sigma}\right)dt^2 - \frac{4Mar\sin^2\vartheta}{\Sigma}dtd\phi + \frac{\Sigma}{\Delta}dr^2 + \Sigma d\vartheta^2 + \left(r^2 + a^2 + \frac{2Mra^2\sin^2\vartheta}{\Sigma}\right)\sin^2\vartheta d\phi^2.$$

**Linearized gravity**   models the metric as tiny fluctuations or *perturbations* $h_{\alpha\beta}, |h_{\alpha\beta}| \ll 1$ of the flat background metric $\eta_{\alpha\beta}$:

$$g_{\alpha\beta} \approx \eta_{\alpha\beta} + h_{\alpha\beta} + \mathcal{O}(h_{\alpha\beta})^2 \,.$$

Famously, this model can describe GW propagation using a periodic time-dependent perturbation, which has served as the theoretical basis for the Nobel-prize winning detection of GWs generated by binary black hole mergers (Abbott et al., 2016c). As detailed in Appendix B.3, the choice of a certain *gauge* essentially transforms the vacuum EFEs into the wave equation $\Box \overline{h}_{\alpha\beta}^{(\epsilon)} = 0$ where $\Box \equiv \eta^{\alpha\beta}\partial_\alpha\partial_\beta$ is the *d'Alembert* or *wave* operator and $\overline{h}_{\alpha\beta}^{(\epsilon)} = h_{\alpha\beta}^{(\epsilon)} - \frac{1}{2}h^{(\epsilon)}\eta_{\alpha\beta}$. This equation admits a family of GW solutions: we will use the plane wave propagating in the $z$-direction with the angular frequency $\omega$ expressed in the Cartesian coordinates:

$$h_{\alpha\beta}^{\text{TT}} = \begin{pmatrix} 0 & 0 & 0 & 0 \\ 0 & h_+ & h_\times & 0 \\ 0 & h_\times & h_+ & 0 \\ 0 & 0 & 0 & 0 \end{pmatrix} \cos\left(\omega(t-z)\right) \,.$$

Here, $h_+$ and $h_\times$ are the amplitudes of the "+" (plus) polarization and "×" (cross) polarization

# E   RELATED WORK CONTINUED

**Compression techniques in scientific computing.**   classical compression strategies have been a versatile tool in reducing data sizes of large-scale numerical simulation data, which constitute storage memory-intensive domain. Simulation runs can range between petabytes to exabytes of data, thus requiring compression strategies to be integrated within the simulation pipeline, for efficient data volume reduction. These range from lossless and lossy compression techiques (Lindstrom & Isenburg, 2006; Di & Cappello, 2016; ISO Central Secretary, 2024) for multidimensional weather modeling tasks (Huang et al., 2016; 2025), discrete wavelet tranform (DWT)–based compression (Rho et al., 2023) or multidimensional data via tensor decomposition (Ballester-Ripoll et al., 2020).

Recently increasing works have leveraged implicit neural representations for a lossy *neural compression* (Dupont et al., 2021) by embedding high-dimensional (explicit grids), time-dependent physical simulations into compact, differentiable network weights. Such representations achieve compression factors of several orders of magnitude while retaining physical accuracy and offering efficient gradient access for downstream analysis or control. Applications include multidimensional weather and climate modeling (Huang & Hoefler, 2023) or even hybrid compression techniques using turbulent plasma–based simulations (Galletti et al., 2025).

**ML applied to gravitational physics.**   Gravitational wave modeling and numerical relativity problems have been tackled by state-of-the-art deep learning methods. These include DINGO – a rapid gravitational wave parameter estimation toolkit using NNs as surrogates for Bayesian posterior distributions (Dax et al., 2021). They show orders of magnitude reduction in inference time, bringing it down from $\mathcal{O}(\text{day})$ to $20s$. Similar lines of work include DINGO-BNS that performs real-time inference for binary neutron star (BNS) mergers and applicable for multi-messenger astronomy (Dax et al., 2025). On the other hand, physics informed neural networks (PINNs) (Raissi et al., 2019), have found applications in general relativistic phenomena and NR, such as solving the Tekoulsky equation inorder to compute the first quasinormal modes of the blackholes, for e.g., Kerr geometry (Luna et al., 2023; Cornell et al., 2022). Some other works explore physics informed neural operators (PINO) (Rosofsky & Huerta, 2023) for magnetohydrodynamics, or solving vaccum Einstein equations (Hirst et al., 2025)

**Physics-informed neural networks, neural operators and neural fields.** Physics-Informed Neural Networks (PINNs) augment neural network training with physical constraints derived from governing differential equations (Karniadakis et al., 2021). This is typically achieved by adding loss terms that

penalize violations of the residuals of the underlying PDEs or ODEs (Raissi et al., 2019), together with constraints from boundary conditions and conservation laws. PINNs are primarily data-free approaches and have been especially effective for solving forward and inverse problems. While PINNs share with neural fields the use of coordinate-based neural networks, their purpose is fundamentally different: PINNs *solve* a physical equation by enforcing its residual during optimization, whereas neural fields *represent* a given physical field directly from data without solving the governing equations. In this sense PINNs can be considered as a special case of neural fields (Xie et al., 2021), that satisfy the govering physical equations at each step.

Neural operators, in contrast, aim to learn mappings between infinite-dimensional function (Banach) spaces (Kovachki et al., 2023), providing a data-driven approach to approximate solution operators for entire families of PDEs. They prioritize generalization across different input functions, geometries, or forcing conditions, typically requiring large training sets covering many PDE instances.

Implicit neural representations (Müller et al., 2022) occupy a distinct position relative to both paradigms. Rather than enforcing a PDE residual (as in PINNs) or learning an operator over families of solutions (as in neural operators), neural fields provide a compact, continuous, and fully differentiable representation of a *single* high-dimensional physical field. This makes them especially well suited for encoding scientific data with high fidelity, enabling continuous spatial (and temporal) query access, implicit compression, and differentiable downstream analysis. In this sense, neural fields are not a competing method for PDE solution or operator learning, but a complementary representation framework for capturing and reconstructing complex physical domains.nstructing the dynamics with high fidelity, setting them apart from PINNs and neural operators in general.

## F EXPERIMENTAL DETAILS

This appendix provides detailed experimental specifics, including: (i) AD as a superior differentiation framework for tensor fields compared to higher-order finite-differencing methods; (ii) Effectivity of higher derivative losses for retrieving high-precision dynamics and higher-order curvature tensors & invariants; (iii) setup – data preparation (iv) gradient alignment aspects relevant to SOAP optimizers; (v) Component-wise tomography of compression error on the implicit metric and its derived higher-rank differential geometric quantities; (vi) training across varied coordinate systems to illustrate the coordinate-choice flexibility of NeFs; (vii) hyperparameter configurations employed; and (viii) the hardware and software environments used for these experiments.

### F.1 EVALUATION CRITERIA

We flatten the ground truth tensor at point $p \in \mathcal{M}$ with its components indexed by $k$ be denoted by $f_k(p) \in \mathbb{R}^n$, with $1 \leq k \leq n$ and the corresponding `EinFields` *parametrized* tensors are denoted by $\hat{f}_k(p)$. The dimensionality $n$ depends on the tensor under consideration. For instance, for a symmetric metric tensor $n = 10$, corresponding to its independent components, while for the Riemann curvature tensor, $n = 256$ when considering all components explicitly, or $n = 20$ when accounting only for the independent components under the symmetries inherent to the tensor, respectively.

We evaluated these quantities over a set of $m \approx 125,000$ validation collocation points $\mathcal{D} = \{p_i\}_{1 \leq i \leq m}$ and use standard error criteria in discretized form, which includes double sums: one over the total number of tensor components $\{f_k\}_{1 \leq k \leq n}$, while the other for the total number of collocation points $\{p_i\}_{1 \leq i \leq m}$:

$$\text{Mean-absolute error (MAE)} = \frac{1}{mn} \sum_{i=1}^{m} \sum_{k=1}^{n} |\hat{f}_k(p_i) - f_k(p_i)| \tag{104a}$$

$$\text{Relative } \ell_2 \text{ error (Rel. } \ell_2) = \sqrt{\frac{\sum_{i=1}^{m} \sum_{k=1}^{n} |\hat{f}_k(p_i) - f_k(p_i)|^2}{\sum_{i=1}^{m} \sum_{k=1}^{n} |f_k(p_i)|^2}} . \tag{104b}$$

These are applied to the metric tenors and their derived quantities, illustrated in Figures 10 and 2. Recall that the tensor components are coordinate-dependent (and even more so, the metric Jacobian,

metric Hessian, and Christoffel symbols are not even tensors), and, hence, these errors lack an immediate physical meaning. This is improved with the consideration of scalar quantities such as the Ricci scalar, Kretschmann invariants, and Weyl scalars, which by definition are coordinate-independent quantities.

## F.2 DATA GENERATION

Our use cases are exact analytic solutions to the EFEs, i.e., the set of metrics $g_{\alpha\beta}$ that satisfy the Eq. (2). These solutions describe the exterior (vacuum) solutions around massive gravitating objects. For our main set of experiments, we fit a NeF against the analytic solutions introduced in Section D.0.1, each having different features and spatio-temporal symmetries:

- Schwarzschild metric in spherical coordinates – Eq. (98),
- Kerr metric in Boyer-Lindquist and Kerr-Schild coordinates – Eqs. (101, 81),
- gravitational waves metric (TT gauge) in Cartesian coordinates – Eq. (110).

For each, we compute the distortion after subtracting the flat background metric in that particular coordinate chart. Detailed information on data specifications is provided in Table 7.

Additionally, we train each geometry in different coordinate systems to investigate how the choice of coordinates impacts NeFs (recall: the physical laws do not depend on the coordinate system).

Table 7: Training data generation specifications: spacetime metric, coordinate system, domain extent, grid resolution, and physical parameters.

| Metric | Coordinates | Domain | Resolution | Parameters |
|---|---|---|---|---|
| Schwarzschild | Spherical $(t, r, \theta, \phi)$ | $t = 0$ $r \in [2.5, 150]$ $\theta \in (0, \pi)$ $\phi \in [0, 2\pi)$ | 1 128 128 128 | M = 1 |
| Kerr | Boyer-Lindquist $(t, r, \vartheta, \phi)$ | $t = 0$ $r \in [3, 14]$ $\vartheta \in (0, \pi)$ $\phi \in [0, 2\pi)$ | 1 128 128 128 | M = 1 $a \in [0.628, 0.95]$ |
| | Kerr-Schild $(t, x, y, z)$ | $t = 0$ $x \in [-3, 3]$ $y \in [-3, 3]$ $z \in [0.1, 3]$ | 1 128 128 128 | M = 1 $a = 0.7$ |
| Linearized gravity | Cartesian $(t, x, y, z)$ | $t \in [0, 10]$ $x \in [0, 10]$ $y \in [0, 10]$ $z \in [0, 10]$ | 140 10 10 140 | $\omega = 1$ $\epsilon = 10^{-6}$ |

## F.3 COMPARING AD VS FD BASED METHODS.

We quantify the performance of automatic-differentiation operations on the ground truth metric against the 6-th order finite difference stencils. We test it against the Kretschmann scalar $\mathscr{K} = R_{\alpha\beta\gamma\delta} R^{\alpha\beta\gamma\delta}$, which is prone to errors, especially due to floating point errors accumulated in the Riemann curvature tensor:

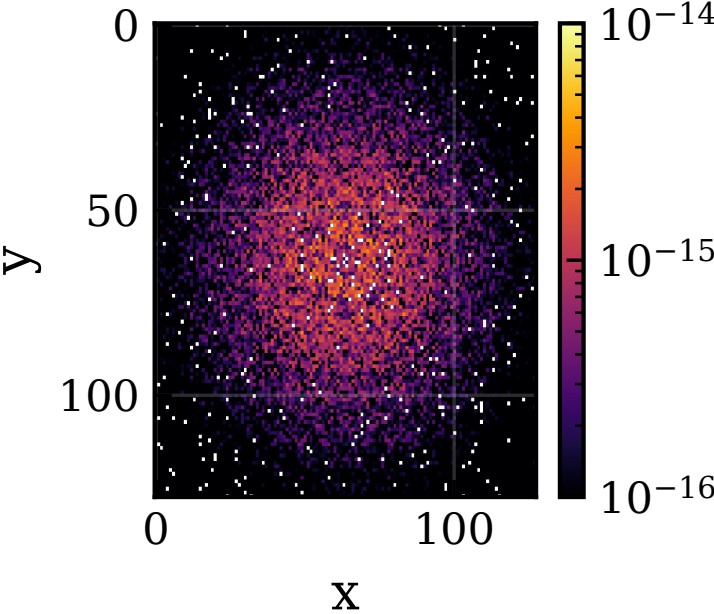

Figure 11: Absolute error $|\mathscr{K}_{\text{analytic}} - \mathscr{K}_{\text{AD}}|$ profile plotted for $z = 0.3$ between the analytic Kretschmann scalar and the ground truth Kretschmann scalar obtained via AD implemented on the ground truth (analytic) metric.

## F.4 HIGHER TENSOR DERIVATIVE LOSSES – SOBOLEV TRAINING

**Sobolev training** (Czarnecki et al., 2017) refers to a class of learning paradigms where NNs are trained not only to match target function values but also additionally its derivatives. Formally, given a target function $f : \mathcal{X} \to \mathbb{R}$, and a NN approximation $\hat{f}_\theta$, Sobolev training minimizes a joint loss involving the functional and its derivatives:

$$\mathcal{L}_{\text{Sob}}(\theta) = \mathbb{E}_x \left[ \lambda_0 \|f(x) - \hat{f}_\theta(x)\|^2 + \sum_{j=1}^{N} \lambda_j \left\| D^{(j)} f(x) - D^{(j)} \hat{f}_\theta(x) \right\|^2 \right], \quad (105)$$

where $D^{(j)}$ denotes the $j^{\text{th}}$ derivative operator, which in our case could be the partial derivatives $\partial^j$ or covariant derivatives $\nabla^j$, and $\lambda_j$ are weighting coefficients. This loss promotes alignment not only in function space but also in the Sobolev space $W^{N,2}(\mathcal{X})$, which encodes both value and derivative information. Sobolev training enhances generalization, stability, and accuracy of NeF derivatives (Chetan et al., 2024).

---

**Algorithm 1:** `EinField` training scheme

---

1: **Input:** Training dataset $\{(x^i, g(x^i), \partial_x^{(1)} g(x^i), \partial_x^{(2)} g(x^i))\}_{i=1}^m$, number of epochs $N_{\text{epochs}}$, learning rate $\eta$, optimizer $\mathcal{O}$, Sobolev order $N \in \{0, 1, 2\}$
2: Initialize neural field parameters $\theta$ on device $\mathcal{D}$ (e.g., GPU) in single (`FLOAT32`) precision
3: **for** epoch $= 1$ to $N_{\text{epochs}}$ **do**
4:      **for** each mini-batch $(x_{\text{batch}}, g_{\text{batch}}, \partial_x^{(1)} g_{\text{batch}}, \partial_x^{(2)} g_{\text{batch}})$ in dataset **do**
5:          Move $(x_{\text{batch}}, g_{\text{batch}})$ to device $\mathcal{D}$
6:          $\hat{g}_{\text{batch}} \leftarrow$ `EinFields`$(x_{\text{batch}}; \theta)$
7:          loss $\leftarrow \mathcal{L}(g_{\text{batch}}, \hat{g}_{\text{batch}})$
8:          **if** $N \geq 1$ **then** {Jacobian supervision}
9:              Compute $\partial_x^{(1)} \hat{g}_{\text{batch}}$ through AD
10:              loss $\leftarrow$ loss $+ \lambda_1 \cdot \mathcal{L}(\partial_x^{(1)} g_{\text{batch}}, \partial_x^{(1)} \hat{g}_{\text{batch}})$
11:          **end if**
12:          **if** $N \geq 2$ **then** {Hessian supervision}
13:              Compute $\partial_x^{(2)} \hat{g}_{\text{batch}}$ through AD
14:              loss $\leftarrow$ loss $+ \lambda_2 \cdot \mathcal{L}(\partial_x^{(2)} g_{\text{batch}}, \partial_x^{(2)} \hat{g}_{\text{batch}})$
15:          **end if**
16:          Compute gradients: $\partial_\theta \leftarrow \partial_\theta$ loss
17:          Update parameters: $\theta \leftarrow \mathcal{O}(\theta, \partial_\theta, \eta)$
18:          Optionally: synchronize gradients across devices if using distributed training
19:      **end for**
20:      Optionally: evaluate on validation set, log MAE and memory usage for monitoring
21:      Optionally: checkpoint $\theta$ for fault tolerance and reproducibility
22: **end for**
23: **return** optimized parameters $\theta$

---

The expected losses $\mathcal{L}(\partial_x^{(j)} g, \partial_x^{(j)} \hat{g})$ put-forth in Algorithm 1 is a short-hand notation for $\mathbb{E}_x \left\| \partial_x^{(j)} g_{\alpha\beta}(x) - \partial_x^{(j)} \hat{g}_{\alpha\beta}(x) \right\|^2$.

## F.5 GRADIENT ALIGNMENT

Competing tasks is a well-known problem in multi-objective learning (Yu et al. (2020), Liu et al. (2021), Shi et al. (2023)), the gradients of the loss functions pull the weights in different directions. In Scientific Machine Learning (SciML), a lot of work emerged in analyzing and mitigating gradient conflicts in the context of PINNs (Wang et al. (2025a), Lui et al. (2025), Hwang & Lim (2025)). Although Sobolev training differs from PINNs, particularly from a supervision perspective, it exhibits the same problem where some loss terms dominate others. In PINNs, the training is highly dependent on first satisfying the initial/boundary conditions, which provide uniqueness to the solution. The different levels of complexity between these and the residual loss create different optimization priorities, but both losses are equally important. Similarly, Sobolev training faces analogous challenges with competing loss components. The Jacobian data serve to constrain the model's derivatives, while the target function outputs determine the integration "constant", both components being equally valuable. However, the sources of complexity differ between these approaches. In PINNs, the primary challenge stems from determining a solution through unsupervised learning on PDE losses, whereas in our Sobolev training specifically, the complexity arises from managing optimization stability in high-dimensional spaces: a 16-dimensional output space, 64-dimensional Jacobian, and 256-dimensional Hessian. Moreover, this complexity is accompanied by the challenge of handling gradient imbalances. Depending on the point in spacetime, the metric or its derivatives dominate in the loss. Generally speaking, an analogy is to think $g_{\mu\nu} \propto \frac{1}{r}, \partial_\rho g_{\mu\nu} \propto \frac{1}{r^2}$ and $\partial_{\sigma\rho} g_{\mu\nu} \propto \frac{1}{r^3}$, making it clear how gradient magnitudes differ depending on the radius.

Mitigating gradient conflicts does not necessarily result in better accuracy, but it explains a possible reason why the loss does not improve further, the optimization being stuck in the local minimum of one of the objectives. The intra-step gradient alignment scores presented in Wang et al. (2025a) demonstrate SOAP as a far superior alternative compared to other well-established optimizers, or at least for the experiments considered in that study. To provide a direct comparison using the same methodology as in Wang et al. (2025a), we evaluated both ADAM and SOAP on the Cartesian KS representation of the Kerr metric, chosen as the most complex metric investigated in this work. The Sobolev training contains only two objectives: metric and Jacobian supervision. For our experimental setup, we employed an MLP architecture with 5 hidden layers, 190 hidden units per layer, and SILU activation function to compare gradient conflicts between the two optimizers. The training utilized a cosine decay learning rate schedule, starting from an initial learning rate of 1E−2 and decaying to a final rate of 1e−8 over 200 epochs. For weighting the losses, gradient normalization was used with and without an exponential moving average. As shown in Figure 15, even though Adam is providing twice as much gradient alignment score in almost all epochs, SOAP's second-order and preconditioning capabilities allow for a 100x training loss improvement.

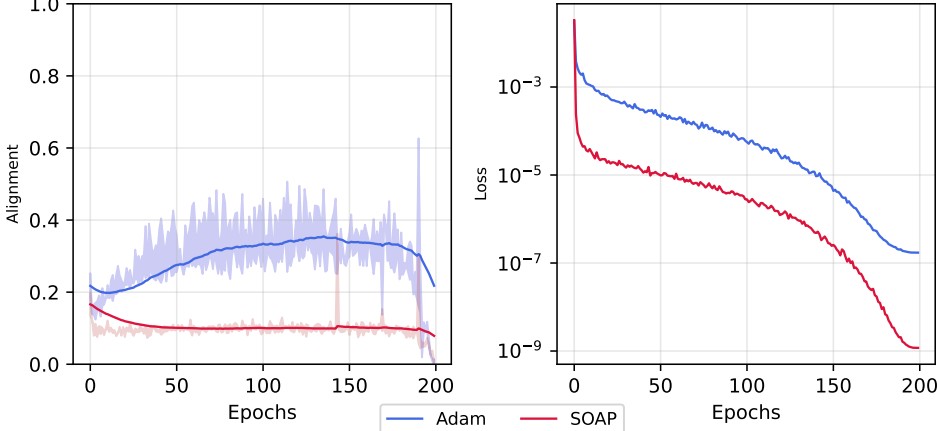

Figure 12: Average gradient alignment per epoch (left) and MSE loss during training for Adam and Soap optimizers (right). The shaded light color in the alignment plot represents a minimum and maximum deviation compared to using an exponential moving average or not for the weights multiplying the gradients.

## F.6 RECONSTRUCTING GENERAL RELATIVISTIC DYNAMICS AND CURVATURE SCALARS

We use synthetic data generated from analytical solutions to validate and characterize `EinFields`. We primarily focus on two interesting aspects of general relativity: (i) general relativistic dynamics, particularly geodesic motion around massive gravitating objects (see Section D) and (ii) global curvature structures encoded in tensorial invariants.

### F.6.1 FLOATING POINT PRECISION CAVEAT.

NR simulations are inherently high-precision endeavors, with the accurate modeling of complex gravitational phenomena critically reliant on high-fidelity numerical computations. In contrast to traditional machine learning domains, such as large language models (LLMs), where reduced-precision arithmetic (`FLOAT16` or `BFLOAT16`) yields strong results in both training accuracy and memory efficiency (Dean et al., 2012), this paradigm does not extend to NR workflows, where floating-point precision is a dominant factor influencing the fidelity of the results.

**`EinField`-based geodesics.** To compute geodesic motion, we numerically integrate the trajectories using a fifth-order explicit singly diagonally implicit Runge-Kutta (ESDIRK) solver. Specifically, we evolve Eq. (56) with respect to the affine parameter $\tau$ (proper time)[9], generating ground-truth geodesics from the analytic Christoffel symbols for the Schwarzschild and Kerr spacetimes. These are then compared against the rollouts obtained using the `EinField`-reconstructed Christoffel symbols.

To accurately retrace geodesic orbits, it is essential to incorporate Jacobian supervision within the Sobolev training framework. In contrast, additional Hessian supervision results in only marginal improvements for geodesic simulations and is not required in practice. Following Section F.6.1, all geodesic solvers are executed in double precision to ensure numerical stability and high-fidelity trajectory reconstruction.

Following the strategy presented in Section 3.2, we leverage `EinFields`' ability to yield high-precision derivatives of the spacetime metric, which includes Christoffel symbols, Riemann curvature tensors, and scalar invariants. In this section, we demonstrate how to accurately model particle trajectories derived from geodesic integration with our implicit parameterizations.

### F.6.2 SCHWARZSCHILD METRIC GEODESIC SETUP

Geodesics in the Schwarzschild spacetime are of fundamental interest, as they underlie phenomena such as gravitational lensing and the perihelion precession of Mercury, as well as the motion of planets in the solar system more generally.

The initial conditions of the trajectories chosen in the experiments are fully specified by the *initial position*

$$(t, r, \theta, \phi)(t = 0) = \big(0, a_0 r_s, \pi/2, 0\big) \tag{106}$$

and the *initial four-velocity*

$$(v^t, v^r, v^\theta, v^\phi)(t = 0) = \Big(\frac{1}{\sqrt{(1 - r_s/r_0)(1 - v_0^2)}}, 0, 0, v_0 \frac{\cos \phi_0}{\sqrt{r_0^2(1 - v_0^2)}}\Big), \tag{107}$$

Where $v_0 = b_0 \sqrt{1/(r_0 - r_s)}$ and $a_0, b_0 \in \mathbb{R}$ can be chosen freely to select the desired orbit in the $\theta = \pi/2$ plane. The geodesics in Figure 4 demonstrate a good qualitative agreement over several orbits. The error is quantified and discussed further in Section F.7.2.

### F.6.3 SCHWARZSCHILD BLACKHOLE RENDER

Being able to compute geodesics is sufficient to perform rendering. We use the Schwarzschild `EinFields` metric to render a black-hole on a celestial background. This requires propagating geodesics from the camera observer via the spacetime terminating at the distant background. The resulting ray-traced image as shown in the main text provides visual evidence for the global consistency and quality of the metric and the derived Christoffel symbols.

---

[9]Not to be confused with the coordinate time $t$.

### F.6.4 KERR METRIC GEODESIC SETUP

Geodesics in a Kerr spacetime around a rotating body (see details in Appendix B.2) play a central role in several key astrophysical observations and experimental tests of GR. Notably, photon geodesics determine the black hole shadow images captured by the Event Horizon Telescope (Fuerst, S. V. & Wu, K., 2004), and frame-dragging (Lense-Thirring) effects (Misner et al., 2017) are a hallmark of the Kerr geometry. These have been measured experimentally by the Gravity Probe B mission (Everitt et al., 2011) and recently via radio pulses arriving from pulsars (Krishnan et al., 2020).

While, the mathematical description of geodesics around Kerr metric is beyond scope for this work (see Teo (2003) for detailed exposition), we consider solving three different cases numerically. These include Zackiger orbits (retrograde geodesics – stable geodesics with larger radii), prograde orbits (stable geodesics with smaller radii), and arbitrary eccentric orbits, which depend on the initial conditions, including choice of energy $E$ and angular momentum $L_z$ of the test particle.

### F.6.5 KERR KRETSCHMANN INVARIANT

The Kretschmann invariant (scalar), $\mathscr{K} = R^{\alpha\beta\gamma\delta}(x^\mu)R_{\alpha\beta\gamma\delta}(x^\mu)$, is a key curvature invariant distinguishing true (curvature) singularities from coordinate (apparent) singularities. The nontrivial part of the Kretschmann invariant for the Kerr metric reads:

$$R_{\alpha\beta\gamma\delta}R^{\alpha\beta\gamma\delta} = C_{\alpha\beta\gamma\delta}C^{\alpha\beta\gamma\delta} = \frac{48M^2(r^2 - a^2\cos^2\vartheta)\left[(r^2 + a^2\cos^2\vartheta)^2 - 16r^2a^2\cos^2\vartheta\right]}{(r^2 + a^2\cos^2\vartheta)^6} .$$
(108)

This guarantees that the curvature singularity ($\mathscr{K} \to \infty$) occurs at the ring $\Sigma \equiv r^2 + a^2\cos^2\vartheta = 0$, with zeros at:

$$r = 0 \text{ , and, } \vartheta = \frac{\pi}{2} .$$

Thus, the rotation $a$ induces a ring singularity at radius $a$ on the equatorial plane $\theta = \pi/2$, where the curvature diverges. Accurately capturing this geometric structure requires isolating true singularities from coordinate artifacts, which can otherwise lead to incorrect classification of singularities.

We perform training in Cartesian KS coordinates (see Eq. (81)) to eliminate coordinate singularities that would otherwise impede convergence. We first train `EinFields` (+Jac + Hess) on Cartesian KS coordinates, subsequently constructing the Riemann tensor – see Section A.3.5.1 via successive automatic differentiation steps and raising indices using the parametrized metric $\hat{g}$ (see Eq. (108)). The NeF reconstructed $\hat{\mathscr{K}}$ Figure 13b captures the ring singularity structure and agrees well with the analytical solution, as shown in Figures 13a. However, the reconstruction remains sensitive to floating-point errors and requires high NeF accuracy for stability as seen from Figure 13c.

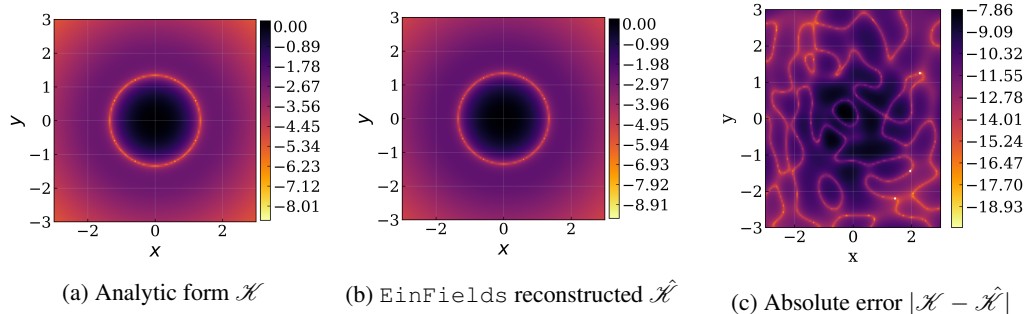

(a) Analytic form $\mathscr{K}$     (b) `EinFields` reconstructed $\hat{\mathscr{K}}$     (c) Absolute error $|\mathscr{K} - \hat{\mathscr{K}}|$

Figure 13: The Kretschmann scalar $\mathscr{K}$ of the Kerr metric computed in Cartesian Kerr-Schild form (Eq. (81)) in the $x$-$y$ plane for $z = 0.3$.

## F.7 TRAINING ON VARIED COORDINATE SYSTEMS

Table 8: Relative L2 error considered on a grid of validation collocation points (i) EinFields, (ii) EinFields (+Jac) and, (iii) EinFields (+Jac + Hess) supervision. As described above in the text, we quantify the effect of inputs queries in varied coordinate charts and how `EinFields` training generalizes over these different metric (geometry) representations.

| Metric | Representation | Coordinate | Rel. L2 |
|---|---|---|---|
| Schwarzschild | EinFields | Spherical | 2.26e-7 |
| | | Cartesian Kerr-Schild | 1.37e-5 |
| | | Eddington-Finkelstein | 9.21e-9 |
| | EinFields (+ Jac) | Spherical | 1.37e-7 |
| | | Cartesian Kerr-Schild | 3.00e-6 |
| | | Eddington-Finkelstein | 6.47e-9 |
| | EinFields (+ Jac + Hess) | Spherical | 1.20e-7 |
| | | Cartesian Kerr-Schild | 1.53e-6 |
| | | Eddington-Finkelstein | 9.08e-9 |
| Kerr | EinFields | Boyer-Lindquist | 6.95e-8 |
| | | Cartesian Kerr-Schild | 4.47e-6 |
| | | Eddington-Finkelstein | 6.44e-8 |
| | EinFields (+ Jac) | Boyer-Lindquist | 4.72e-8 |
| | | Cartesian Kerr-Schild | 8.83e-7 |
| | | Eddington-Finkelstein | 4.95e-8 |
| | EinFields (+ Jac + Hess) | Boyer-Lindquist | 4.69e-8 |
| | | Cartesian Kerr-Schild | 4.95e-7 |
| | | Eddington-Finkelstein | 4.72e-8 |

Results reported in Table 8 suggest that the choice of coordinates has a strong impact on the metric up to three orders of magnitude. This aspect should be investigated further in future work.

NeFs take physical coordinates as inputs and map them directly to field values. Unlike traditional machine learning architectures that ingest abstract learned feature spaces (such as token embeddings or extracted features), INRs operate directly on the physical coordinate space, enabling them to represent continuous signals in a domain-agnostic manner.

In the context of GR, this implies that a four-dimensional representation of metric tensor fields by an INR explicitly depends on the input coordinate system, or more generally, on the chosen frame of reference. Despite this apparent dependency, GR possesses the fundamental property of diffeomorphism covariance (see Section A.3.8), which asserts that the laws of gravitation remain invariant under smooth coordinate transformations. However, the choice of coordinate system remains an essential practical tool for simplifying the form of the metric tensor. For example, while the Schwarzschild metric is diagonal in spherical coordinates (albeit with a coordinate singularity at the event horizon), transforming to Cartesian KS coordinates produces a dense, off-diagonal metric representation, or, for that matter, moving to Eddington-Finkelstein coordinates, which both remove coordinate-related artifacts (see Paragraph A.3.5.2).

Understanding this behavior is essential for developing robust INR-based frameworks for representing geometric quantities in numerical relativity, while respecting the underlying diffeomorphism invariance of general relativity. For the Schwarzschild case, we initiate the training by sampling query spacetime coordinates in spherical representation $(t, r, \theta, \phi)$. These sampled collocation points are then transformed into their corresponding collocation points in Cartesian coordinates $(t, x, y, z)$ and ingoing Eddington-Finkelstein coordinates $(v, r, \theta, \tilde{\phi})$ (see Section B.1.1 for explicit transformation details). Subsequently, `EinFields` outputs the metric tensors corresponding to these coordinate systems, yielding Eqs. (70, 74, 77). For the Kerr metric, which is characterized by its oblate spheroidal geometry, we sample query collocation points in the Boyer-Lindquist coordinates $(t, r, \vartheta, \phi)$, followed by the collocation points transformed into Cartesian $(t, x, y, z)$ and ingoing

Eddington-Finkelstein coordinates $(t, r, \theta, \tilde{\phi})$. `EinFields` then outputs the Kerr metric tensors in these respective coordinate systems, resulting in Eqs. (80, 81, 83).

Table 9: Col. 1 lists the spacetime metrics (Schwarzschild and Kerr). Cols. 2–4 indicate the coordinate charts used for NeF training: spherical-like, Cartesian-like, and lightcone-like. For **Schwarzschild**, these correspond to spherical coordinates $(t, r, \theta, \phi)$, Cartesian Kerr-Schild (KS) coordinates $(t, x, y, z)$, and ingoing Eddington-Finkelstein (EF) coordinates $(v, r, \theta, \phi)$, trained on the metrics described in Eqs. (98–100) respectively. For **Kerr**, these correspond to Boyer-Lindquist (BL) coordinates $(t, r, \vartheta, \phi)$, Cartesian KS coordinates $(t, x, y, z)$, and ingoing EF coordinates $(v, r, \theta, \tilde{\phi})$, trained on the metrics in described in Eqs. (101–102) respectively.

| Metric | Spherical-like | Cartesian-like | Lightcone-like |
|---|:---:|:---:|:---:|
| Schwarzschild | ✓ | ✓ | ✓ |
| Kerr | ✓ | ✓ | ✓ |

This multi coordinate training strategy ensures that the neural tensor field learns consistent representations across coordinate systems while maintaining geometric and physical consistency under diffeomorphisms, facilitating generalization and stability in downstream geometric learning tasks.

### F.7.1 LINEARIZED GRAVITY: GEODESIC DEVIATION, GRAVITATIONAL-WAVE STRAINS AND WEYL SCALARS

Linearized gravity models the solution of the EFEs via periodic perturbations on a fixed background metric. These linearized solutions are highly relevant in numerical relativity, as they describe the groundbreaking, experimentally verified discovery of *gravitational waves* generated by binary black hole mergers (Abbott et al., 2016c). The metric tensor can be written as

$$g_{\alpha\beta} \approx \eta_{\alpha\beta} + h_{\alpha\beta} + \mathcal{O}(h_{\alpha\beta})^2 \,, \tag{109}$$

where $|h_{\alpha\beta}| \ll 1$ is the perturbation term. As detailed in Section B.3, a plane gravitational wave propagating in the $z$-direction with angular frequency $\omega$ can be described in the *tranverse-traceless* (TT) gauge as

$$h_{\alpha\beta}^{\text{TT}} = \begin{pmatrix} 0 & 0 & 0 & 0 \\ 0 & h_+ & h_\times & 0 \\ 0 & h_\times & h_+ & 0 \\ 0 & 0 & 0 & 0 \end{pmatrix} \cos\left(\omega(t-z)\right) \,. \tag{110}$$

Here, $h_+$ and $h_\times$ are the amplitudes of the "+" (plus) polarization and "×" (cross) polarization.

**Validation problems for GW metric and derivatives quality.** Compared to Schwarzschild and Kerr metrics, a key distinction of the linearized gravity setting describing gravitational waves is its time dependence – see Eq. (110). Although it does not depend on $x$ and $y$, the temporal dependence motivates us to consider our model trained on a full spacetime grid of size $\mathtt{N_t} \times \mathtt{N_x} \times \mathtt{N_y} \times \mathtt{N_z}$.

**Distortion of ring of test-particles.** When the described gravitational wave interacts with a ring of freely falling test particles initially at rest in the x-y plane, it induces periodic deformations of the ring. For a purely + polarized wave, the resulting motion causes the ring to stretch and squeeze along the x- and y-axes, leading to a characteristic "plus" deformation pattern.

The motion of the test particles under the influence of this gravitational wave is obtained by solving the geodesic deviation equation, up to leading order in the strain amplitude $h_+$. As a result, the particle trajectories in the TT gauge are

$$x(t) = \left(1 + \frac{1}{2}h_+ \cos\left(\omega(t-z)\right)\right)x(0) \,, \quad y(t) = \left(1 - \frac{1}{2}h_+ \cos\left(\omega(t-z)\right)\right)y(0) \,. \tag{111}$$

Here, $x(0)$ and $y(0)$ denote the initial coordinates of a test particle, and the time-dependent perturbations reflect the tidal nature of gravitational waves. The cosine dependence captures the periodic stretching and squeezing of spacetime caused by the wave as it traverses the particle ring. Figure 5 and Table 10 show how the famous ring oscillation experiment can be reproduced with `EinFields`. This is done by parametrizing the perturbation $h_{\alpha\beta}^{\text{TT}}$ and captures the famous *stretching and squeezing* effect.

**Weyl scalars of gravitational radiation field.** The *Weyl scalars* are five complex quantities $\Psi_0, \Psi_1, \Psi_2, \Psi_3, \Psi_4$ that arise in the *Newman–Penrose* formalism of GR (Newman & Penrose, 1962). They encode all the independent components of the Weyl tensor $C_{\alpha\beta\gamma\delta}$ (see Eq. (63)), representing the "free" gravitational field – the part of spacetime curvature that can propagate as gravitational waves, distinct from the curvature directly caused by matter. In NR and GW modeling, $\Psi_4(t)$ is the primary scalar quantity used to extract observable GW signals from simulations. It is defined as

$$\Psi_4 := C_{\alpha\beta\gamma\delta} n^\alpha \bar{k}^\beta n^\gamma \bar{k}^\delta \tag{112}$$

with $n, k$ being a particular choice of Newman–Penrose tetrads and $\bar{k}$ its complex conjugate [10]. The central relation in an asymptotically flat spacetime (cf. Boyle et al. (2019) for details) is that $\Psi_4(t)$ is equivalent to the second coordinate-time derivative of the strain $h(t) = h_+(t) + ih_\times(t)$:

$$\Psi_4 \equiv -\ddot{h}_+ + i\ddot{h}_\times \,. \tag{113}$$

We compute $\Psi_4$ from the NeF-parameterized strain $h_{\alpha\beta}$ in two distinct ways:

---

[10]Note that the Weyl scalars are not invariant and depend on a particular choice of the tetrad fields.

1. *indirectly* via the Weyl tensor obtained with the differential-geometric chain (see also Figures 10 and 2): $h_{\alpha\beta}^{\text{TT}} \xrightarrow{+\eta_{\alpha\beta}} g_{\alpha\beta} \xrightarrow{\partial} \Gamma_{\alpha\beta}^{\gamma} \xrightarrow{\nabla} R^{\delta}{}_{\alpha\beta\gamma} \to C_{\alpha\beta\gamma\delta} \xrightarrow{\text{Eq. (112)}} \Psi_4$;

2. *directly* via the second time-derivative: $h_{\alpha\beta}^{\text{TT}} \xrightarrow{\text{Eq. (113)}} \Psi_4$.

**Spin-weighted spherical harmonic representation for GW extraction.** A quantity of central interest in gravitational waveform construction is the *mode decomposition* of the GW strain into its angular components. The complex strain $h(t, r, \theta, \phi) \equiv h_+(t, r, \theta, \phi) - i\, h_\times(t, r, \theta, \phi)$ can be expanded in terms of *spin-weighted spherical harmonics* (SWSHs) as

$$h(t, r, \theta, \phi) = \frac{M}{r} \sum_{\ell=2}^{\infty} \sum_{m=-\ell}^{\ell} h^{\ell,m}(t) \,_{-2}Y_{\ell m}(\theta, \phi) \,, \tag{114}$$

where $_{-2}Y_{\ell m}(\theta, \phi)$ are the SWSHs (see Eq. (90)) with spin-weight $s = -2$ reflecting the helicity of GWs in the TT gauge (see Eqs. (88 and 89). In practice, the dominant contributions to the strain arise from the quadrupole ($\ell = 2, m = \pm 2$) modes, denoted by $h^{2,\pm2}(t)$, which capture the leading-order gravitational radiation (detailed in Section B.3).

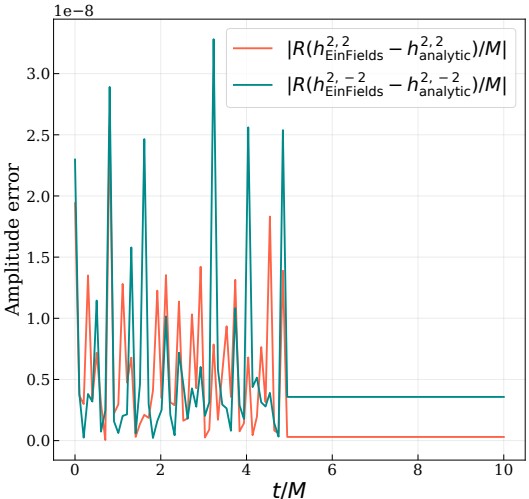

Figure 14: The absolute error of the amplitude between the `EinFields` and analytic values $|R/M h^{2,\pm2}(t)|$ (see Eq. (89)) at a fixed radial distance $R = 1$ plotted against $t/M$. The amplitudes agree to 1E-8, indicating that `EinFields` can capture the complex strain $h$ and subsequently $h^{2,\pm2}(t)$ GW signals.

**Radiated power of GWs.** Another important physical observable for GWs is the radiated power loss given by the famous *quadrupole* formula (Carroll et al., 2004). The time-averaged power or *luminosity* radiated by GWs is given by

$$\frac{\mathrm{d}E}{\mathrm{d}t} = \frac{r^2}{32\pi} \int \mathrm{d}\Omega \langle \dot{h}_{ij}^{TT} \dot{h}^{TT\ ij} \rangle = \frac{1}{4} \langle \dot{h}_+^2 + \dot{h}_\times^2 \rangle .\tag{115}$$

The particular perturbation metric in the above experiments (see Eq. (110)) has equal amplitude $A = h_+ = h_\times$ for both $+$ and $\times$ polarizations. As a consequence, the radiated power loss simplifies to

$$\frac{\mathrm{d}E}{\mathrm{d}t} = \frac{\omega^2 A^2}{4}.\tag{116}$$

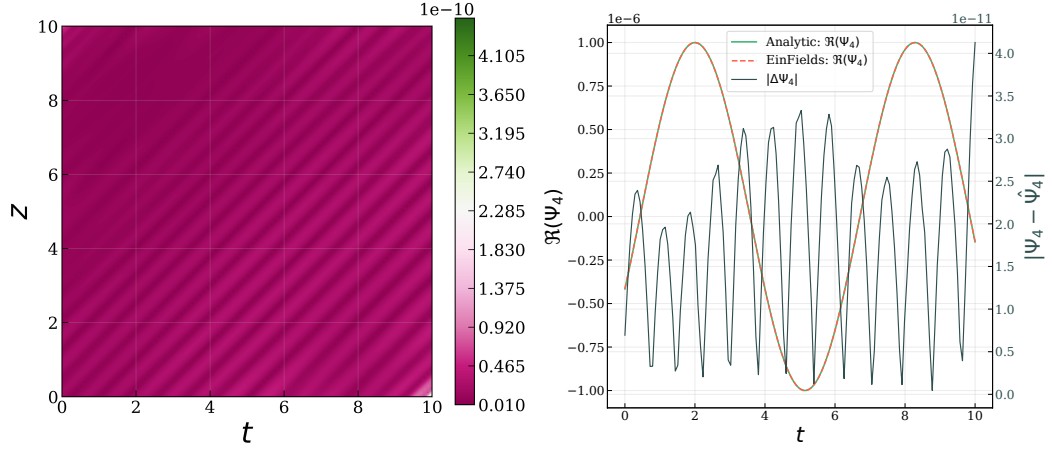

(a) Absolute error in the $(z, t)$ plane, averaged over $x$–$y$ slices.

(b) Temporal evolution of the Weyl scalar computed from the analytic and `EinFields` perturbations (left axis) and their absolute error (right axis) at a fixed position.

Figure 15: Comparison of the real part of the Weyl scalar $\Re(\Psi_4)$ (Eq. (112)) computed from the `EinFields` and the analytic metric. The errors are on the order of E-10 and E-11, respectively, indicating highly accurate gravitational waveform reconstruction capacities of `EinFields`.

Table 10: Rel. $\ell_2$ for key quantities in the linearized gravity case with two different NeF architectures: (i) perturbation metric, (ii) perturbation metric trained with Sobolev loss and gradient normalization, (iii) reconstructed Ricci scalar, and (iv) reconstructed real part of the Weyl scalar $\Psi_4$, where $\Psi_4 \propto \ddot{h}_+(t, \mathbf{x})$. The final column reports the absolute difference in the predicted gravitational radiation energy loss, integrated over the unit sphere.

| Model | $h_{\alpha\beta}^{TT}$ (+Jac + Hess) (GradNorm) | Ricci scalar $R$ | Weyl scalar $\Re(\Psi_4)$ | Luminosity $\mathrm{d}E/\mathrm{d}t$ |
|---|---|---|---|---|
| SiLU | 8.56e-4 | 5.90e-13 | 2.53e-5 | 2.71e-4 |
| SIREN | 3.78e-2 | 1.08e-12 | 9.56e-5 | 3.34e-4 |
| WIRE | 1.68e-2 | 1.55e-13 | 1.81e-5 | 3.69e-4 |

### F.7.2 ACCUMULATION OF ROLLOUT ERRORS FOR GEODESICS

Minute floating-point inaccuracies (around 1E-5 to 1E-6) arising from Christoffel symbols retrieved via `EinFields` autoregressively accumulate when evolving the equations of motion for test particles along geodesics.

To quantify the inaccuracies between the ground truth and NeF-evolved geodesics, we compute the deviation between the position vectors $r(\tau) \in \mathbb{R}^3$ as a function of the affine parameter (proper time) $\tau$ in Cartesian coordinates. Specifically, for the ground truth trajectory, the spatial coordinates corresponding to the position vector are given by $r(\tau) = \big(x(\tau), y(\tau), z(\tau)\big)$, while for the NeF-evolved trajectory, we denote $\hat{r}(\tau) = \big(\hat{x}(\tau), \hat{y}(\tau), \hat{z}(\tau)\big)$. The deviation at each proper time $\tau$ is then computed as the Euclidean norm,

$$\delta r(\tau) = \|r(\tau) - \hat{r}(\tau)\|_2 = \sqrt{\big(x(\tau) - \hat{x}(\tau)\big)^2 + \big(y(\tau) - \hat{y}(\tau)\big)^2 + \big(z(\tau) - \hat{z}(\tau)\big)^2}. \quad (117)$$

In practice, the geodesic trajectories are computed in $(r, \vartheta, \phi)$ (e.g., Boyer–Lindquist) coordinates and subsequently transformed into Cartesian coordinates before evaluating the deviation using the above expression.

While single-precision (`FLOAT32`) arithmetic is sufficient for training `EinFields` in most experiments and downstream tasks presented, geodesic simulations indicate the need for `FLOAT64` precision results in MAE and relative $\ell_2$ error for the reconstructed metric and its derivatives. Only `FLOAT64` ensures the mitigation of error accumulation during temporal rollout, preserving the accuracy necessary for reliable scientific inference in gravitational physics.

Given the high sensitivity of time-stepped trajectories to such numerical inaccuracies, we quantify this error accumulation by explicitly presenting the deviation as a function of the affine parameter $\tau$, especially for *eccentric orbits* for both Schwarzschild and Kerr metric for $a = 0.628$. These are reported for the Schwarzschild use case in Figure 16a, and Figure 16b for the Kerr metric use case, respectively. For Schwarzschild, the error accumulates stably, while for Kerr it is erratic. We hypothesize this is likely due to the stable versus chaotic nature of orbits in the respective spacetimes. Eventually, orbits diverge significantly, especially when leaving the NeF training domain.

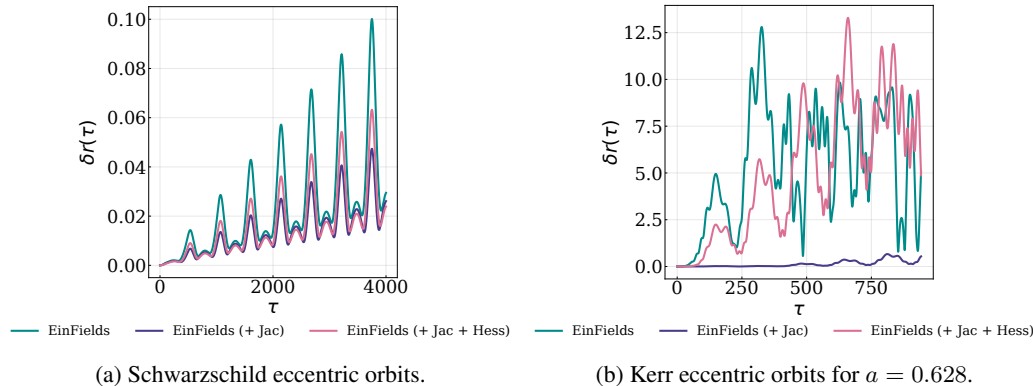

(a) Schwarzschild eccentric orbits.    (b) Kerr eccentric orbits for $a = 0.628$.

Figure 16: Geodesic rollout deviation $\delta r$ over proper time $\tau$.

The results suggest that incorporating the Hessian supervision into training may introduce noise that can hinder convergence, performing worse than using metric Jacobian supervision or, for that matter, metric alone. For geodesic equations, supervising second derivatives is often unnecessary, and Jacobians alone provide significant improvements in trajectory reconstruction. However, Hessians become essential when computing Riemann tensors and curvature-related quantities, and are required in applications such as numerically solving the geodesic deviation equation (see Eq. (60)), which are typically encountered for solving for the test ring oscillation in linearized gravity use cases.

F.8   TOMOGRAPHY: METRIC, METRIC JACOBIAN AND METRIC HESSIAN COMPONENTS

Here, we demonstrate the quality of `EinFields` parametrized metric tensor fields for the Kerr metric with spin parameter $a = 0.7$[11], we report the mean absolute error (MAE) between the ground truth and the NeF-fitted metric tensors in Figure 17. The evaluation is performed on a validation grid with collocation points sampled arbitrarily within the training range but distinct from the training collocation points. Using a model configured with SiLU activations, SOAP optimizer, GradNorm, and without Sobolev regularization, we observe agreement with the ground truth up to six decimal places, achieving an MAE on the order of $1E{-}6$. The effect of introducing losses pertaining to metric

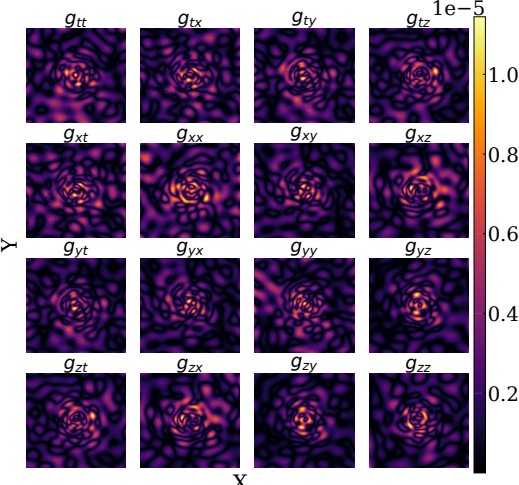

Figure 17: Kerr metric absolute error between ground truth (analytic) metric and the EinFields parametrized metric. The metrics are depicted in the Cartesian Kerr-Schild (KS) representation as presented in Eq. (81). The 2D slice of all the metric components captured in the x-y plane at fixed $z = 1.4$ for a spin parameter value $a = 0.7$.

Jacobian and Hessian supervision, apart from the metric loss that EinFields predominantly uses, can be quantified and visualized with the following plots below. Here, for the sake of visualization, we do a tomography (2D cuts) of different metric components along a particular axis for the Kerr metric in Cartesian KS coordinates (Eq. (81)).

The first, second, and third columns in each figure correspond to `EinFields` training without Sobolev supervision, `EinFields` (+Jac), and `EinFields` (+Jac + Hess) trained, respectively, for randomly sampled components of differential geometric quantities.

---

[11]Cartesian Kerr-Schild coordinates are chosen to avoid coordinate singularities, enabling tomography over larger coordinate ranges.

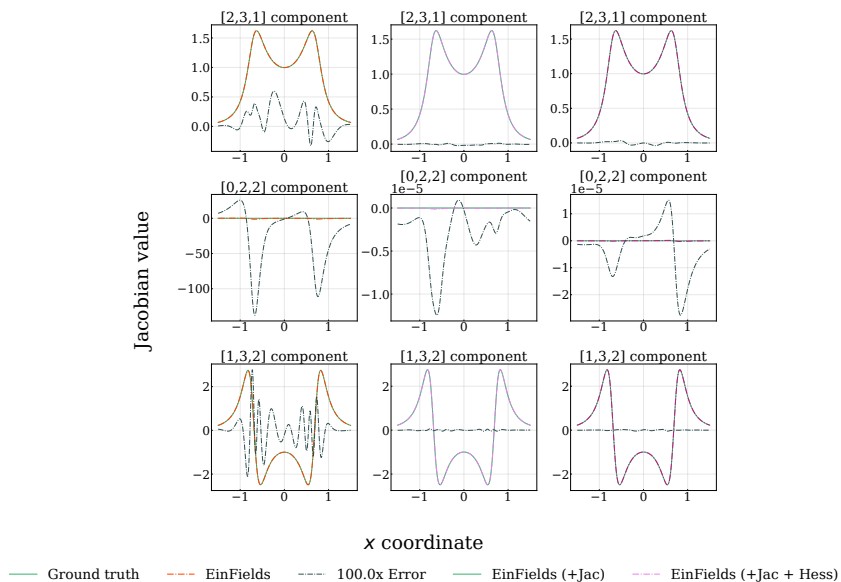

Figure 18: 2D Tomography of Kerr *metric Jacobian* components in Cartesian KS representation.

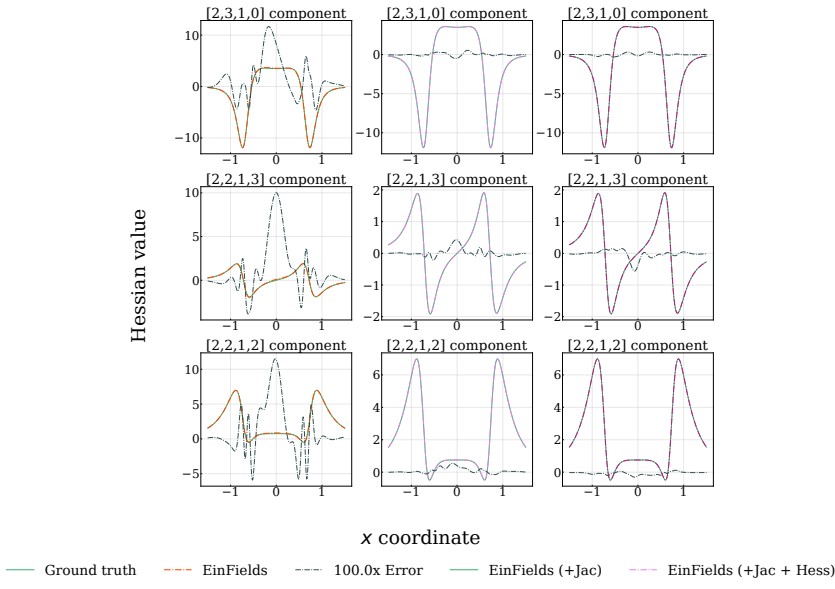

Figure 19: 2D Tomography of Kerr *metric Hessian* components in Cartesian KS representation.

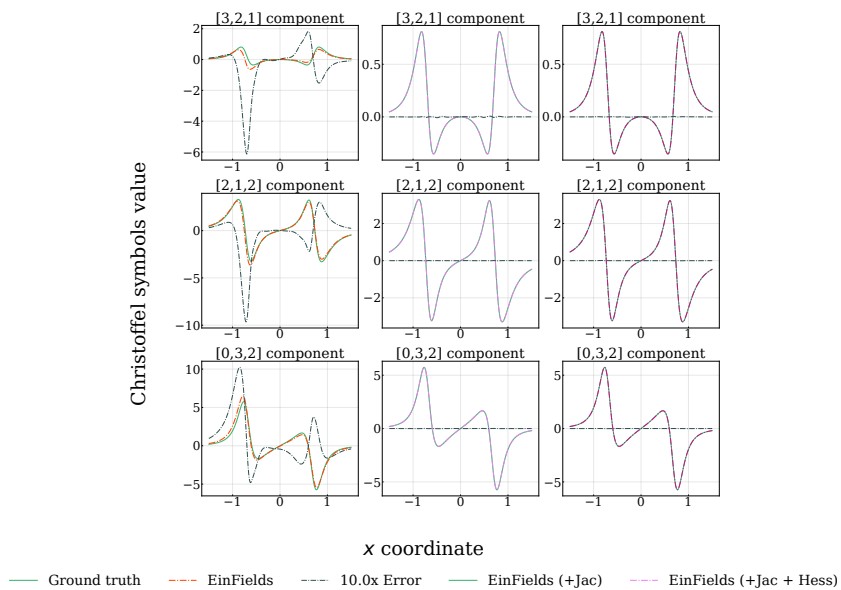

Figure 20: 2D Tomography of Kerr *Christoffel symbols* components in Cartesian KS representation.

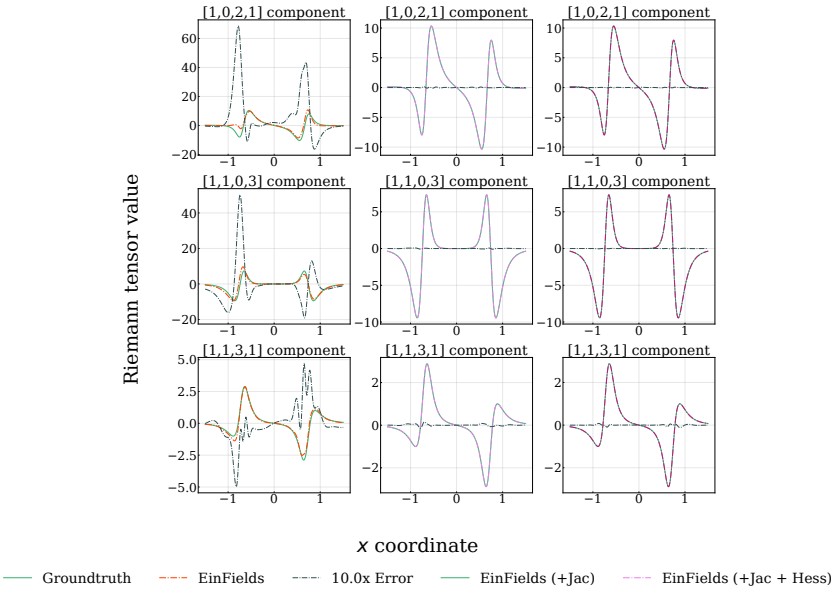

Figure 21: 2D Tomography of Kerr *Riemann tensor* components in Cartesian KS representation.

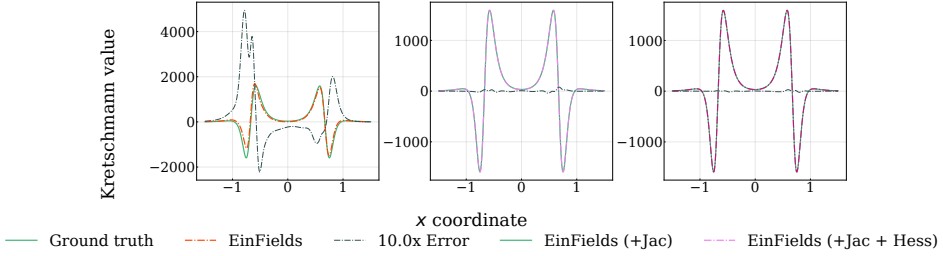

Figure 22: 2D Tomography of Kerr *Kretschmann invariant* in Cartesian KS representation.

### F.9 Training hyperparameters

Table 11: Training configurations for Schwarzschild, Kerr, and GWs used in the geodesics, Kretschmann plots, Table 8, and linearized gravity section.

| Parameter | Schwarzschild | Kerr | GWs |
|---|---|---|---|
| **Architecture** | | MLP | |
| Depth | 3 / 3 / 5 / 7 | 5 | 5 |
| Width | 64 / 128 / 256 / 512 | 190 | 128 / 128 / 90 |
| Activation | SiLU | SiLU | SiLU / SIREN / WIRE |
| Input dimension | 4 | 4 | 4 |
| Output dimension | 10 / 16 | 16 | 16 |
| # Parameters | 13.5K / 50K / 332K / 1.5M | 185K | 85K |
| **Optimizer** | | SOAP | |
| $\beta_1$ | | 0.95 | |
| $\beta_2$ | | 0.95 | |
| Precondition frequency | | 1 | |
| **Learning rate schedule** | | | |
| Initial learning rate | | E-2 / E-3 | |
| Decay steps | $10^4$ | $6 \times 10^3 / 2, 4, 6 \times 10^4$ | $4 \times 10^3 / 4 \times 10^4$ |
| Final learning rate | E-5 / E-6 | E-7 / E-8 | E-9 |
| **Training** | | | |
| Epochs | 100 | 200 | 200 |
| Number of batches | 100 | 30 / 100 / 200 / 300 | 20 / 200 |
| Gradient weighting scheme | | None / GradNorm | |

### F.10 Hardware & Licenses

For our primary computational work, we utilize a high-performance CPU system equipped with 2×32-core Intel® Xeon® Platinum 8452Y+ processors, each operating at 4.1 GHz, and 2048 `GiB` of RAM. All NeF-related training is performed on a single NVIDIA H200 SXM GPU with 144 `GiB` of HBM3e memory. For prototyping and preliminary experiments, we employ a single NVIDIA Tesla A100 GPU with 40 `GiB` of memory.

This work would not have been possible without the open-source software ecosystem. Our implementation is built upon multiple community-maintained libraries, and we gratefully acknowledge their licenses below. The core computations were performed using `JAX[cuda12]` (Bradbury et al., 2018) with `CUDA` support, licensed under the Apache 2.0 License. For model definition and training, we relied on `Equinox` (Kidger & Garcia, 2021) and `Flax.Linen` (Heek et al., 2024), both also under Apache 2.0. For solving differential equations, we employed `Diffrax` (Kidger, 2021), distributed under the Apache 2.0 License. All libraries used are permissively licensed, enabling free academic and non-commercial research.

## G ADDITIONAL EXPERIMENTS

### G.1 Oscillating neutron star NR simulation Specifications

**Simulator setup and solver.** The system is evolved using the Baumgarte–Shapiro–Shibata–Nakamura (BSSN) formulation of Einstein's equations (Shibata & Nakamura, 1995; Baumgarte & Shapiro, 1998) as implemented in the `McLachlan` code, together with general-relativistic hydrodynamics provided by `GRHydro` of EinsteinToolkit (Löffler et al., 2012). Reconstructing the full 4D spacetime metric $g_{\alpha\beta}$ around the neutron star from the numerical BSSN solver output variables is done by collecting the ADM variables (Arnowitt et al., 1959) (via

the `ADMBase Thorn` in EinsteinToolkit) in the following manner:

$$g_{\mu\nu} = \begin{pmatrix} -\alpha^2 + \beta_i\beta^i & \beta_i \\ \beta_i & \gamma_{ij} \end{pmatrix}. \tag{118}$$

where $\alpha$ corresponds to the *lapse* function, $\beta_i := \{\beta_x, \beta_y, \beta_z\}$ is the shift-vector and the 3-metric $\gamma_{ij} := \{\gamma_{xx}, \gamma_{xy}, \cdots, \gamma_{zz}\}$. The simulation employs *fixed mesh refinement* (FMR) (Schnetter et al., 2004), i.e., grid patches of non-uniform resolution. The coarsest grid usually encloses the whole simulation domain. Successively finer grids overlay the coarse grid at those locations where a higher resolutions is needed. This is done by the `Carpet` infrastructure (Löffler et al., 2012). Four distinct, static grid configurations are used, namely *low*, *medium*, *high*, and *highest* resolutions, which differ only by their number of collocation points per refinement level. FMR requires far less resources than globally increasing the resolution. No dynamic or adaptive regridding is performed during the evolution; instead, each run is performed independently at a fixed resolution to assess convergence and scalability. into a unigrid application with minimal changes to its structure. Instead of only one grid, there are several grids or grid patches with different resolutions[12].

**Training data generation.** The training data are obtained by aggregating all spatial collocation points $(x_i, y_i, z_i)$ from the refinement levels $\{\texttt{rl0}, \texttt{rl1}, \ldots, \texttt{rl4}\}$ at a single time slice $t$. The NeFs are trained on this coalesced hierarchy, resembling an effective multiresolution representation, while discarding any duplicate points, yielding a single non-uniform grid.

Table 12: **Grid refinement hierarchy and resolution parameters** for the fixed mesh refinement (FMR) configuration used in the TOV benchmark. The spatial resolution $\Delta x$ and number of time slices $N_t$ increase with refinement level, while the physical domain shrinks correspondingly. The last column lists the full 4D grid shape $(N_t, N_x, N_y, N_z)$ for each refinement level. Apart from the medium resolution simulation grid, all other grids don't contain the ghost zones.

| NR Simulation Dataset | Refinement level | Spatial domain | $\mathbf{\Delta x}$ | 4D grid shape $(\mathbf{N_t, N_x, N_y, N_z})$ |
|---|---|---|---|---|
| **Medium resolution** (simulation grid) | rl0 | $[-38.4, 345.6]^3$ | 12.8 | (313, 31, 31, 31) |
| | rl1 | $[-19.2, 217.6]^3$ | 6.4 | (626, 38, 38, 38) |
| | rl2 | $[-9.6, 108.8]^3$ | 3.2 | (1251, 38, 38, 38) |
| | rl3 | $[-4.8, 52.8]^3$ | 1.6 | (2501, 37, 37, 37) |
| | rl4 | $[-2.4, 24.8]^3$ | 0.8 | (5001, 35, 35, 35) |
| **Medium resolution** (training grid) | rl0 | $[0.0, 307.2]^3$ | 12.8 | (313, 25, 25, 25) |
| | rl1 | $[0.0, 198.4]^3$ | 6.4 | (626, 32, 32, 32) |
| | rl2 | $[0.0, 99.2]^3$ | 3.2 | (626, 32, 32, 32) |
| | rl3 | $[0.0, 48.0]^3$ | 1.6 | (626, 31, 31, 31) |
| | rl4 | $[0.0, 22.4]^3$ | 0.8 | (626, 29, 29, 29) |
| **High resolution** (evaluation grid) | rl0 | $[0.0, 307.2]^3$ | 8.0 | (501, 40, 40, 40) |
| | rl1 | $[0.0, 198.4]^3$ | 4.0 | (1001, 40, 40, 40) |
| | rl2 | $[0.0, 99.2]^3$ | 2.0 | (2001, 40, 40, 40) |
| | rl3 | $[0.0, 48.0]^3$ | 1.0 | (4001, 40, 40, 40) |
| | rl4 | $[0.0, 22.4]^3$ | 0.5 | (8001, 40, 40, 40) |

**Tensor differentiation on NR grids.** Numerical relativity solvers augment the primary computational domain with *ghost zones* (Thornburg, 2004), which provide additional layers of points surrounding the grid. These zones support high-order finite difference stencils, ensure that boundary conditions are applied consistently, and help manage coordinate singularities. The simulations used here employ three ghost zones in each spatial direction. Tensor derivatives are evaluated using a sixth-order central finite difference stencil applied to the metric tensor grid functions:

$$f'(x) \approx \frac{-f(x+3h) + 9f(x+2h) - 45f(x+h) + 45f(x-h) - 9f(x-2h) + f(x-3h)}{60h}.$$

---

[12]See https://einsteintoolkit.org/gallery/ns/index.html for details

The three ghost zones are required because this stencil accesses three neighbouring points on both sides of each evaluated location.

**Training specifics.** The full four-dimensional numerical relativity dataset contains approximately 160 million collocation points, including contributions from the refinement hierarchy. This scale necessitates hyperparameters distinct from those used for the analytic benchmarks in EinFields. We retain SiLU activations and the SOAP optimizer, and introduce the following modifications: (i) a Fourier embedding layer of size 256 for input normalization with embedding frequency scale of 0.01, (ii) a cosine learning-rate schedule with initial learning rate $10^{-3}$ annealed upto $10^{-5}$, (iii) a batch size of $10^5$, and (iv) 200 training epochs with 71666 decay steps.

**Evaluation procedure.** EinFields is trained on the medium-resolution coalesced FMR grid (see Table 12). As no analytical solution exists for the oscillating TOV configuration, we adopt the highest-resolution BSSN simulation (Table 12) as our ground truth metric field values. Although this evaluation grid features finer spatial (and temporal) resolution at every refinement level `rlx`, its spatial domain is restricted to the same coordinate ranges corresponding to the medium-resolution training grid data. We therefore evaluate EinFields on this high-resolution grid within the identical domain range, enabling a direct comparison against the ground truth solution. This procedure yields the results reported in Table 3 and assesses the ability of the implicit representations to provide continuous query access of the metric tensor field values at coordinates not present in the training grid.

**Run-time tradeoffs and query speed.** We plot the query speeds of EinFields and its AD-derived Jacobian values over the collocation points.

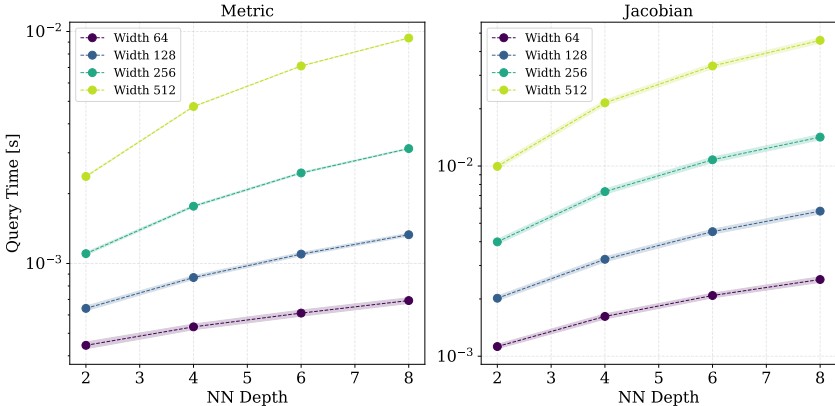

Figure 23: The time to query EinFields on a $10^5$ batch of points. The model is trained on the neutron star numerical relativity simulation containing approximately 71 million collocation points. There is a clear trend of increasing query time as the MLP model size increases. Shaded regions indicate a small uncertainty (only visible when zoomed in). Timings are performed on an NVIDIA H200 GPU with `jAX.JIT`.

**Retrieving higher derivative quantities.** To extend our analysis to the differentiation of tensorial quantities, we additionally evaluate the Christoffel symbols. Specifically, we compare the Christoffel symbols obtained by our automatic differentiation–based EinFields pipeline with those computed using a fourth-order central finite-difference stencil applied to the reconstructed four-metric on this multi-resolution coalesced mesh. The finite-difference operator acts on metric data that include three padded ghost zones and are generated from the BSSN formulation. We use a fourth-order stencil rather than a sixth-order one to maintain numerical stability. All comparisons are carried out at high resolution, using the simulation configurations listed in Table 12, and as done for the metric tensor field. The accuracy results for the NR simulation Christoffel symbols are reported in Table 13. Additionally we report the compression ratio obtained for storing such quantities in explicit form as compared to our implicit machinery.

Table 13: Performance and compression for oscillating neutron star: EinFields retrieved Christoffel symbols as compared to FD methods applied on the multi-resolution coalesced FMR grids.

| Geometric quantity | Storage [GiB] | | MAE | Compression factor (Sym.) |
|---|---|---|---|---|
| | **Full** | **Sym.** | **EinFields (AD)** | |
| Christoffel symbol | 17.0 | 10.6 | 8.61e-4 | **7753×** |

**Static mesh refinement evaluation.** In the previous set of numerical relativity experiments, both the simulation and training grids preserved a fixed block structure over the full BSSN time evolution. Each refinement level was pre-defined and spatially static, with no patch motion or regridding. However, a technical inconsistency arises from the fact that coalesced FMR grids, as commonly used, contain coarse-level cells embedded inside regions where finer patches are available. As a result, multiple grid resolutions represent the same physical region simultaneously. Retaining all levels produces redundant sampling density, and therefore does not reflect a strictly hierarchical representation of the spacetime domain.

A more consistent procedure is to remove coarse-level grid points that fall within the bounding region of any finer-level patch. In practice, we retain only the highest-resolution data available at each spatial location. This construction yields what is effectively a *static mesh refinement* (SMR) hierarchy, analogous to a static octree, in which the resolution is high only near the neutron star and decreases smoothly with distance. Unlike adaptive mesh refinement (AMR), SMR does not dynamically move patches, perform tagging, or trigger online mesh adaptation. Nonetheless, this setting forms a strong and meaningful test of implicit field models, since the resulting sampling distribution is highly non-uniform and reflects true octree-like scaling.

Formally, the refinement bounding boxes satisfy:

$$\text{BBox}(\texttt{rl4}) \subset \text{BBox}(\texttt{rl3}) \subset \cdots \subset \text{BBox}(\texttt{rl0}), \tag{119}$$

where $\text{BBox}(\texttt{rlk})$ denotes the spatial extent of refinement level $\texttt{rlk}$, with $\texttt{rl4}$ the finest and $\texttt{rl0}$ the coarsest. Coarse-level points inside the domain of any finer box are removed, eliminating multi-resolution overlap. The result for this strategy are also detailed in Table 14.

Table 14: Evaluation of EinFields (best model $6 \times 256$) on a single stable neutron star simulation under static octree/static mesh refinement (SMR). We remove coarse-level samples that lie inside the regions covered by finer patches (Eq. 119), retaining only the highest available resolution at each spatial location. The table reports the relative $\ell_2$ error, MAE, storage footprint, and the resulting compression ratio. Under SMR reduction, EinFields attains a compression ratio of approximately $1900\times$ while preserving low reconstruction error, suggesting that eliminating overlapping coarse-resolution points improves metric fidelity in high-resolution regions.

| Representation | Rel. $\ell_2$ | MAE | Storage | Compression |
|---|---|---|---|---|
| EinFields | 1.09e-5 | 4.4e-5 | **1.4** `MiB` | **1901** |
| EinFields (+ Jacobian) | 6.82e-6 | 9.24e-6 | **1.4** `MiB` | **1901** |
| SMR grid | — | — | 2.6 `GiB` | — |

This benchmark provides a representative NR workflow in which matter and spacetime are tightly coupled. It therefore offers an ideal context for assessing the robustness and efficiency of EINFIELDS when applied to high-resolution simulations of relativistic astrophysical systems.

