# OpenReview forum: "Einstein Fields: A Neural Perspective To Computational General Relativity"
_ICLR.cc/2026/Conference — ICLR 2026 Poster_

### Official Review · Reviewer_DSBV · 2025-10-31

**Soundness:** 3
**Presentation:** 3
**Contribution:** 3
**Rating:** 8
**Confidence:** 3

**Summary:**

This paper introduces Einstein Fields (EinFields), a neural field approach to compress and represent 4D spacetime metrics from general relativity simulations. The method achieves up to 4,000× compression of metric tensor fields with 7-9 decimal digit accuracy (1E-7 to 1E-9 relative precision) while providing discretization-free continuous representations that can be trained on arbitrary point samples and queried at arbitrary resolutions. A key contribution is improved differentiation accuracy through automatic differentiation (AD), computing Christoffel symbols, Riemann tensors, and other derived quantities with up to 10^5 better accuracy than finite difference methods in FLOAT32. The approach parametrizes the metric distortion (deviation from flat space) using MLPs and employs Sobolev supervision. The validation focuses on three analytical solutions to Einstein's field equations: Schwarzschild, Kerr, and linearized gravitational waves, successfully reconstructing key relativistic phenomena.

**Strengths:**

## Strengths

**Originality**: This paper presents a novel application of neural fields to general relativity, introducing the first implicit neural representation for tensor-valued spacetime geometries. The approach creatively adapts neural field techniques from computer vision to computational physics, with several original contributions.

**Quality**: The paper demonstrates strong technical rigor with comprehensive validation across multiple canonical GR test cases (Schwarzschild, Kerr, gravitational waves). The evaluation methodology is sound. Ablation studies (Table 3) properly isolate the contributions of different design choices. The authors are transparent about limitations.

**Clarity**: The paper is well-written and accessible. The background section (Section 2) effectively introduces both GR concepts and neural fields. Figure 1 provides an excellent conceptual overview of the pipeline. The mathematical notation is consistent and properly defined (though dense in places).

**Significance**: This work addresses genuine computational bottlenecks in numerical relativity—storage (petabytes per simulation) and accurate tensor differentiation. The 4,000× compression factor and 10^5 improvement in derivative accuracy (FLOAT32) represent substantial practical gains.

**Weaknesses:**

## Weaknesses

**Limited experimental scope**: The validation is restricted to three analytical solutions to Einstein's field equations (Schwarzschild, Kerr, linearized gravitational waves). While these are canonical test cases, they represent idealized scenarios far simpler than realistic numerical relativity (NR) simulations.

**Limited contextualization within scientific computing:** While the introduction mentions neural fields and ML for scientific computing, it lacks: (1) discussion of prior ML work specifically targeting numerical relativity or gravitational physics, (2) comparison with traditional compression methods used in scientific computing, and (3) detailed positioning relative to neural operators and PINNs. A dedicated related work section would help readers better understand the landscape and the paper's specific contributions.

**Missing Error Quantification:** Tables 1-3 report single-valued metrics without error bars or confidence intervals. Table 1 mentions selecting 'the model with the lowest MAE,' suggesting multiple runs were performed but statistics are not reported.

### Minor issues:

Page 10, line 490: "supplimentary" → "supplementary"

Figure 4 a caption: "Perihilion precession" → "Perihelion precession"

**Questions:**

## Questions


* Actual NR simulation data: Your validation strategy using analytical solutions (Schwarzschild, Kerr, linearized GW) with known ground truth is appropriate for demonstrating the method's capabilities. As a natural next step, have you tested EinFields on any actual numerical relativity simulation outputs, even at small scale? What additional challenges arise with real NR data?

* Table 1 mentions selecting "the model with the lowest MAE" - how many training runs were performed? Can you report mean ± standard deviation over multiple random seeds for the key results in Tables 1-3? This is important for assessing reproducibility and typical vs. best-case performance.

* Parameter generalization: Do you train a separate network for each physical configuration (M, a, etc.), or can one network generalize across parameter ranges?

---

> ### Author Response · Authors · 2025-11-20
> **EinFields Applied to Actual NR simulation data**
>
> **Q1, (and W1) Actual numerical relativity simulation data+ EinField training**
>
> We thank the reviewer for this relevant comment. We have added an additional experiment to evaluate EinFields on a more practical numerical relativity simulation. In particular, we focus on the evolutionary osciallation of a neutron star. This is a time-dependent problem that, contrary to our previous use-cases, does not possess an analytic solution and is computed numerically on a refined mesh. The simulation is evolved in time using the state-of-the-art BSSN (Baumgarte Shapiro Shibata Nakamura) formulation as implemented in the well-known open source framework (EinsteinToolkit~https://einsteintoolkit.org/gallery/ns/index.html). The numerical evolution employs fixed mesh refined (FMR) patches with time steps that vary across refinement levels in order to satisfy the Courant Friedrichs Levy (CFL) condition. This yields us a multi resolution grid obtained by coalescing these different patches.
>
> We evaluate EinFields on this numerical relativity simulation and provide a detailed analysis of this use case in the revised manuscript. The discussion appears in the section on the oscillating neutron star simulation. This point is related to the concern raised by reviewer kkPc, and we address both issues together, including evaluation results and the corresponding Pareto fronts.
> Addressing the question raised, we identified several challenges that arise when working with real numerical relativity data.
>  a) The number of collocation points is significantly larger than in our previous use cases. Typical simulations contain more than 71 million collocation points  that are refined on multiple patches. This substantially increases the training time of the neural fields, compared to the previous use cases with around 2 million points.
>  b) Numerical relativity simulations often employ different gauge choices, which serve as symmetries that simplify the evolution. The post hoc analysis performed by EinFields requires appropriate handling or integration of these gauge conditions. This is an aspect that we plan to address in future work.
>  c) Both training time and query time increase in the presence of large scale time evolving meshes, as also noted by reviewer kkPC. We expect, however, that several of these limitations can be mitigated in the near future and can be integrated into NR workflows.
> Despite these challenges, our differentiable modeling framework with automatic differentiation provides clear advantages. Furthermore, the neural representation offers continuous access at arbitrary resolution within the simulation domain and provides a highly compressed storage format (2000x) compared to explicit refined grids used in standard numerical relativity simulations. All of this is detailed in the *main paper between pg 8-10*.

---

> ### Author Response · Authors · 2025-11-20
>
> **Q2 (and W3) Error bars and statistics on pareto-fronts**
>
> We appreciate the reviewer’s helpful observation regarding the reporting of statistics on the Pareto front plots. We agree with this recommendation and will upload revised figures that include error bars in the main manuscript. The training runs presented in the current version were performed using a fixed random seed for each model size. Because the Pareto fronts were primarily intended to illustrate the relationship between relative L2 or MAE error and the degree of neural compression, we did not conduct a sweep over multiple random seeds/initializations. We recognize that this is an important open point for ensuring reproducibility and for reporting best case performance, and we will address it in the revised version.

---

> ### Author Response · Authors · 2025-11-20
>
> **Q3 Parameter generalization**
>
> The training procedure employs the same network architecture for each analytic physical configuration, including the Schwarzschild spacetime, the Kerr spacetime with nonzero spin parameter a, and the linearized gravity setting. For the numerical relativity simulation, however, we adopt a different network configuration. This is necessary because the time-evolved simulation grid extends to T=1000M and contains between 1000 and 10000 time steps depending on the refinement level, which imposes requirements that differ significantly from those of the analytic cases (for e.g. normalization layers or Fourier feature layers). The details of the network architecture used for the NR experiment are discussed at detail in Appendix – Additional Experiments.

---

> > ### Author Response · Authors · 2025-11-20
> >
> > **W2 Limited contextualization within scientific computing**
> >
> > We thank the reviewer for highlighting this point. In response, we have added a dedicated discussion of these topics in Appendix E.1  Related Work Continued (Extension). Due to strict page limitations, this material could not be included in the main paper, but we will integrate it into the primary manuscript upon acceptance.
> > Appendix E.1 now provides (a) an overview of prior machine-learning work in gravitational physics and multi-messenger astronomy, (b) a discussion of both lossless and lossy compression methods commonly used in scientific computing and large-scale simulation workflows, and (c) a detailed positioning of neural operators, PINNs, and neural fields within the broader landscape of scientific ML.

---

> ### Comment · Reviewer_DSBV · 2025-11-26
> **Response to Authors Comments**
>
> Thank you for the comprehensive responses and the additional experimental work. I appreciate the effort to address the concerns raised.
>
> **Regarding Q1/W1 (NR simulation data):**
> The addition of the oscillating neutron star experiment substantially strengthens the paper and directly addresses my primary concern about limited experimental scope. The challenges you identify (71M collocation points, gauge choices, increased training/query time) are valuable insights for the community. A few follow-up questions, (mostly for personal interest):
>
> * What is the actual training time for the neutron star case compared to the analytical solutions?
>
> * You mention 2000× compression for the NR simulation—how does the accuracy (MAE/L2 error) compare to the analytical cases?
>
> * Could you clarify what you mean by "appropriate handling or integration of gauge conditions" for post hoc analysis?
>
> **Regarding Q2/W3 (Error bars)**
> Thank you for acknowledging this point. Including error bars over multiple random seeds will significantly strengthen the reproducibility claims. I recommend reporting statistics for at least the key results in Tables 1-2, even if not all configurations.
>
> **Regarding W2 (Related work)**
> The addition of Appendix E.1 is definitely helpful.
>
> **Overall:**
> The authors have made substantial improvements, particularly the NR simulation experiment. Pending the inclusion of error bars and minor clarifications, I am raising my score and suggesting this submission to be considered for Spotlight, due to its relevance for the AI4Science community.

---

> ### Author Response · Authors · 2025-11-28
> **Acknowledment on the increased score and Spotlight recomendation**
>
> We thank for positively recognizing the improvements in the manuscript resulting from the inclusion of realistic NR simulation data for single stable neutron stars. We also sincerely appreciate the reviewer for raising the scores after our revisions and for recommending the work for **Spotlight** consideration.

---

> ### Author Response · Authors · 2025-11-28
> **NR simulation data follow up**
>
> **Q1 (and W1)**
>
> We also correspondingly thank the reviewer for the highly relevant follow-up questions, which also gives us room to work-in these other crucial aspects into the camera ready version, if selected:
>
> 1. **Training time comparison**
> We report the wall-clock training time for the TOV/single stable neutron star use case using our best performing architecture, which is a 6×256 MLP with Fourier layers (Fourier embedding frequency 0.01) trained on 71 million collocation points.
>
> | Method                    | Training Time          |
> | ------------------------- | ---------------------- |
> | Without Sobolev training  | 00:34:45 (approximately  34 mins) |
> | With Jacobian supervision | 02:41:07 (approximately 2hrs 41 mins) |
>
> Training becomes substantially more expensive once scaled to the four dimensional NR simulation with multi-resolution patches consisting of approximately *71–78* million collocation points on an irregular grid. This requires roughly *34 minutes*, compared with approximately *2.5 minutes* for the analytic case using only about two million collocation points on a regular grid. The analytic scenarios are far less complex and, importantly, static in time for the Schwarzschild and Kerr cases. The gravitational wave case is constrained primarily along the t and z directions and exhibits significant variation effectively only along two axes, further reducing complexity.
>
> It is worth noting that the primary objective of this work was not training speed. This project represents an initial step toward applying neural fields to general relativistic simulations, with emphasis placed on neural compression and enabling differentiable workflows based on implicit spacetime representations. Future work will include improvements in training efficiency, particularly through hybrid neural field architectures such as those discussed in hybrid neural fields as in the case of InstantNGP type frameworks (https://arxiv.org/abs/2201.05989)
>
> 2. **Rel. L2 and MAE for analytic versus numerical relativity cases**
>
> The compression ratio remains substantial even for the NR experiments, on the order of 2000×. Accuracy metrics comparing analytic and NR data are summarized below.
>
> For the numerical relativity data, both Rel. L2 and MAE decrease by up to one order of magnitude with Jacobian supervision and also without it. This is expected, since the numerically evolved BSSN simulations used for neutron star configurations are inherently more complex than the analytic scenarios. The analytic cases involve far simpler spacetime geometries, which naturally admit lower error under comparable network capacity.
>
> Due to limited time for conducting new experiments during the rebuttal period, we did not perform an extensive hyperparameter sweep. It is highly plausible that a more comprehensive search would further reduce the reported errors. These results therefore should be understood not as the attainable limit, but as the performance under a reasonable and constrained training configuration.
>
> 3. **Handling of gauge conditions and reconstruction**
>
> The numerical relativity experiments in the rebuttal are based on Einstein Toolkit simulations that use the standard BSSN formulation. In particular, the evolution employs the following gauge choices
>
> (i) the $1+log$ slicing condition for the lapse $\partial_t \alpha = -2 \alpha K$ which determines the foliation into hypersurfaces of constant coordinate time, and,
>
> (ii) the Gamma-driver condition for the shift $\beta^i$ which governs evolution of spatial coordinates.
>
> The BSSN formulation evolves the conformal metric and the conformal factor, which jointly determine the relation between the evolved fields and the physical spacetime metric.
>
> In our setup, EinFields reconstructs the full spacetime metric from the ADM variables $(\alpha, \beta^i, \gamma_{ij})$ post-hoc through a neural representation. Any downstream operation such as differentiation, curvature computation, or Jacobian estimation must therefore remain consistent with the gauge used during the simulation. These derived geometric quantities depend explicitly on lapse, shift, and conformal scaling, all of which are gauge-dependent. Consequently, post-hoc tensor-based analysis requires that gauge information be preserved and interpreted correctly.
>
> If the gauge is not treated consistently, geometric quantities and cross-dataset comparisons can become ambiguous or physically misleading. Explicit gauge treatment and inclusion of geometric priors are important next steps for applying EinFields to broader and more varied NR datasets.

---

> ### Author Response · Authors · 2025-11-28
> **Error bars and statistics**
>
> **Q2 (and W3)** :
>
> We thank the reviewer for the constructive suggestion regarding the inclusion of error bars and statistical measures in the Pareto front plots and in **Tables 1 and 2**. We agree that reporting variance across independent runs is important, especially since different random initializations of the NNs can introduce noticeable fluctuations in accuracy metrics such as Relative L2 and MAE. While the qualitative conclusions of our work remain unchanged, this observation highlights the relevance of random seed sweeps in neural field training and provides a quantitative measure of variability in accuracy. Adding this, not only strengthens our paper but also equally makes our experiments reproducible.
>
> We have incorporated this feedback in the revised manuscript. The updated *Pareto fronts* now include error bars, and the revised versions of **Tables 1 and 2** contain corresponding statistical measures. We note that random initialization studies were not extended to the NR simulations due to limited available time and the significantly higher training cost of these runs. Nevertheless, we acknowledge that this type of analysis is particularly important for realistic NR datasets and for the stability of post-hoc derived quantities.
>
> We plan to include random initialization statistics for NR experiments as well, and to update Pareto plots **Fig. 7a** and **Table 3** accordingly upon acceptance.

---

> > ### Author Response · Authors · 2025-11-28
> > **Additional experiments using NR simulation data**
> >
> > 1. We have expanded the manuscript with an additional paragraph in **Appendix G**, focused on **Retrieving higher derivative quantities** from NR simulations. In this section, we present the accuracy of the AD based EinFields reconstructed Christoffel symbols, evaluated against a fourth order central difference estimate obtained directly from explicit grid data (padded with 3 ghost zones for accurate derivative computation on these multi-resolution/heterogeneous grids) for the single stable neutron star 4D simulation. The MAE for this comparison is reported in **Table 13 in Appendix G**, together with the associated neural compression factor. We note that the experiment does not include a comparison between medium resolution FD and high resolution FD (which serves as ground truth for this evaluation). Performing such an analysis would require accurate interpolation between heterogeneous grids, which is nontrivial and outside the scope of this study. The intention of this experiment is instead to demonstrate that EinFields achieves a moderate MAE of approximately $8e{−4}$ while providing a compression factor of roughly $8000\times$ for Christoffel symbols.
> >
> >
> >
> > 2. We also implemented a new set of experiments for the same neutron star simulations that employ fixed mesh refinement. The simulation consists of nested grid patches $rl4$ (finest) through $rl0$ (coarsest), which are typically combined into a multi resolution collocation set. In the default configuration, coarse level grid points remain present inside finer refinement regions, which leads to redundant sampling. A more principled strategy is to remove coarse level points wherever higher resolution samples exist. This retains only the most resolved data available at each spatial point and produces a hierarchy that resembles a static octree. In numerical relativity this configuration is usually referred to as *static mesh refinement (SMR)*. The results from these additional SMR based experiments are included in **Appendix G** under Static mesh refinement evaluation, along with quantitative metrics in **Table 14**. We are open to integrating these results into the main text upon acceptance, provided page constraints allow it.

---

### Official Review · Reviewer_Lt1j · 2025-11-01

**Soundness:** 3
**Presentation:** 4
**Contribution:** 4
**Rating:** 8
**Confidence:** 2

**Summary:**

This paper derives a novel neural representation method to compress the relativity simulations.

**Strengths:**

- Outstanding efficiency and accuracy are shown when representing the symmetry of the simulations.
- This paper is especially well written and well presented.
- The problems addressed by the new tool is of interest to a wide community.

**Weaknesses:**

- I am not entirely sure that how interesting this paper will be for the readership of ICLR, of whom so few are well versed in this area.

**Questions:**

See above.

---

> ### Author Response · Authors · 2025-11-20
>
> We thank the reviewer for their feedback. We would like to briefly highlight why we believe this work is relevant to the conference’s audience. ICLR has active communities on implicit neural representations and scientific ML, which this work addresses on several fronts, e.g., by extending INRs from scalar fields to tensor fields with broader implications for representation learning of geometric objects (discussed more in our response to reviewer kkPc, W4). It also includes highly topical ideas from differentiable physics and complex automatic differentiation pipelines, continuous representations, compression, and high-precision NN training. Beside the technical aspects, we hope the introduction of numerical general relativity to the ML community will catalyze exciting research efforts at the intersection of these two fields, opening up new avenues for exploration.

---

### Official Review · Reviewer_kkPc · 2025-11-04

**Soundness:** 2
**Presentation:** 3
**Contribution:** 2
**Rating:** 4
**Confidence:** 4

**Summary:**

This paper presents a neural tensor‑field (tensors in the General Relativity sense) parametrization of GR metrics with a JAX implementation for differential geometry operators wired through automatic differentiation. The demos (Schwarzschild/Kerr orbits, Weyl scalars, ring deformation under a linearized GW) look correct and generate polished images, and the ablations around Jacobian/Hessian supervision are good.

That said, I have major concerns with the claims and evaluations. I believe these are not aligned with numerical relativity (NR) practice, and so this is not yet at the point of being useful for actual science. Thus, the current narrative is a bit misleading. The headline is "compressing 4D NR simulations by 1000-4000x with better derivatives than FD in FLOAT32," but most experiments are static analytic 3D cases (t = 0) plus one simple time‑varying GW; storage comparisons are made against a dense "explicit grid" strawman (modern NR code uses AMR or pseudo-spectral methods); baselines omit spectral/ROM methods; coordinate chart sensitivity is large; and long‑horizon dynamics need FLOAT64 to avoid divergence. I think it's a promising direction for ML to help, but I think this paper needs to be honest about the current status of such an approach. Doing so would not only be better for the paper, but I think also benefit the authors in that it would point out to the ML community where more work is needed. Basically I would like to see the narrative of this paper modified to be honest about the practicality of this, and about the toy baselines, before I consider acceptance, especially as general ML audiences will not know how to evaluate this.

**Strengths:**

- I like the clear pipeline and library, and a JAX code for this seems useful for the NR community. The graph from metric to derived quantities is explicit and leverages forward‑mode Jacobians/Hessians with einsum operations which is nice. This is a useful contribution of reusable tooling for GR in ML, and I think it is a good contribution by itself.
- I liked the ablations accounting done in 4.2, it is nice and I think quite useful to see the effect of every modification to the training process. It is interesting to see that Jacobian/Hessian supervision indeed helps.
- Neat canonical tests: precession, circular/eccentric Kerr orbits, ring response under a linearized GW, and Weyl/Kretschmann diagnostics are shown and largely match analytics on short horizons.
- Multi‑chart training attempt: training/evaluating in multiple coordinate charts acknowledges a real pain point in GR ML (though this is near the end of the appendix, I think it should come earlier)

**Weaknesses:**

First, my main concerns:

1. First, I think the scope and storage comparisons are misaligned with NR. The paper advertises 4D compression of NR simulations, but the primary experiments are analytic snapshots at t = 0 (Schwarzschild/Kerr); only the linearized‑gravity toy has time evolution. The compression factors compare MLP weights to explicit dense grids counted as "#points x 4 bytes" in FLOAT32, which is not how NR codes actually store data (they would use adaptive mesh refinement stored with an octree). So the 1000–4000x headline is basically comparing against a strawman and misleading to the ML community about the state of this domain.
2. The paper itself notes that modern NR "increasingly opts for (pseudo‑)spectral methods ... up to 1000–5000x faster on CPUs than FD on GPUs at comparable accuracy." But all quantitative baselines in this paper are finite difference stencils (on a uniform grid - but the only actual finite difference codes used in NR are based on adaptive meshes) and an analytic AD. There's no head‑to‑head vs actual NR codes used in GW modeling. This makes it hard to situate their method, even for someone who knows NR, let alone the ML community.
3. The paper claims up to five orders of magnitude derivative gains over FD in FLOAT32, but geodesic integration requires FLOAT64 and long‑time rollouts still diverge (the authors show this in their own figures and explicitly state that they only surpass FD in single precision). This puts the method far from NR‑readiness, where double precision (or even higher) is standard.
4. There is large coordinate‑chart sensitivity which is a bit worrisome. Table 8 (deep in the appendix) reports up to three orders of magnitude variation in "Rel‑L2" error across charts for the same spacetime. That undermines generality claims unless the representation or training explicitly handles diffeomorphisms or evaluates with chart‑invariant metrics.
5. The physics is not enforced or audited. The pipeline mentions Bianchi identities, but experiments focus on pointwise tensor errors, scalar invariants, and geodesic tests. While the pointwise tensor errors are no doubt useful in clarifying (to a NR person) that these methods aren't yet ready, there's no reporting of physics checks, like conservation laws, etc., which are exactly the diagnostics one needs to trust a compressed metric in downstream NR workflows.
6. I am a bit confused about the "discretization‑free" claims, since it seems the method is ultimately trained on a grid. Several places describe training on 4D spacetime grids or "4D training and validation grid data," which undercuts the claim of being discretization‑free. Even an INR is ultimately a finite parametrization/basis.
7. The throughput trade‑off is not discussed. Even if the file is tiny, post‑processing requires many MLP queries to reconstruct fields, whereas decompressing spectral grids is reading coefficients + evaluating polynomials. You still likely win on storage, but the compute/runtime trade‑off for analysis & viz should be stated.

Second, some other suggestions/comments:

- I make the following comment purely to help the authors improve their work, and do not include this comment as part of my score for the paper, so feel free to not address this in your rebuttal. It is simply a suggestion/idea. So I think the branding of the method as "Einstein Fields" is not an optimal choice, because it would likely conflict with "Einstein Field Equations" in any search. Also, to the physicist, who I assume you would like to include in the audience of the paper, it does not give them any idea that this is related to machine learning.
- Obviously GR is quite complicated to someone with no background. I am not sure it is possible to give it much of an introduction here, and I worry the current introduction might give the wrong ideas. I think it is better to simply direct the reader to an online resource, rather than give an inevitably incomplete description of the mathematics in the appendix. Try to think think about what purpose it serves: (1) for those who don't know GR, this is not going to be nearly enough to introduce them even to the basics; (2) for those who do know GR, they will not need this anyways. So, why include it at all? Consider, for audience (1) I think you need to simply target the intuition for each variable you model. That is the fundamentally useful thing to write about. And for audience (2) (curious physicists, maybe) I think you simply need to _translate_ GR concepts to machine learning for them. So, when you consider these two audiences, the appendix seems to serve little purpose. I recommend trying the split approach above: focus on intuition of the key target variables for the non‑physicists (and point them to other resources), and focus on translation of the machine learning concepts for the physicists. This would be much more effective in my view.
  - Also, I did not check through all the math in the appendix. Thus, there could be errors.
- It might be worth calling a GR tensor exactly that: a "GR tensor," to differentiate from the ML meaning.
- Use proper scientific notation instead of "5.37E‑6" in tables.
- Even if the file is small, you still must run tens of thousands of MLP queries for post‑analysis/viz; be transparent about that runtime trade‑off versus reading spectral coefficients.
- Figure 1 is noisy. Consider simplifying or splitting. (The caption itself also states training on a 4D spacetime grid, which conflicts with "discretization‑free" messaging.)

**Questions:**

- Please clarify the precise data format used for training ("4D spacetime grid" vs "arbitrary samples"), since the paper simultaneously describes the approach as discretization-free yet refers to training on regular grids.
- Could you provide any comparison, even small-scale, against a spectral or AMR baseline (ideally an actual code used by the GR community) to contextualize the claimed compression?
- Please explain how derivative accuracy scales in FLOAT64 and whether the observed long-time geodesic divergence persists under higher precision.

---

> ### Author Response · Authors · 2025-11-20
>
> We thank the reviewer for their considerate feedback which helps us improve and better position our work. In summary, we have performed an additional experiment on a time-varying numerical solution on a refined mesh from a well-established NR solver, addressing the highlighted shortcoming of the previous experiments. We also thank the reviewer for their feedback on the narrative, which we failed to convey as intended. We have made an effort to improve the positioning in the added revision, in addition to other changes all marked in blue. In the following rebuttal, we address both the primary questions and the mentioned weaknesses to the extent possible. We hope the additional results, the more appropriate positioning, and the clarification of the remaining points makes for an improved work.
>
> **Q1 (and W6) Training point arrangement**:
> As an MLP, EinFields can be queried at and trained on arbitrary point sets (line 61). As the reviewer rightfully points out, despite this, the submission repeatedly (e.g., lines 158, 260, 266, fig. 1, tab. 6) states that the models are trained on 4D grids. To clarify, the synthetic data we produce from the analytical solutions is indeed queried on regular grids in the respective canonical coordinate systems. This has no particular motivation other than a grid being the simplest choice. As a counterexample, the experiments in the varied coordinate systems (appendix E.7) transform canonical coordinate grids into different charts (e.g., spherical grid to Cartesian). As a result, these training points do not form a grid. We hope this and the additional NR data experiment (described next under Q2) illustrate the input flexibility. We have adjusted the phrasing in the submission to clarify this point.

---

> ### Author Response · Authors · 2025-11-20
>
> **Q3 (and W3) float64 -- tensor derivatives and long horizon time rollout errors:**
> We appreciate the reviewer’s insightful comment and fully acknowledge this limitation. As noted, state of the art numerical relativity simulations typically employ double precision (float64) or higher, which is essential for achieving the extremely high accuracy required in these computations. At present, this remains an open challenge in the broader scientific machine learning community. Frameworks such as physics informed neural networks (PINNs) and neural field based parameterizations are generally unable to reach floating point accuracy, even for scalar valued PDEs. Reported errors commonly fall in the range of $10^{-6}-10^{-8}$, and the use of FP64 weights instead of FP32 provides only marginal improvements. This limitation stems from the representational and optimization properties of neural network parameterizations rather than from the floating point format itself.
> Consequently, matching double precision accuracy for tensor valued quantities in EinFields, including recovered dynamical quantities such as geodesic trajectories and higher order tensors, is not currently feasible. FP64 remains orders of magnitude more accurate for these operations, and divergence in geodesic evolution persists even when FP64 is used. Nonetheless, recent progress in implicit neural representations and physics informed learning, including multi stage training, second order Gauss Newton optimization, and more advanced higher order loss formulations (see for example https://arxiv.org/abs/2509.14185), suggests that accuracies on the order of 10{−15}-10^{-16} may become attainable.
> Since the present work represents an initial step in integrating machine learning and computer vision based techniques into numerical relativity, our future efforts will focus on incorporating these more advanced strategies and benchmarking the results directly against FP64 baselines.
>
> We continue with addressing the remaining weaknesses:
>
> **W4 Coordinate-chart sensitivity**:
> We appreciate the reviewer sharing our concern about the coordinate-chart sensitivity. While tensors in GR are fundamentally geometric objects (i.e., coordinate independent), their numerical manipulation requires tensor *components* which only exist in a coordinate chart. This is a characteristic of any numerical treatment of tensors, whether in simulators or NNs. This means that the usual error norms of the components are not coordinate invariant. As a consequence, one can construct charts in which the components of any tensor (including the “error tensor”) can all be arbitrarily driven toward 0 by simply rescaling the coordinates. We point out this “exploit” and avoid it by sticking to canonical coordinates. This also motivated experiments in appendix E.7 that the reviewer refers to. Lastly, this is also the reason why we put emphasis on evaluating coordinate independent phenomena, like curvature invariants and geodesics. To the best of our understanding, coordinate sensitivities in both evaluation and training are an underrepresented problem in the scientific ML literature. While we agree with the limitation, we encourage the reviewer to not view this a fallacy of our particular work, and rather recognize these results and the accompanying discussion as a productive contribution as already acknowledged under the Strengths.

---

> ### Author Response · Authors · 2025-11-20
>
> **W5 Physics auditing**:
> Contrary to the concern, we believe we do audit the physics to the extent possible. An important necessary condition for the physical plausibility of a metric is that it satisfies the EFEs. In vacuum, as in the considered examples, these equations simplify to requiring a vanishing Ricci tensor. The deviation from zero (the “ground truth”) is quantified in Table 2 for the Schwarzschild geometry.
> Furthermore, we put an emphasis on measuring problem-specific scalars (e.g., the Kretschmann scalar localizing the ring singularity of a Kerr black hole), as these are numerically accessible coordinate-chart independent objects. While still defined point-wise, these do reveal global structure of the solution.
> Lastly, we explicitly reconstruct some seminal tests of GR, most notably using geodesics, which is a challenging and revealing test because of the error accumulation. In many senses, exactly these phenomena capture the historically relevant physics of these setups.
>
> As for the Bianchi identities specifically, these are pure identities that follow from the definition of the Levi-Civita connection and the Riemann tensor. Consequently, they are satisfied for any metric, and cannot measure the quality of the metric approximation. They can, however, be used to assess the quality of the numerical pipeline that involves repeated differentiation and tensor contractions. Per reviewers suggestion, we report the second Bianchi identity in Table 2 that arises from energy conservation (corresponding to vanishing divergence of the matter). The results indicate that the approximate and analytical metrics both satisfy the Bianchi identity to around $10^{-8}$ verifying the correctness of the AD pipeline.
>
>
> **W7 Query time**:
> We thank the reviewer for highlighting the aspect of query time, which was previously overlooked. The query speed of implicit neural representations is a well-known challenge, and developments in computer graphics have required several innovations to optimize the trade-offs between compression, quality, and evaluation speed.
> Figure 23 in Appendix F now includes the query times over different model sizes. For a large batch of $10^5$ points, a query is on the order of 1-10ms which can be significant in sequential applications, such as geodesic rollouts. We have also added a discuss of this under Limitations.

---

> ### Author Response · Authors · 2025-11-20
> **EinFields application to realistic NR simulation data**
>
> **Q2 (and W1, W2) Numerical relativity usecase**:
>
> We thank the reviewer for the suggestion of including a numerical relativity test-case. We share the sentiment that using NR data is a natural next step, however, a comprehensive study of advanced (pseudo-)spectral methods was beyond the scope of this work, which we tried to indicated in the original future work paragraph, but have now made more explicit in the limitations.
> To address this concern to the best possible extent in the available time, we conducted a numerical relativity simulation of the oscillatory evolution of a perturbed neutron star. Since this problem does not have an analytic solution, its evolution requires a numerical solution of the Einstein field equations. The system is numerically evolved using the well-established Baumgarte-Shapiro-Shibata-Nakamura (BSSN) framework using the EinsteinToolkit implementation (https://einsteintoolkit.org/gallery/ns/index.html).
>
> For this problem, EinsteinToolkit employs fixed-mesh refinement (FMR), with each refinement level evolved independently in time, with more time-steps for the finer levels. Training data is constructed by collecting all spatial collocation points from the available refinement levels at each time slice, while removing duplicates in favor of the finer levels. This procedure produces a multiresolution grid which are frequently deployed for such simulations.
> The first set of results, appended in the *main paper (between pg 8-10)* suggests that even for the numerical time-evolved 3+1D FMR spacetime with over 71 million collocation points, EinFields perform effectively, achieving a relative L2 error of approximately $10^{-5}$ (without Sobolev training) when evaluated against the high-resolution simulation (assumed ground truth as no analytical solution is available), with a compression ratio of roughly 2000x. We are currently extending this to include Sobolev training and will report the corresponding numbers and Pareto fronts for Christoffel symbols as soon as possible.

---

> ### Author Response · Authors · 2025-11-23
> **Clarification: EinFields application to realistic NR simulation data**
>
> **Q2 (and W1, W2) Numerical relativity usecase (evaluation clarification):**
>
> Additionally, we would like to clarify the evaluation strategy used in our NR–simulation experiments, as this was not described in sufficient detail in the revised manuscript.
>
> **Training setup.**
> EinFields is trained on the medium-resolution coalesced FMR grid together with the corresponding metric-tensor gridfunctions (see Table 12, Appendix G). For the metric-only experiments, we train on grids excluding the 3 ghost zones. For the Sobolev (Jacobian-supervised) experiments [in preparation], we include the 3 ghost-zone layers to enable the use of sixth-order centered finite-difference stencils.
>
> **Evaluation setup.**
> For evaluation, we run the highest-resolution (rl0:dx=dy=dz=8, rl4:dx=dy=dz=0.5)  simulation available for this test (cf. Table 12), corresponding to the setup used in the Einstein Toolkit gallery for the oscillating TOV star (runtime ≈ 11 h 38 min). Because no analytical solution exists for this scenario, this high-resolution simulation serves as our ground truth for the metric tensor fields.
>
> We evaluate EinFields for the spatial domain range that falls within the medium-resolution training grid, i.e. identical bounding-box extents for each refinement level, but consisting of higher-resolution grid spacings, giving us access to field values on query coordinates different from our training grid. This allows us to directly assess the model’s ability to interpolate using continuous functions and provide continuous query access on spatiotemporal coordinates not present in the training set.
>
> **Correction in Table 12.**
> We also wish to correct a minor typo in the originally reported high-resolution grid spacings. The values for
> dx,dy,dz. have now been updated. This correction does not affect the results presented in the main text (e.g., Table 3), and the reported relative-L2 errors remain unchanged.
>
> We have uploaded the corrected supplementary material and added a dedicated paragraph on the evaluation procedure in Appendix G (p. 62).
>
> Finally, we note that experiments involving EinFields with Sobolev (Jacobian) supervision are currently in progress, and we will report these results shortly.

---

> ### Author Response · Authors · 2025-11-25
> **Metric Jacobian supervision included evaluation**
>
> **Q2 (and W1, W2) Numerical relativity usecase (evaluation)**
>
> As a follow-up to the discussion above, we additionally report results for EinFields trained with Jacobian supervision. We use the best-performing architecture, i.e., a 6×256 with an additional Fourier layer of size 256 and Fourier embedding frequency 0.01 (see Training specifics, Appendix G). Consistent with our findings on synthetic datasets, Sobolev-style supervision also improves performance on NR-generated data, yielding lower relative L2 and MAE errors and consequently enhancing the reconstruction quality of the implicit neural representations.
>
> The corresponding evaluation results are now included in **Table 3** in the revised version of the manuscript. We note that we did not incorporate Hessian supervision in these experiments, as the associated computational cost becomes substantial given the number of collocation points in the simulation grid. For completeness, we plan to include Hessian-based results in the final camera-ready version, should the paper be accepted.

---

> > ### Author Response · Authors · 2025-11-29
> > **Query speed/compute/runtime trade‑off**
> >
> > **W7 Query time**:
> >
> > For completeness, we have also included timing results for Jacobian evaluation in EinFields, since this quantity is central for computing Christoffel symbols and for downstream geometric tasks such as curvature tensor estimation. The corresponding query speed measurements are now presented in Fig. 23b. Thus, the runtime/query speeds of both a) the metric (EinFields) and its Jacobian are depicted in Figs 23.

---

> > > ### Author Response · Authors · 2025-11-29
> > > **EinFields application to realistic NR simulation data (Miscellaneous)**
> > >
> > > **Q2 (and W1, W2) Numerical relativity usecase (continued)**
> > >
> > > *Additional experiments using NR simulation data*
> > >
> > > While this is also detailed in the last set of threads in the rebuttal comments below, we mention this here too:
> > >
> > > 1. We have expanded the manuscript with an additional paragraph in **Appendix G**, focused on **Retrieving higher derivative quantities from NR simulations**. All specifics are detailed there and the caveats are mentioned below in detail.
> > >
> > > 2. Extending to **static mesh refinement** (remove coarse level points wherever higher resolution samples exist) experiments are conducted in **Appendix G**  along with quantitative accuracy and storage metrics in **Table 14**.
> > >
> > > *Caveats*
> > >
> > > To enable a fair comparison between Christoffel symbols obtained on the high-resolution grid and those reconstructed using EinFields trained on a medium-resolution grid, we train and evaluate only at temporal values $t$ that exist in both datasets. This restriction is necessary because the CFL condition introduces subcycling in time, producing a denser set of time steps wherever spatial resolution is higher. As a result, a one-to-one comparison is only feasible at temporal instances that coincide across refinement levels, thus avoiding any temporal inconsistency. A detailed evaluation is provided in **Table 13** (Appendix G). We note that achieving fully consistent temporal alignment with neural field interpolation and reconstructing short-scale curvature features remains an open problem, and is part of future work.
> > >
> > > Secondly, spatial correspondence must also be handled carefully. The medium-resolution grid does not cover the full spatial domain of the high-resolution grid (see **Table 12** for grid extents across refinement levels). Therefore we restrict the high-resolution grids to the medium-resolution bounding box before evaluation (also detailed in **Evaluation procedure** in **Appendix G**. Neural fields do not extrapolate reliably to out-of-distribution (OOD) collocation points, which makes this restriction necessary for a meaningful and controlled comparison.
> > >
> > > **Derivatives (Jacobians) of the BSSN-reconstructed spacetime metric**
> > >
> > > A fully gauge-consistent comparison would ideally require Einstein Toolkit (ETK) to provide solver-level access to derivatives of the evolved BSSN variables. In principle, this would involve producing Jacobians, Hessians, and temporal derivatives, and recording gauge variables such as the Gamma-driver shift evolution. These modifications require custom ETK hooks and fall beyond the present rebuttal. Such integration is, however, a priority for future development.
> > >
> > > In the current study we instead compute derivatives using a fourth-order centered finite-difference scheme applied to stored simulation fields. Time derivatives are less accurate due to the large interval between output snapshots, and spatial derivatives are degraded on coarser refinement patches because of larger grid spacings. These limitations propagate into Sobolev-based training and also affect the ground-truth Christoffel symbols used for comparison. Despite this, EinFields attains a mean absolute error of $7 \times 10^{−4}$ on Christoffel symbols reconstruction (**Table 13**), and still produces a compression factor of around $8000 \times$, relative to explicit storage of symmetric components of the Christoffel, if needed for any analysis purposes.

---

### Author Response · Authors · 2025-11-20

We thank the reviewers for their valuable feedback that helps us improve this work. We have strived to provide an early rebuttal in order to facilitate the discussion with the reviewers and leverage their feedback. We have attached a revised document with the changes highlighted in blue. These mainly concern additional evaluation on a numerical relativity usecase (appendix F and parts of main text), adjusted positioning, and other particular adjustments in response to questions, concerns, and feedback. We will refine the formatting of the revision, prioritising sharing the key content as early as possible.

---

### Author Response · Authors · 2025-11-30
**"Authors' comment post security incident"**

We are very sorry about the security incident. We fully align with the way forward, and want to summarize our (biased) view of the review process.

Our paper got reviews **884**. Especially reviewer kkPc asked for many additional experiments, most importantly for real world numerical relativity data generation. We provided the required additional experiment: which included a i) fixed mesh refinement (FMR) and static mesh refinement (SMR)-based NR simulation, which is qualitatively similar to an adaptive mesh refinement (AMR)-based simulations, at least from the neural field training and post-hoc analysis point of view, ii) clarified most of the questions and weaknesses, such as run-time/query speeds, precision of geodesic rollouts in FLOAT64, discretization-free clarification of implicit neural representations training (except Q2 on (pseudo)spectral baselines, which we have clearly mentioned in the future work/conclusion, given that the community has only relatively recently moved to these methods, while the primary workhorse until recently has been finite-differencing (FD) methodologies).

For these real world numerical experiments, we set up a full scale simulation using Einstein Toolkit open source NR software of an oscillatory evolution of a perturbed neutron star using the Baumgarte-Shapiro-Shibata-Nakamura (BSSN) framework. We have significantly updated the paper and uploaded the fully revised version of the manuscript, addressing all these aspects in detail.

Reviewer DSBV found the additions so important that they were willing to raise their score to **10** ("I am raising my score and suggesting this submission to be considered for **Spotlight**, due to its relevance for the AI4Science community.", see discussion).

Unfortunately, we didn't have the chance to engage in further discussions with reviewer kkPc and reviewer Lt1j.  The review process helped to improve our paper significantly and has strengthened our work with additions such as realistic NR experiments, error bars on pareto fronts and query speeds of EinFields, and by addition of copious related works.

We thank the AC in advance for their thoughtful consideration pertaining to the improved scores of **10 8 4**, the *Spotlight* recommendation, and mainly the additional set of experiments conducted to improve the manuscript.

---

### Meta-Review · Area_Chair_k4Xa · 2026-01-06

**Summary:**

This paper proposes a neural tensor-field parametrization of general-relativity metrics and the computation of geometric quantities via automatic differentiation, thereby compressing simulations while maintaining accuracy. Of the three reviewers, two gave high scores. In particular, one of them was raised to 10. On the other hand, the remaining reviewer gave a score of 4 and expressed doubts about the usefulness of this method in practical applications. While the other two reviewers gave high scores, this reviewer's feedback is detailed, and in my opinion, the concern regarding effectiveness remains. Overall, I recommend accepting this paper as a poster presentation.

**Reviewer Concerns:**

The main concern raised, particularly by the reviewer who gave a score of 4, is whether the contribution of this method is overstated. While the paper states a compression ratio of several thousand times, this comparison is primarily against the finite difference method and does not include comparisons with spectral methods or reduced order models, which are typical methods in this area. Another concern is that the results are mainly for FP32, and evaluations for FP64, which is necessary for practical applications, have not been performed.

Regarding the first concern, the authors conducted additional experiments, but these did not include comparisons with spectral methods or ROMs. Concerning FP64, they responded that it is “infeasible,” however, I suppose that perhaps this infeasibility is exactly the reviewer's concern.

**Reviewer Scores:**

Considering the above, this reviewer's concerns may not have been fully addressed, and hence the score may remain at 4. Given the high scores from the other two reviewers, the paper should be accepted; however, the limitations should also be carefully explained.

---

### Decision · Program_Chairs · 2026-01-26

Accept (Poster)